# Bias no more: high-probability data-dependent regret bounds for adversarial bandits and MDPs

**Chung-Wei Lee**
University of Southern California
leechung@usc.edu

**Haipeng Luo**
University of Southern California
haipengl@usc.edu

**Chen-Yu Wei**
University of Southern California
chenyu.wei@usc.edu

**Mengxiao Zhang**
University of Southern California
mengxiao.zhang@usc.edu

## Abstract

We develop a new approach to obtaining high probability regret bounds for online learning with bandit feedback against an adaptive adversary. While existing approaches all require carefully constructing optimistic and biased loss estimators, our approach uses standard unbiased estimators and relies on a simple increasing learning rate schedule, together with the help of logarithmically homogeneous self-concordant barriers and a strengthened Freedman's inequality.

Besides its simplicity, our approach enjoys several advantages. First, the obtained high-probability regret bounds are data-dependent and could be much smaller than the worst-case bounds, which resolves an open problem asked by Neu [31]. Second, resolving another open problem of Bartlett et al. [12] and Abernethy and Rakhlin [1], our approach leads to the first general and efficient algorithm with a high-probability regret bound for adversarial linear bandits, while previous methods are either inefficient or only applicable to specific action sets. Finally, our approach can also be applied to learning adversarial Markov Decision Processes and provides the first algorithm with a high-probability small-loss bound for this problem.

## 1 Introduction

Online learning with partial information in an adversarial environment, such as the non-stochastic Multi-armed Bandit (MAB) problem [10], is by now a well-studied topic. However, the majority of work in this area has been focusing on obtaining algorithms with sublinear *expected* regret bounds, and these algorithms can in fact be highly unstable and suffer a huge variance. For example, it is known that the classic EXP3 algorithm [10] for MAB suffers *linear regret with a constant probability* (over its internal randomness), despite having nearly optimal expected regret (see [26, Section 11.5, Note 1]), making it a clearly undesirable choice in practice.

To address this issue, a few works develop algorithms with regret bounds that hold with *high probability*, including those for MAB [10, 8, 31], linear bandits [12, 1], and even adversarial Markov Decision Processes (MDPs) [25]. Getting high-probability regret bounds is also the standard way of deriving guarantees against an *adaptive adversary* whose decisions can depend on learner's previous actions. This is especially important for problems such as routing in wireless networks (modeled as linear bandits in [11]) where adversarial attacks can indeed adapt to algorithm's decisions on the fly.

As far as we know, all existing high-probability methods (listed above) are based on carefully constructing *biased* loss estimators that enjoy smaller variance compared to standard unbiased ones. While this principle is widely applicable, the actual execution can be cumbersome; for example, the

scheme proposed in [1] for linear bandits needs to satisfy seven conditions (see their Theorem 4), and other than two examples with specific action sets, no general algorithm satisfying these conditions was provided.

In this work, we develop a new and simple approach to obtaining high-probability regret bounds that works for a wide range of bandit problems with an adaptive adversary (including MAB, linear bandits, MDP, and more). Somewhat surprisingly, in contrast to all previous methods, our approach uses standard *unbiased* loss estimators. More specifically, our algorithms are based on Online Mirror Descent with a self-concordant barrier regularizer [2], a standard approach with expected regret guarantees. The key difference is that we adopt an increasing learning rate schedule, inspired by several recent works using similar ideas for completely different purposes (e.g., [5]). At a high level, the effect of this schedule magically cancels the potentially large variance of the unbiased estimators.

Apart from its simplicity, there are several important advantages of our approach. First of all, our algorithms all enjoy *data-dependent regret bounds*, which could be much smaller than the majority of existing high-probability bounds in the form of $\tilde{\mathcal{O}}(\sqrt{T})$ where $T$ is the number of rounds. As a key example, we provide details for obtaining a particular kind of such bounds called "small-loss" bounds in the form $\tilde{\mathcal{O}}(\sqrt{L^\star})$, where $L^\star \leq T$ is the loss of the benchmark in the regret definition. For MAB and linear bandits, our approach also obtains bounds in terms of the variation of the environment in the vein of [23, 33, 37, 17], resolving an open problem asked by Neu [31].

Second, our approach provides the *first general and efficient* algorithm for adversarial linear bandits (also known as bandit linear optimization) with a high-probability regret guarantee. As mentioned, Abernethy and Rakhlin [1] provide a general recipe for this task but in the end only show concrete examples for two specific action sets. The problem of obtaining a general and efficient approach with regret $\tilde{\mathcal{O}}(\sqrt{T})$ was left open since then. The work of [12] proposes an inefficient but general approach, while the work of [22, 13] develop efficient algorithms for polytopes but with $\tilde{\mathcal{O}}(T^{2/3})$ regret. We not only resolve this long-standing open problem, but also provide improved data-dependent bounds.

Third, our approach is also applicable to learning episodic MDPs with unknown transition, adversarial losses, and bandit feedback. The algorithm is largely based on a recent work [25] on the same problem where a high-probability $\tilde{\mathcal{O}}(\sqrt{T})$ regret bound is obtained. We again develop the first algorithm with a high-probability small-loss bound $\tilde{\mathcal{O}}(\sqrt{L^\star})$ in this setting. The problem in fact shares great similarity with the simple MAB problem. However, none of the existing methods for obtaining small-loss bounds for MAB can be generalized to the MDP setting (at least not in a direct manner) as we argue in Section 4. Our approach, on the other hand, generalizes directly without much effort.

**Techniques.** Most new techniques of our work is in the algorithm for linear bandits (Section 3), which is based on the SCRIBLE algorithm from the seminal work [2, 3]. The first difference is that we propose to lift the problem from $\mathbb{R}^d$ to $\mathbb{R}^{d+1}$ (where $d$ is the dimension of the problem) and use a *logarithmically homogeneous* self-concordant barrier of the conic hull of the action set (which always exists) as the regularizer for Online Mirror Descent. The nice properties of such a regularizer lead to a smaller variance of the loss estimators. Equivalently, this can be viewed as introducing a new sampling scheme for the original SCRIBLE algorithm in the space of $\mathbb{R}^d$. The second difference is the aforementioned new learning rate schedule, where we increase the learning rate by a small factor whenever the Hessian of the regularizer at the current point is "large" in some sense.

In addition, we also provide a strengthened version of the Freedman's concentration inequality for martingales [21], which is crucial to all of our analysis and might be of independent interest.

**Related work.** In online learning, there are subtle but important differences and connections between the concept of pseudo-regret, expected regret, and the actual regret, in the context of either oblivious or adaptive adversary. We refer the readers to [8] for detailed related discussions.

While getting expected small-loss regret is common [7, 32, 20, 6, 4, 27], most existing high-probability bounds are of order $\tilde{\mathcal{O}}(\sqrt{T})$. Although not mentioned in the original paper, the idea of implicit exploration from [31] can lead to high-probability small-loss bounds for MAB (see [26, Section 12.3, Note 4]). Lykouris et al. [29] adopt this idea together with a clipping trick to derive small-loss bounds for more general bandit problems with graph feedback. We are not aware of other works with high-probability small-loss bounds in the bandit literature. Note that in [8, Section 6], some high-probability "small-reward" bounds are derived, and they are very different in nature from small-loss

bounds (specifically, the former is equivalent to $\tilde{\mathcal{O}}(\sqrt{T - L^\star})$ in our notation). We are also not aware of high-probability version of other data-dependent regret bounds such as those from [23, 33, 37, 17].

The idea of increasing learning rate was first used in the seminal work of Bubeck et al. [15] for convex bandits. Inspired by this work, Agarwal et al. [5] first combined this idea with the log-barrier regularizer for the problem of "corralling bandits". Since then, this particular combination has proven fruitful for many other problems [37, 28, 27]. We also use it for MAB and MDP, but our algorithm for linear bandits greatly generalizes this idea to any self-concordant barrier.

**Structure and notation.** In Section 2, we start with a warm-up example on MAB, which is the cleanest illustration on the idea of using increasing learning rates to control the variance of unbiased estimators. Then in Section 3 and Section 4, we greatly generalize the idea to linear bandits and MDPs respectively. We focus on showing small-loss bounds as the main example, and only briefly discuss how to obtain other data-dependent regret bounds, since the ideas are very similar.

We introduce the notation for each setting in the corresponding section, but will use the following general notation throughout the paper: for a positive integer $n$, $[n]$ represents the set $\{1, \ldots, n\}$ and $\Delta_n$ represents the $(n-1)$-dimensional simplex; $e_i$ stands for the $i$-th standard basis vector and $\mathbf{1}$ stands for the all-one vector (both in an appropriate dimension depending on the context); for a convex function $\psi$, the associated Bregman divergence is $D_\psi(u, w) = \psi(u) - \psi(w) - \nabla\psi(w)^\top(u - w)$; for a positive definite matrix $M \in \mathbb{R}^{d \times d}$ and a vector $u \in \mathbb{R}^d$, $\|u\|_M \triangleq \sqrt{u^\top M u}$ is the quadratic norm of $u$ with respect to $M$; $\lambda_{\max}(M)$ denotes the largest eigenvalue of $M$; $\mathbb{E}_t[\cdot]$ is a shorthand for the conditional expectation given the history before round $t$; $\tilde{\mathcal{O}}(\cdot)$ hides logarithmic terms.

## 2 Multi-armed bandits: an illustrating example

We start with the most basic bandit problem, namely adversarial MAB [10], to demonstrate the core idea of using increasing learning rate to reduce the variance of standard algorithms. The MAB problem proceeds in rounds between a learner and an adversary. For each round $t = 1, \ldots, T$, the learner selects one of the $d$ available actions $i_t \in [d]$, while simultaneously the adversary decides a loss vector $\ell_t \in [0, 1]^d$ with $\ell_{t,i}$ being the loss for arm $i$. An adaptive adversary can choose $\ell_t$ based on the learner's previous actions $i_1, \ldots, i_{t-1}$ in an arbitrary way, while an oblivious adversary cannot and essentially decides all $\ell_t$'s ahead of time (knowing the learner's algorithm). At the end of round $t$, the learner observes the loss of the chosen arm $\ell_{t,i_t}$ and nothing else. The standard measure of the learner's performance is the regret, defined as $\text{Reg} = \sum_{t=1}^T \ell_{t,i_t} - \min_{i \in [d]} \sum_{t=1}^T \ell_{t,i}$, that is, the difference between the total loss of the learner and that of the best fixed arm in hindsight.

A standard framework to solve this problem is Online Mirror Descent (OMD), which at time $t$ samples $i_t$ from a distribution $w_t$, updated in the following recursive form: $w_{t+1} = \text{argmin}_{w \in \Delta_d} \langle w, \widehat{\ell}_t \rangle + D_{\psi_t}(w, w_t)$, where $\psi_t$ is the regularizer and $\widehat{\ell}_t$ is an estimator for $\ell_t$. The standard estimator is the importance-weighted estimator: $\widehat{\ell}_{t,i} = \ell_{t,i}\mathbb{1}\{i_t = i\}/w_{t,i}$, which is clearly unbiased. Together with many possible choices of the regularizer (e.g., the entropy regularizer recovering EXP3 [10]), this ensures (nearly) optimal expected regret bound $\mathbb{E}[\text{Reg}] = \tilde{\mathcal{O}}(\sqrt{dT})$ against an oblivious adversary.

To obtain high-probability regret bounds (and also as a means to deal with adaptive adversary), various more sophisticated loss estimators have been proposed. Indeed, the key challenge in obtaining high-probability bounds lies in the potentially large variance of the unbiased estimators: $\mathbb{E}_t[\widehat{\ell}_{t,i}^2] = \ell_{t,i}^2/w_{t,i}$ is huge if $w_{t,i}$ is small. The idea of all existing approaches to addressing this issue is to introduce a slight bias to the estimator, making it an optimistic underestimator of $\ell_t$ with lower variance (see e.g., [10, 8, 31]). Carefully balancing the bias and variance, these algorithms achieve $\text{Reg} = \tilde{\mathcal{O}}(\sqrt{dT \ln(d/\delta)})$ with probability at least $1 - \delta$ against an adaptive adversary.

**Our algorithm.** In contrast to all these existing approaches, we next show that, perhaps surprisingly, using the standard unbiased estimator can also lead to the same (in fact, an even better) high-probability regret bound. We start by choosing a particular regularizer called log-barrier with time-varying and individual learning rate $\eta_{t,i}$: $\psi_t(w) = \sum_{i=1}^d \frac{1}{\eta_{t,i}} \ln \frac{1}{w_i}$, which is a self-concordant barrier for the positive orthant [30] and has been used for MAB in several recent works [20, 5, 16, 37, 17]. As mentioned in Section 1, the combination of log-barrier and a particular increasing learning rate

---
**Algorithm 1** OMD with log-barrier and increasing learning rates for Multi-armed Bandits
---
**Input:** initial learning rate $\eta$.
**Define:** increase factor $\kappa = e^{\frac{1}{\ln T}}$, truncated simplex $\Omega = \left\{ w \in \Delta_d : w_i \geq \frac{1}{T}, \forall i \in [d] \right\}$.
**Initialize:** for all $i \in [d]$, $w_{1,i} = 1/d$, $\rho_{1,i} = 2d$, $\eta_{1,i} = \eta$.
**for** $t = 1, 2, \ldots, T$ **do**
> Sample $i_t \sim w_t$, observe $\ell_{t,i_t}$, and construct estimator $\widehat{\ell}_{t,i} = \frac{\ell_{t,i} \mathbb{1}\{i_t = i\}}{w_{t,i}}$ for all $i \in [d]$.
> Compute $w_{t+1} = \operatorname{argmin}_{w \in \Omega} \left\langle w, \widehat{\ell}_t \right\rangle + D_{\psi_t}(w, w_t)$ where $\psi_t(w) = \sum_{i=1}^d \frac{1}{\eta_{t,i}} \ln \frac{1}{w_i}$.
> **for** $i \in [d]$ **do**
>> **if** $\frac{1}{w_{t+1,i}} > \rho_{t,i}$ **then** set $\rho_{t+1,i} = \frac{2}{w_{t+1,i}}, \eta_{t+1,i} = \eta_{t,i}\kappa$;
>> **else** set $\rho_{t+1,i} = \rho_{t,i}, \eta_{t+1,i} = \eta_{t,i}$.
---

schedule has been proven powerful for many different problems since the work of [5], which we also apply here. Specifically, the learning rates start with a fixed value $\eta_{1,i} = \eta$ for all arm $i \in [d]$, and every time the probability of selecting an arm $i$ is too small, in the sense that $1/w_{t+1,i} > \rho_{t,i}$ for some threshold $\rho_{t,i}$ (starting with $2d$), we set the new threshold to be $2/w_{t+1,i}$ and increase the corresponding learning rate $\eta_{t,i}$ by a small factor $\kappa$.

The complete pseudocode is shown in Algorithm 1. The only slight difference compared to the algorithm of [5] is that instead of enforcing a $1/T$ amount of uniform exploration explicitly (which makes sure that each learning rate is increased by a most $\mathcal{O}(\ln T)$ times), we directly perform OMD over a truncated simplex $\Omega = \{w \in \Delta_d : w_i \geq 1/T, \forall i \in [d]\}$, making the analysis cleaner.

As explained in [5], increasing the learning rate in this way allows the algorithm to quickly realize that some arms start to catch up even though they were underperforming in earlier rounds, which is also the hardest case in our context of obtaining high-probability bounds because these arms have low-quality estimators at some point. At a technical level, this effect is neatly presented through a *negative* term in the regret bound, which we summarize below.

**Lemma 2.1.** Algorithm 1 ensures $\sum_{t=1}^T \ell_{t,i_t} - \sum_{t=1}^T \left\langle u, \widehat{\ell}_t \right\rangle \leq \mathcal{O}\left( \frac{d \ln T}{\eta} + \eta \sum_{t=1}^T \ell_{t,i_t} \right) - \frac{\langle \rho_T, u \rangle}{10 \eta \ln T}$ for any $u \in \Omega$.

The important part is the last negative term involving the last threshold $\rho_T$ whose magnitude is large whenever an arm has a small sampling probability at some point over the $T$ rounds. This bound has been proven in previous works such as [5] (see a proof in Appendix A.2), and next we use it to show that the algorithm in fact enjoys a high-probability regret bound, which is not discovered before.

Indeed, comparing Lemma 2.1 with the definition of regret, one sees that as long as we can relate the estimated loss of the benchmark $\sum_t \left\langle u, \widehat{\ell}_t \right\rangle$ with its true loss $\sum_t \langle u, \ell_t \rangle$, then we immediately obtain a regret bound by setting $u = (1 - \frac{d}{T})e_{i^\star} + \frac{1}{T}\mathbf{1} \in \Omega$ where $i^\star = \operatorname{argmin}_i \sum_t \ell_{t,i}$ is the best arm. A natural approach is to apply standard concentration inequality, in particular Freedman's inequality [21], to the martingale difference sequence $\left\langle u, \widehat{\ell}_t - \ell_t \right\rangle$. The deviation from Freedman's inequality is in terms of the variance of $\left\langle u, \widehat{\ell}_t \right\rangle$, which in turn depends on $\sum_i u_i/w_{t,i}$. As explained earlier, the negative term is exactly related to this and can thus cancel the potentially large variance!

One caveat, however, is that the deviation from Freedman's inequality also depends on a *fixed* upper bound of the random variable $\left\langle u, \widehat{\ell}_t \right\rangle \leq \sum_i u_i/w_{t,i}$, which could be as large as $T$ (since $w_{t,i} \geq 1/T$) and ruin the bound. If the dependence on such a fixed upper bound could be replaced with the (random) upper bound $\sum_i u_i/w_{t,i}$, then we could again use the negative term to cancel this dependence. Fortunately, since $\sum_i u_i/w_{t,i}$ is measurable with respect to the $\sigma$-algebra generated by everything *before* round $t$, we are indeed able to do so. Specifically, we develop the following strengthened version of Freedman's inequality, which might be of independent interest.

**Theorem 2.2.** *Let $X_1, \ldots, X_T$ be a martingale difference sequence with respect to a filtration $\mathcal{F}_1 \subseteq \cdots \subseteq \mathcal{F}_T$ such that $\mathbb{E}[X_t | \mathcal{F}_t] = 0$. Suppose $B_t \in [1, b]$ for a fixed constant $b$ is $\mathcal{F}_t$-measurable and such that $X_t \leq B_t$ holds almost surely. Then with probability at least $1 - \delta$ we have $\sum_{t=1}^T X_t \leq C\left( \sqrt{8V \ln(C/\delta)} + 2B^\star \ln(C/\delta) \right)$, where $V = \max\left\{ 1, \sum_{t=1}^T \mathbb{E}[X_t^2 | \mathcal{F}_t] \right\}$, $B^\star = \max_{t \in [T]} B_t$, and $C = \lceil \log(b) \rceil \lceil \log(b^2 T) \rceil$.*

This strengthened Freedman's inequality essentially recovers the standard one when $B_t$ is a fixed quantity. In our application, $B_t$ is exactly $\langle \rho_t, u \rangle$ which is $\mathcal{F}_t$-measurable. With the help of this concentration result, we are now ready to show the high-probability guarantee of Algorithm 1.

**Theorem 2.3.** *Algorithm 1 with a suitable choice of $\eta$ ensures that with probability at least $1 - \delta$,* $\text{Reg} = \widetilde{\mathcal{O}}\big(\sqrt{dL^\star \ln(d/\delta)} + d \ln(d/\delta)\big)$, *where $L^\star = \min_i \sum_{t=1}^T \ell_{t,i}$ is the loss of the best arm.*

The proof is a direct combination of Lemma 2.1 and Theorem 2.2 and can be found in Appendix A.3. Our high-probability guarantee is of the same order $\tilde{\mathcal{O}}(\sqrt{dT \ln(d/\delta)})$ as in previous works [10, 8] since $L^\star = \mathcal{O}(T)$. However, as long as $L^\star = o(T)$ (that is, the best arm is of high quality), our bound becomes much better. This kind of high-probability small-loss bounds appears before (e.g., [29]). Nevertheless, in Section 4 we argue that only our approach can directly generalize to learning MDPs.

Finally, we remark that the same algorithm can also obtain other data-dependent regret bounds by changing the estimator to $\widehat{\ell}_{t,i} = (\ell_{t,i} - m_{t,i}) \mathbb{1}\{i_t = i\}/w_{t,i} + m_{t,i}$ for some optimistic prediction $m_t$. We refer the reader to [37] for details on how to set $m_t$ in terms of observed data and what kind of bounds this leads to, but the idea of getting the high-probability version is completely the same as what we have illustrated here. This resolves an open problem mentioned in [31, Section 5].

## 3  Generalization to adversarial linear bandits

Next, we significantly generalize our approach to adversarial linear bandits, which is the main algorithmic contribution of this work. Linear bandits generalize MAB from the simplex decision set $\Delta_d$ to an arbitrary convex body $\Omega \subseteq \mathbb{R}^d$. For each round $t = 1, \ldots, T$, the learner selects an action $\widetilde{w}_t \in \Omega$ while simultaneously the adversary decides a loss vector $\ell_t \in \mathbb{R}^d$, assumed to be normalized such that $\max_{w \in \Omega} |\langle w, \ell_t \rangle| \leq 1$. Again, an adaptive adversary can choose $\ell_t$ based on the learner's previous actions, while an oblivious adversary cannot. At the end of round $t$, the learner suffers and only observes loss $\langle \widetilde{w}_t, \ell_t \rangle$. The regret of the learner is defined as $\text{Reg} = \max_{u \in \Omega} \sum_{t=1}^T \langle \widetilde{w}_t - u, \ell_t \rangle$, which is the difference between the total loss of the learner and that of the best fixed action within $\Omega$. Linear bandits subsume many other well-studied problems such as online shortest path for network routing, online matching, and other combinatorial bandit problems (see e.g., [9, 18]).

The seminal work of Abernethy et al. [2] develops the first general and efficient linear bandit algorithm (called SCRIBLE in its journal version [3]) with expected regret $\tilde{\mathcal{O}}(d\sqrt{\nu T})$ (against an oblivious adversary), which uses a $\nu$-self-concordant barrier as the regularizer for OMD. It is known that any convex body in $\mathbb{R}^d$ admits a $\nu$-self-concordant barrier with $\nu = \mathcal{O}(d)$ [30]. The minimax regret of this problem is known to be of order $\tilde{\mathcal{O}}(d\sqrt{T})$ [19, 14], but efficiently achieving this bound (in expectation) requires a log-concave sampler and a volumetric spanner of $\Omega$ [24].

High-probability bounds for linear bandits are very scarce, especially for a general decision set $\Omega$. In [12], an algorithm with high-probability regret $\tilde{\mathcal{O}}(\sqrt{d^3 T} \ln(1/\delta))$ was developed, but it cannot be implement efficiently. In [1], a general recipe was provided, but seven conditions need to be satisfied to arrive at a high-probability guarantee, and only two concrete examples were shown (when $\Omega$ is the simplex or the Euclidean ball). We propose a new algorithm based on SCRIBLE, which is the first general and efficient linear bandit algorithm with a high-probability regret guarantee, resolving the problem left open since the work of [12, 1].

**Issues of SCRIBLE.** To introduce our algorithm, we first review SCRIBLE. As mentioned, it is also based on OMD and maintains a sequence $w_1, \ldots, w_T \in \Omega$ updated as $w_{t+1} = \operatorname{argmin}_{w \in \Omega} \langle w, \widehat{\ell}_t \rangle + \frac{1}{\eta} D_\psi(w, w_t)$ where $\widehat{\ell}_t$ is an estimator for $\ell_t$, $\eta$ is some learning rate, and importantly, $\psi$ is a $\nu$-self-concordant barrier for $\Omega$ which, again, always exists. Due to space limit, we defer the definition and properties of self-concordant barriers to Appendix B.2. To incorporate exploration, the actual point played by the algorithm at time $t$ is $\widetilde{w}_t = w_t + H_t^{-1/2} s_t$ where $H_t = \nabla^2 \psi(w_t)$ and $s_t$ is uniformly randomly sampled from the $d$-dimensional unit sphere $\mathbb{S}^d$.[1] The point $\widetilde{w}_t$ is on the boundary of the *Dikin ellipsoid* centered at $w_t$ (defined as $\{w : \|w - w_t\|_{H_t} \leq 1\}$)

**Algorithm 2** SCRIBLE with lifting and increasing learning rates

---

**Input:** decision set $\Omega \subseteq \mathbb{R}^d$, a $\nu$-self-concordant barrier $\psi$ for $\Omega$, initial learning rate $\eta$.

**Define:** increase factor $\kappa = e^{\frac{1}{100d\ln(\nu T)}}$, normal barrier $\Psi(\boldsymbol{w}) = \Psi(w, b) = 400\left(\psi\left(\frac{w}{b}\right) - 2\nu\ln b\right)$.

**Initialize:** $w_1 = \operatorname{argmin}_{w\in\Omega}\psi(w)$, $\boldsymbol{w_1} = (w_1, 1)$, $\boldsymbol{H_1} = \nabla^2\Psi(\boldsymbol{w_1})$, $\eta_1 = \eta$, $\mathcal{S} = \{1\}$.

**Define:** shrunk lifted decision set $\boldsymbol{\Omega}' = \{\boldsymbol{w} = (w, 1) : w \in \Omega, \pi_{w_1}(w) \le 1 - \frac{1}{T}\}$.

1 **for** $t = 1, 2, \ldots, T$ **do**

2      Uniformly at random sample $\boldsymbol{s}_t$ from $\left(\boldsymbol{H}_t^{-\frac{1}{2}}\boldsymbol{e}_{d+1}\right)^{\perp} \cap \mathbb{S}^{d+1}$.

3      Compute $\widetilde{\boldsymbol{w}}_t = \boldsymbol{w}_t + \boldsymbol{H}_t^{-\frac{1}{2}}\boldsymbol{s}_t \triangleq (\widetilde{w}_t, 1)$.

4      Play $\widetilde{w}_t$, observe loss $\langle\widetilde{w}_t, \ell_t\rangle$, and construct loss estimator $\widehat{\boldsymbol{\ell}}_t = d\langle\widetilde{w}_t, \ell_t\rangle\boldsymbol{H}_t^{\frac{1}{2}}\boldsymbol{s}_t$.

5      Compute $\boldsymbol{w}_{t+1} = \operatorname{argmin}_{\boldsymbol{w}\in\boldsymbol{\Omega}'}\langle\boldsymbol{w}, \widehat{\boldsymbol{\ell}}_t\rangle + D_{\Psi_t}(\boldsymbol{w}, \boldsymbol{w}_t)$, where $\Psi_t = \frac{1}{\eta_t}\Psi$.

6      Compute $\boldsymbol{H}_{t+1} = \nabla^2\Psi(\boldsymbol{w}_{t+1})$.

7      **if** $\lambda_{\max}(\boldsymbol{H}_{t+1} - \sum_{\tau\in\mathcal{S}}\boldsymbol{H}_\tau) > 0$ **then** $\mathcal{S} \leftarrow \mathcal{S} \cup \{t+1\}$ and set $\eta_{t+1} = \eta_t\kappa$;

8      **else** set $\eta_{t+1} = \eta_t$.

---

and is known to be always within $\Omega$. Finally, the estimator $\widehat{\ell}_t$ is constructed as $d\langle\widetilde{w}_t, \ell_t\rangle H_t^{1/2}s_t$, which can be computed using only the feedback $\langle\widetilde{w}_t, \ell_t\rangle$ and is unbiased as one can verify.

The analysis of [2] shows the following bound related to the loss estimators: $\sum_{t=1}^T \langle w_t - u, \widehat{\ell}_t\rangle \le \tilde{\mathcal{O}}(\frac{\nu}{\eta} + \eta d^2 T)$ for any $u \in \Omega$ (that is not too close to the boundary). Since $\mathbb{E}_t[\langle w_t - u, \widehat{\ell}_t\rangle] = \mathbb{E}_t[\langle\widetilde{w}_t - u, \ell_t\rangle]$, this immediately yields an expected regret bound (for an oblivious adversary). However, to obtain a high-probability bound, one needs to consider the deviation of $\sum_{t=1}^T \langle w_t - u, \widehat{\ell}_t\rangle$ from $\sum_{t=1}^T \langle\widetilde{w}_t - u, \ell_t\rangle$. Applying our strengthened Freedman's inequality (Theorem 2.2) with $X_t = \langle\widetilde{w}_t - u, \ell_t\rangle - \langle w_t - u, \widehat{\ell}_t\rangle$, with some direct calculations one can see that both the variance term $V$ and the range term $B^\star$ from the theorem are related to $\max_t\|w_t\|_{H_t}$ and $\max_t\|u\|_{H_t}$, both of which can be prohibitively large. We next discuss how to control each of these two terms, leading to the two new ideas of our algorithm (see Algorithm 2).

**Controlling $\|w_t\|_{H_t}$.** Readers who are familiar with self-concordant functions would quickly realize that $\|w_t\|_{H_t} = \sqrt{w_t^\top\nabla^2\psi(w_t)w_t}$ is simply $\sqrt{\nu}$ provided that $\psi$ is also *logarithmically homogeneous*. A logarithmically homogeneous self-concordant barrier is also called a *normal barrier* (see Appendix B.2 for formal definitions and related properties). However, normal barriers are only defined for cones instead of convex bodies.

Inspired by this fact, we propose to *lift the problem to* $\mathbb{R}^{d+1}$. To make the notation clear, we use bold letters for vectors in $\mathbb{R}^{d+1}$ and matrices in $\mathbb{R}^{(d+1)\times(d+1)}$. The lifting is done by operating over a lifted decision set $\boldsymbol{\Omega} = \{\boldsymbol{w} = (w, 1) \in \mathbb{R}^{d+1} : w \in \Omega\}$, that is, we append a dummy coordinate with value 1 to all actions. The conic hull of this set is $\mathcal{K} = \{(w, b) : w \in \mathbb{R}^d, b \ge 0, \frac{1}{b}w \in \Omega\}$. We then perform OMD over the lifted decision set but with a normal barrier defined over the cone $\mathcal{K}$ as the regularizer to produce the sequence $\boldsymbol{w}_1, \ldots, \boldsymbol{w}_T$ (Line 5). In particular, using the original regularizer $\psi$ we construct the normal barrier as: $\Psi(w, b) = 400\left(\psi\left(\frac{w}{b}\right) - 2\nu\ln b\right)$.[2] Indeed, Proposition 5.1.4 of [30] asserts that this is a normal barrier for $\mathcal{K}$ with self-concordant parameter $\mathcal{O}(\nu)$.

So far nothing really changes since $\Psi(w, 1) = 400\psi(w)$. However, the key difference is in the way we sample the point $\widetilde{\boldsymbol{w}}_t$. If we still follow SCRIBLE to sample from the Dikin ellipsoid centered at $\boldsymbol{w}_t$, it is possible that the sampled point leaves $\boldsymbol{\Omega}$. To avoid this, it is natural to sample only the intersection of the Dikin ellipsoid and $\boldsymbol{\Omega}$ (again an ellipsoid). Algebraically, this means setting $\widetilde{\boldsymbol{w}}_t = \boldsymbol{w}_t + \boldsymbol{H}_t^{-1/2}\boldsymbol{s}_t$ where $\boldsymbol{H}_t = \nabla^2\Psi(\boldsymbol{w}_t)$ and $\boldsymbol{s}_t$ is sampled uniformly at random from $(\boldsymbol{H}_t^{-1/2}\boldsymbol{e}_{d+1})^{\perp} \cap \mathbb{S}^{d+1}$ ($v^{\perp}$ is the space orthogonal to $v$). Indeed, since $\boldsymbol{s}_t$ is orthogonal to $\boldsymbol{H}_t^{-1/2}\boldsymbol{e}_{d+1}$, the last coordinate of $\boldsymbol{H}_t^{-1/2}\boldsymbol{s}_t$ is zero, making $\widetilde{\boldsymbol{w}}_t = (\widetilde{w}_t, 1)$ stay in $\boldsymbol{\Omega}$. See Line 2 and Line 3. To sample $\boldsymbol{s}_t$ efficiently, one can either sample a vector uniformly randomly from $\mathbb{S}^{d+1}$,

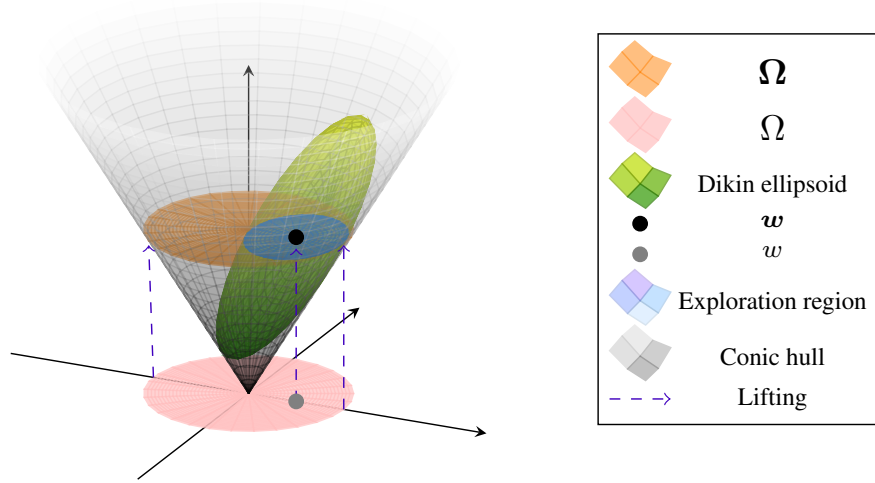

Figure 1: An illustration of the concept of lifting, the conic hull, and the Dikin ellipsoid. In this example $d$ is 2, and the pink disk at the bottom is the original decision set $\Omega$. The gray dot $w$ is a point in $\Omega$. In Algorithm 2, we lift the problem from $\mathbb{R}^2$ to $\mathbb{R}^3$, and obtain the lifted, orange, decision set $\boldsymbol{\Omega}$. For example, $w$ is lifted to the black dot $\boldsymbol{w} = (w, 1)$. Then we construct the conic hull of the lifted decision set, that is, the gray cone, and construct a normal barrier for this conic hull. By Lemma B.1, the Dikin ellipsoid centered at $\boldsymbol{w}$ of this normal barrier (the green ellipsoid), is alway within the cone. In Algorithm 2, if $\boldsymbol{w}$ is the OMD iterate, we explore and play an action within the intersection of $\boldsymbol{\Omega}$ and the Dikin ellipsoid centered at $\boldsymbol{w}$, that is, the (boundary of) the blue ellipse.

project it onto the subspace perpendicular to $\boldsymbol{H}_t^{-1/2} \boldsymbol{e}_{d+1}$, and then normalize; or sample a vector $s_t$ uniformly randomly from $\mathbb{S}^d$, then normalize $\boldsymbol{H}_t^{\frac{1}{2}} (s_t^\top, 0)^\top$ to obtain $\boldsymbol{s}_t$.

Finally, after playing $\widetilde{w}_t$ and observing $\langle \widetilde{w}_t, \ell_t \rangle$, we construct the loss estimator the same way as SCRIBLE: $\widehat{\boldsymbol{\ell}}_t = d \langle \widetilde{w}_t, \ell_t \rangle \boldsymbol{H}_t^{1/2} \boldsymbol{s}_t$ (Line 4). Lemma B.9 shows that the first $d$ coordinates of $\widehat{\boldsymbol{\ell}}_t$ is indeed an unbiased estimator of $\ell_t$. This makes the entire analysis of SCRIBLE hold in $\mathbb{R}^{d+1}$, but now the key term $\|\boldsymbol{w}_t\|_{\boldsymbol{H}_t}$ we want to control is *exactly* $20\sqrt{2\nu}$ (see Lemma B.5 and Lemma B.6)!

We provide an illustration of the lifting idea in Figure 1. One might ask whether this lifting is necessary; indeed, one can also spell out the algorithm in $\mathbb{R}^d$ (see Appendix B.1). Importantly, compared to SCRIBLE, the key difference is still that the sampling scheme has changed: the sampled point is not necessarily on the Dikin ellipsoid with respect to $\psi$. In other words, another view of our algorithm is that it is SCRIBLE with a new sampling scheme. We emphasize that, however, it is important (or at least much cleaner) to perform the analysis in $\mathbb{R}^{d+1}$. In fact, even in Algorithm 1 for MAB, similar lifting implicitly happens already since $\Delta_d$ is a convex body in dimension $d-1$ instead of $d$, and log-barrier is indeed a canonical normal barrier for the positive orthant.

**Controlling** $\|u\|_{H_t}$. Next, we discuss how to control the term $\|u\|_{H_t}$, or rather $\|\boldsymbol{u}\|_{\boldsymbol{H}_t}$ after the lifting. This term is the analogue of $\sum_i \frac{u_i}{w_{t,i}}$ for the case of MAB, and our goal is again to cancel it with the negative term introduced by increasing the learning rate. Indeed, a closer look at the OMD analysis reveals that increasing the learning rate at the end of time $t$ brings a negative term involving $-D_\Psi(\boldsymbol{u}, \boldsymbol{w}_{t+1})$ in the regret bound. In Lemma B.13, we show that this negative term is upper bounded by $-\|\boldsymbol{u}\|_{\boldsymbol{H}_{t+1}} + 800\nu \ln(800\nu T + 1)$, making the canceling effect possible.

It just remains to figure out when to increase the learning rate and how to make sure we only increase it logarithmic (in $T$) times as in the case for MAB. Borrowing ideas from Algorithm 1, intuitively one should increase the learning rate only when $\boldsymbol{H}_t$ is "large" enough, but the challenge is how to

measure this quantitatively. Only looking at the eigenvalues of $\boldsymbol{H}_t$, a natural idea, does not work as it does not account for the fact that the directions of eigenvectors are changing over time.

Instead, we propose the following condition: at the end of time $t$, increase the learning rate by a factor of $\kappa$ if $\lambda_{\max}(\boldsymbol{H}_{t+1} - \sum_{\tau \in \mathcal{S}} \boldsymbol{H}_\tau) > 0$, with $\mathcal{S}$ containing all the previous time steps prior to time $t$ where the learning rate was increased (Line 7). First, note that this condition makes sure that we always have enough negative terms to cancel $\max_t \|\boldsymbol{u}\|_{\boldsymbol{H}_t}$. Indeed, suppose $t$ is the time with the largest $\|\boldsymbol{u}\|_{\boldsymbol{H}_{t+1}}$. If we have increased the learning rate at time $t$, then the introduced negative term exactly matches $\|\boldsymbol{u}\|_{\boldsymbol{H}_{t+1}}$ as mentioned above; otherwise, the condition did not hold and by definition we have $\|\boldsymbol{u}\|_{\boldsymbol{H}_{t+1}} \leq \sqrt{\sum_{\tau \in \mathcal{S}} \|\boldsymbol{u}\|_{\boldsymbol{H}_s}^2} \leq \sum_{\tau \in \mathcal{S}} \|\boldsymbol{u}\|_{\boldsymbol{H}_\tau}$, meaning that the negative terms introduced in previous steps are already enough to cancel $\|\boldsymbol{u}\|_{\boldsymbol{H}_{t+1}}$.

Second, in Lemma B.12 we show that this schedule indeed makes sure that the learning rate is increased by only $\tilde{\mathcal{O}}(d)$ times. The key idea is to prove that $\det(\sum_{\tau \in \mathcal{S}} \boldsymbol{H}_\tau)$ is at least doubled each time we add one more time step to $\mathcal{S}$. Thus, if the eigenvalues of $\boldsymbol{H}_t$ are bounded, $|\mathcal{S}|$ cannot be too large. Ensuring the last fact requires a small tweak to the OMD update (Line 5), where we constrain the optimization over a slightly shrunk version of $\boldsymbol{\Omega}$ defined as $\boldsymbol{\Omega}' = \{\boldsymbol{w} \in \boldsymbol{\Omega} : \pi_{\boldsymbol{w}_1}(\boldsymbol{w}) \leq 1 - \frac{1}{T}\}$. Here, $\pi$ is the Minkowsky function and we defer its formal definition to Appendix B.2, but intuitively $\boldsymbol{\Omega}'$ is simply obtained by shrinking the lifted decision set by a small amount of $1/T$ with respect to the center $\boldsymbol{w}_1$ (which is the analogue of the truncated simplex for MAB). This makes sure that $\boldsymbol{w}_t$ is never too close to the boundary, and in turn makes sure that the eigenvalues of $\boldsymbol{H}_t$ are bounded.

This concludes the two main new ideas of our algorithm; see Algorithm 2 for the complete pseudocode. Clearly, our algorithm can be implemented as efficiently as the original SCRIBLE. The regret guarantee is summarized below.

**Theorem 3.1.** *Algorithm 2 with a suitable choice of $\eta$ ensures that with probability at least $1 - \delta$:*

$$\text{Reg} = \begin{cases} \tilde{\mathcal{O}}\left(d^2\nu\sqrt{T\ln\frac{1}{\delta}} + d^2\nu\ln\frac{1}{\delta}\right), & \text{against an oblivious adversary;} \\ \tilde{\mathcal{O}}\left(d^2\nu\sqrt{dT\ln\frac{1}{\delta}} + d^3\nu\ln\frac{1}{\delta}\right), & \text{against an adaptive adversary.} \end{cases}$$

*Moreover, if $\langle w, \ell_t \rangle \geq 0$ for all $w \in \Omega$ and all $t$, then $T$ in the bounds above can be replaced by $L^\star = \min_{u \in \Omega} \sum_{t=1}^T \langle u, \ell_t \rangle$, that is, the total loss of the best action.*

Our results are the first general high-probability regret guarantees achieved by an efficient algorithm (for either oblivious or adaptive adversary). We not only achieve $\sqrt{T}$-type bounds, but also improve it to $\sqrt{L^\star}$-type small-loss bounds, which does not exist before. Note that the latter holds only when losses are nonnegative, which is a standard setup for small-loss bounds and is true, for instance, for all combinatorial bandit problems where $\Omega \subseteq [0,1]^d$ lives in the positive orthant. Similarly to MAB, we can also obtain other data-dependent regret bounds by only changing the estimator to $d\langle \widetilde{w}_t, \ell_t - m_t \rangle H_t^{1/2} s_t + m_t$ for some predictor $m_t$ (see [33, 17]).[3]

Ignoring lower order terms, our bound for oblivious adversaries is $d\sqrt{\nu}$ times worse than the expected regret of SCRIBLE. For adaptive adversary, we pay extra dependence on $d$, which is standard since an extra union bound over $u$ is needed and is discussed in [1] as well. The minimax regret for adaptive adversary is still unknown. Reducing the dependence on $d$ for both cases is a key future direction.

## 4 Generalization to adversarial MDPs

Finally, we briefly discuss how to generalize Algorithm 1 for MAB to learning adversarial Markov Decision Processes (MDPs), leading to the first algorithm with a high-probability small-loss regret guarantee for this problem. We consider an episodic MDP setting with finite horizon, unknown transition kernel, bandit feedback, and adversarial losses, the exact same setting as the recent work [25] (which is the state-of-the-art for adversarial tabular MDPs; see [25] for related work).

Specifically, the problem is parameterized by a state space $X$, an action space $A$, and an unknown transition kernel $P : X \times A \times X \to [0,1]$ with $P(x'|x,a)$ being the probability of reaching state $x'$

after taking action $a$ at state $x$. Without loss of generality (see discussions in [25]), the state space is assumed to be partitioned into $J + 1$ layers $X_0, \ldots, X_J$ where $X_0 = \{x_0\}$ and $X_J = \{x_J\}$ contain only the start and end state respectively, and transitions are only possible between consecutive layers.

The learning proceeds in $T$ rounds/episodes. In each episode $t$, the learner starts from state $x_0$ and decides a stochastic policy $\pi_t : X \times A \to [0,1]$, where $\pi_t(a|x)$ is the probability of selecting action $a$ at state $x$. Simultaneously, the adversary decides a loss function $\ell_t : X \times A \to [0,1]$, with $\ell_t(x,a)$ being the loss of selecting action $a$ at state $x$. Once again, an adaptive adversary chooses $\ell_t$ based on all learner's actions in previous episodes, while an oblivious adversary chooses $\ell_t$ only knowing the learner's algorithm. Afterwards, the learner executes the policy in the MDP for $J$ steps and generates/observes a state-action-loss sequence $(x_0, a_0, \ell_t(x_0, a_0)), \ldots, (x_{J-1}, a_{J-1}, \ell_t(x_{J-1}, a_{J-1}))$ before reaching the final state $x_J$. With a slight abuse of notation, we use $\ell_t(\pi) = \mathbb{E}\left[\sum_{k=1}^{J-1} \ell_t(x_k, a_k) \mid P, \pi\right]$ to denote the expected loss of executing policy $\pi$ in episode $t$. The regret of the learner is then defined as $\text{Reg} = \sum_{t=1}^{T} \ell_t(\pi_t) - \min_\pi \sum_{t=1}^{T} \ell_t(\pi)$, where the min is over all possible policies.

Based on several prior works [38, 35], Jin et al. [25] showed the deep connection between this problem and adversarial MAB. In fact, with the help of the "occupancy measure" concept, this problem can be reformulated in a way that becomes very much akin to adversarial MAB and can be essentially solved using OMD with some importance-weighted estimators. We refer the reader to [25] and Appendix C.1 for details. The algorithm of [25] achieves $\text{Reg} = \tilde{\mathcal{O}}(J|X|\sqrt{|A|T})$ with high probability.

Since the problem has great similarity with MAB, the natural idea to improve the bound to a small-loss bound is to borrow techniques from MAB. Prior to our work, obtaining high-probability small-loss bounds for MAB can only be achieved by either the implicit exploration idea from [31] or the clipping idea from [7, 29]. Unfortunately, in Appendix C.4, we argue that neither of them works for MDPs, at least not in a direct way we can see, from perspectives of both the algorithm and the analysis.

On the other hand, our approach from Algorithm 1 immediately generalizes to MDPs without much effort. Compared to the algorithm of [25], the only essential differences are to replace their regularizer with log-barrier and to apply a similar increasing learning rate schedule. Due to space limit, we defer the algorithm to Appendix C.2 and show the main theorem below.

**Theorem 4.1.** *Algorithm 4 with a suitable choice of $\eta$ ensures that with probability at least $1 - \delta$,* $\text{Reg} = \tilde{\mathcal{O}}\left(|X|\sqrt{J|A|L^\star \ln \frac{1}{\delta}} + |X|^5|A|^2 \ln^2 \frac{1}{\delta}\right)$, *where* $L^\star = \min_\pi \sum_{t=1}^{T} \ell_t(\pi) \leq JT$ *is the total loss of the best policy.*

We remark that our bound holds for both oblivious and adaptive adversaries, and is the first high-probability small-loss bounds for adversarial MDPs.[4] This matches the bound of [25] in the worst case (including the lower-order term $\tilde{\mathcal{O}}(|X|^5|A|^2)$ hidden in their proof), but could be much smaller as long as a good policy exists with $L^\star = o(T)$. It is still open whether this bound is optimal or not.

## 5  Conclusions

In this work, based on the idea of increasing learning rates we develop a new technique for obtaining high-probability regret bounds against an adaptive adversary under bandit feedback, showing that sophisticated biased estimators used in previous approaches are not necessary. We provide three examples (MAB, linear bandits, and MDPs) to show the versatility of our general approach, leading to several new algorithms and results. Although not included in this work, we point out that our approach can also be straightforwardly applied to other problems such as semi-bandits and convex bandits, based on the algorithms from [37] and [36] respectively, since they are also based on log-barrier OMD or SCRIBLE.

## Broader Impact

This work is mostly theoretical, and we do not foresee any negative ethical or societal outcomes. Researchers working on theoretical aspects of online learning, bandit problems, and Markov Decision Processes may benefit from our results and find our techniques useful for other problems. In the long run, our results might lead to more stable and practical learning algorithms for applications with partial information feedback such as network routing or recommendation systems.

## Acknowledgments and Disclosure of Funding

HL thanks Ashok Cutkosky and Dirk van der Hoeven for many helpful discussions on normal barriers and the lifting idea. We are grateful for the support of NSF Awards IIS-1755781 and IIS-1943607, and a Google Faculty Research Award.

## Footnotes

[1]In fact, $s_t$ can be sampled from any orthonormal basis of $\mathbb{R}^d$ together with their negation. For example, in the original SCRIBLE, the eigenbasis of $H_t$ is used as this orthonormal basis. The version of sampling from a unit sphere first appears in [36], which works more generally for convex bandits.

[2]Our algorithm works with any normal barrier, not just this particular one. We use this particular form to showcase that we only require a self-concordant barrier of the original set $\Omega$, exactly the same as SCRIBLE.

[3]One caveat is that this requires measuring the learner's loss in terms of $\langle w_t, \ell_t \rangle$, as opposed to $\langle \widetilde{w}_t, \ell_t \rangle$, since the deviation between these two is not related to $m_t$.

[4]Obtaining other data-dependent regret bounds as in MAB and linear bandits is challenging in this case, since there are several terms in the regret bound that are naturally only related to $L^\star$.

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
