[Supplementary Material]

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

# A  Omitted details for Section 2

## A.1  Proof of Theorem 2.2

First we generalize the proof of the standard Freedman's inequality in the following way. For any $\lambda_t$ that is $\mathcal{F}_t$-measurable and such that $\lambda_t \leq 1/B_t$, we have with $\mathbb{E}_t[\cdot] \triangleq \mathbb{E}[\cdot|\mathcal{F}_t]$:

$$\mathbb{E}_t\left[e^{\lambda_t X_t}\right] \leq \mathbb{E}_t\left[1 + \lambda_t X_t + \lambda_t^2 X_t^2\right] = 1 + \lambda_t^2 \mathbb{E}_t\left[X_t^2\right] \leq \exp\left(\lambda_t^2 \mathbb{E}_t\left[X_t^2\right]\right). \tag{1}$$

Now for any $t$ define random variable $Z_t$ such that $Z_0 = 1$ and

$$Z_t \triangleq Z_{t-1} \cdot \exp(\lambda_t X_t - \lambda_t^2 \mathbb{E}_t\left[X_t^2\right]) = \exp\left(\sum_{s=1}^{t} \lambda_s X_s - \sum_{s=1}^{t} \lambda_s^2 \mathbb{E}_s\left[X_s^2\right]\right).$$

From Eq. (1), we have

$$\mathbb{E}_t\left[Z_t\right] = Z_{t-1} \cdot \exp(-\lambda_t^2 \mathbb{E}_t\left[X_t^2\right]) \mathbb{E}_t\left[e^{\lambda_t X_t}\right] \leq Z_{t-1} \cdot \exp(-\lambda_t^2 \mathbb{E}_t\left[X_t^2\right]) \exp(\lambda_t^2 \mathbb{E}_t\left[X_t^2\right]) \leq Z_{t-1}.$$

Therefore, taking the overall expectation we have

$$\mathbb{E}\left[Z_T\right] \leq \mathbb{E}\left[Z_{T-1}\right] \leq \cdots \leq \mathbb{E}\left[Z_0\right] = 1.$$

Using Markov's inequality, we have $\Pr\left[Z_T \geq \frac{1}{\delta'}\right] \leq \delta'$. In other words, we have with probability at least $1 - \delta'$,

$$\sum_{t=1}^{T} \lambda_t X_t \leq \ln(1/\delta') + \sum_{t=1}^{T} \lambda_t^2 \mathbb{E}_t\left[X_t^2\right]. \tag{2}$$

The proof of the standard Freedman's inequality takes all $\lambda_t$ to be the same fixed value, while in our case it is important to apply Eq. (2) several times with different sets of values of $\lambda_t$. Specifically, for each $i \in [\lceil \log_2(b^2 T) \rceil]$ and $j \in [\lceil \log_2 b \rceil]$, set

$$\lambda_t = \lambda \triangleq \min\left\{2^{-j}, \sqrt{\ln(1/\delta')/2^i}\right\},$$

for $t \in \mathcal{T}_j$, where

$$\mathcal{T}_j \triangleq \left\{t : 2^{j-1} \leq \max_{s \leq t} B_s \leq 2^j\right\},$$

and $\lambda_t = 0$ otherwise. Clearly $\lambda_t$ is $\mathcal{F}_t$-measurable (since $B_1, \ldots, B_t$ are $\mathcal{F}_t$-measurable). Applying Eq. (2) gives

$$\sum_{t \in \mathcal{T}_j} X_t \leq \frac{\ln(1/\delta')}{\lambda} + \sum_{t \in \mathcal{T}_j} \lambda \mathbb{E}_t\left[X_t^2\right]$$

$$\leq 2^j \ln(1/\delta') + \sqrt{2^i \ln(1/\delta')} + \lambda \sum_{t=1}^{T} \mathbb{E}_t\left[X_t^2\right] \qquad (\tfrac{1}{\lambda} \leq \max\{2^j, \sqrt{2^i/\ln(1/\delta')}\})$$

$$\leq 2\left(\max_{s \in \mathcal{T}_j} B_s\right) \ln(1/\delta') + \sqrt{2^i \ln(1/\delta')} + V\sqrt{\frac{\ln(1/\delta')}{2^i}} \qquad (2^{j-1} \leq \max_{s \in \mathcal{T}_j} B_s)$$

$$\leq 2B^\star \ln(1/\delta') + \sqrt{2^i \ln(1/\delta')} + V\sqrt{\frac{\ln(1/\delta')}{2^i}}.$$

By a union bound, the above holds with probability at least $1 - C\delta'$ for all $i \in [\lceil \log_2(b^2 T) \rceil]$ and $j \in [\lceil \log_2 b \rceil]$. In particular, since $1 \leq V \leq b^2 T$ (almost surely), there exists an $i^\star \in [\lceil \log_2(b^2 T) \rceil]$ such that $2^{i^\star - 1} \leq V \leq 2^{i^\star}$, and thus

$$\sum_{t=1}^{T} X_t = \sum_{j \in [\lceil \log_2 b \rceil]} \sum_{t \in \mathcal{T}_j} X_t \leq C \cdot \left(2B^\star \ln(1/\delta') + \sqrt{2^{i^\star} \ln(1/\delta')} + V\sqrt{\frac{\ln(1/\delta')}{2^{i^\star}}}\right)$$

$$\leq C \cdot \left(2B^\star \ln(1/\delta') + \sqrt{2V \ln(1/\delta')} + \sqrt{V \ln(1/\delta')}\right)$$

$$\leq C \cdot \left(2B^\star \ln(1/\delta') + \sqrt{8V \ln(1/\delta')}\right).$$

Finally replacing $\delta'$ with $\delta/C$ finishes the proof.

## A.2 Proof of Lemma 2.1

First note that $\ell_{t,i_t} = \left\langle w_t, \widehat{\ell}_t \right\rangle$. Using standard OMD analysis (e.g., [5, Lemma 12]), we have

$$\ell_{t,i_t} - \left\langle u, \widehat{\ell}_t \right\rangle \leq D_{\psi_t}(u, w_t) - D_{\psi_t}(u, w_{t+1}) + \sum_{i=1}^{d} \eta_{t,i} w_{t,i}^2 \widehat{\ell}_{t,i}^2. \tag{3}$$

Summing the first two terms on the right hand side over $t$ shows (here $h(y) = y - 1 - \ln y$):

$$\sum_{t=1}^{T} \left( D_{\psi_t}(u, w_t) - D_{\psi_t}(u, w_{t+1}) \right)$$

$$\leq D_{\psi_1}(u, w_1) + \sum_{t=1}^{T-1} \left( D_{\psi_{t+1}}(u, w_{t+1}) - D_{\psi_t}(u, w_{t+1}) \right) \qquad (D_{\psi_T}(u, w_{T+1}) \geq 0)$$

$$= \frac{1}{\eta} \sum_{j=1}^{d} h\left( \frac{u_j}{w_{1,j}} \right) + \sum_{j=1}^{d} \sum_{t=1}^{T-1} \left( \frac{1}{\eta_{t+1,j}} - \frac{1}{\eta_{t,j}} \right) h\left( \frac{u_j}{w_{t+1,j}} \right). \tag{4}$$

For the first term, since $u_j \geq \frac{1}{T}$ and $w_{1,j} = \frac{1}{d}$ for each $j$, we have

$$\frac{1}{\eta} \sum_{j=1}^{d} h\left( \frac{u_j}{w_{1,j}} \right) = \frac{1}{\eta} \sum_{j=1}^{d} -\ln(du_j) \leq \frac{d \ln T}{\eta}.$$

Now we analyze the second term for each $j$. Note that $\eta_{T,j} = \kappa^{n_j} \eta_{1,j}$ where $n_j$ is the number of times Algorithm 1 increases the learning rate for arm $j$. Let $t_j$ be the time step such that $\eta_{T,j} = \eta_{t_j+1,j} = \kappa \eta_{t_j,j}$, that is, the last time step where the learning rate for arm $j$ is increased. Then we have

$$\left( \frac{1}{\eta_{t_j+1,j}} - \frac{1}{\eta_{t_j,j}} \right) h\left( \frac{u_j}{w_{t_j+1,j}} \right) = \frac{1-\kappa}{\kappa^{n_j} \eta} h\left( \frac{u_j}{w_{t_j+1,j}} \right) \leq \frac{-h\left( \frac{u_j}{w_{t_j+1,j}} \right)}{5\eta \ln T} = \frac{-h\left( \frac{u_j \rho_{T,j}}{2} \right)}{5\eta \ln T},$$

where we use the facts $1 - \kappa \leq -\frac{1}{\ln T}$ and $\kappa^{n_j} \leq 5$. The term $-h\left( \frac{u_j \rho_{T,j}}{2} \right)$ is bounded by

$$-h\left( \frac{u_j \rho_{T,j}}{2} \right) = \ln\left( \frac{u_j \rho_{T,j}}{2} \right) - \frac{u_j \rho_{T,j}}{2} + 1 \leq 1 + \ln T - \frac{u_j \rho_{T,j}}{2},$$

where the inequality is because $\frac{u_j \rho_{T,j}}{2} \leq \frac{1}{w_{t_j+1,j}} \leq T$. Plugging this result for every $j$ back to Eq. (4), we get

$$\sum_{t=1}^{T} D_{\psi_t}(u, w_t) - D_{\psi_t}(u, w_{t+1}) \leq \frac{d \ln T}{\eta} + \sum_{j=1}^{d} \frac{2 + 2\ln T - u_j \rho_{T,j}}{10\eta \ln T} = \mathcal{O}\left( \frac{d \ln T}{\eta} \right) - \frac{\langle \rho_T, u \rangle}{10\eta \ln T}.$$

Finally, since $\eta_{t,i} w_{t,i}^2 \widehat{\ell}_{t,i}^2 \leq \eta_{t,i_t} \ell_{t,i_t} \leq \eta_{T,i_t} \ell_{t,i_t} \leq 5\eta \ell_{t,i_t}$, summing Eq. (3) over $t$ gives:

$$\sum_{t=1}^{T} \left( \ell_{t,i_t} - \left\langle u, \widehat{\ell}_t \right\rangle \right) \leq \sum_{t=1}^{T} \left( D_{\psi_t}(u, w_t) - D_{\psi_t}(u, w_{t+1}) \right) + \sum_{t=1}^{T} \sum_{i=1}^{d} \eta_{t,i} w_{t,i}^2 \widehat{\ell}_{t,i}^2$$

$$\leq \mathcal{O}\left( \frac{d \ln T}{\eta} \right) - \frac{\langle \rho_T, u \rangle}{10\eta \ln T} + 5\eta \sum_{t=1}^{T} \ell_{t,i_t}$$

$$= \mathcal{O}\left( \frac{d \ln T}{\eta} + \eta \sum_{t=1}^{T} \ell_{t,i_t} \right) - \frac{\langle \rho_T, u \rangle}{10\eta \ln T}.$$

## A.3 Proof of Theorem 2.3

Fix any $i^\star \in [d]$ and let $u = (1 - \frac{d}{T})e^\star + \frac{1}{T}\mathbf{1}$, where $e^\star$ is the one-hot vector for $i^\star$. First note that

$$\sum_{t=1}^{T}(\ell_{t,i_t} - \ell_{t,i^\star}) = \sum_{t=1}^{T}\left(\ell_{t,i_t} - \left\langle u, \hat{\ell}_t \right\rangle\right) + \sum_{t=1}^{T}\left\langle u, \widehat{\ell}_t - \ell_t \right\rangle + \sum_{t=1}^{T}\langle u - e^\star, \ell_t \rangle$$

$$\leq \sum_{t=1}^{T}\left(\ell_{t,i_t} - \left\langle u, \hat{\ell}_t \right\rangle\right) + \sum_{t=1}^{T}\left\langle u, \widehat{\ell}_t - \ell_t \right\rangle + d.$$

For the first term, using Lemma 2.1, we have

$$\sum_{t=1}^{T}\left(\ell_{t,i_t} - \left\langle u, \hat{\ell}_t \right\rangle\right) \leq \mathcal{O}\left(\frac{d\ln T}{\eta} + \eta L_T\right) - \frac{\langle \rho_T, u \rangle}{10\eta \ln T}, \tag{5}$$

where $L_T = \sum_{t=1}^{T} \ell_{t,i_t}$.

For the second term above, we use Theorem 2.2 with $X_t = \left\langle u, \widehat{\ell}_t - \ell_t \right\rangle$, $B_t = \langle \rho_t, u \rangle \in [1, T]$, $b = T$, and the fact

$$\mathbb{E}_t[X_t^2] \leq \mathbb{E}_t\left[\left\langle u, \widehat{\ell}_t \right\rangle^2\right] = \mathbb{E}_t\left[\frac{u_{i_t}^2 \ell_{t,i_t}^2}{p_{t,i_t}^2}\right] \leq \sum_{i=1}^{d} u_i^2 \ell_{t,i} \rho_{T,i} \leq \langle \rho_T, u \rangle \langle u, \ell_t \rangle,$$

showing that with probability at least $1 - \delta'$,

$$\sum_{t=1}^{T}\left\langle u, \widehat{\ell}_t - \ell_t \right\rangle \leq C\left(\sqrt{8 L_u \langle \rho_T, u \rangle \ln(C/\delta')} + 2\langle \rho_T, u \rangle \ln(C/\delta')\right), \tag{6}$$

where $L_u = \left\langle u, \sum_{t=1}^{T} \ell_t \right\rangle$ and $C = \lceil \log(b) \rceil \lceil \log(b^2 T) \rceil = \lceil \log(T) \rceil \lceil 3\log(T) \rceil$. With $\eta \leq \frac{1}{40 C \ln T \ln(C/\delta')}$, we then have with probability at least $1 - \delta'$,

$$\sum_{t=1}^{T} \ell_{t,i_t} - \ell_{t,i^\star}$$

$$\leq \tilde{\mathcal{O}}\left(\frac{d}{\eta} + \eta L_T\right) - \frac{\langle \rho_T, u \rangle}{10\eta \ln T} + C\left(\sqrt{8 L_u \langle \rho_T, u \rangle \ln(C/\delta')} + 2\langle \rho_T, u \rangle \ln(C/\delta')\right)$$
$$\text{(Eq. (5) and Eq. (6))}$$

$$\leq \tilde{\mathcal{O}}\left(\frac{d}{\eta} + \eta L_T\right) + \eta 40 C^2 L_u \ln(C/\delta')\ln T - \frac{\langle \rho_T, u \rangle}{20\eta \ln T} + 2C\langle \rho_T, u \rangle \ln(C/\delta')$$
$$\text{(AM-GM inequality)}$$

$$\leq \tilde{\mathcal{O}}\left(\frac{d}{\eta} + \eta L_T \ln(1/\delta') + \eta L_u \ln(1/\delta')\right). \qquad (\eta < \tfrac{1}{40 C \ln T \ln(C/\delta)})$$

Therefore, rearranging the terms, using the fact $L_u \leq L^\star + d$, and choosing $\eta = \min\left\{\sqrt{\frac{d}{L^\star}\ln(1/\delta')}, \frac{1}{40 C \ln T \ln(C/\delta')}, \frac{1}{2}\right\}$, we have with probability $1 - \delta'$,

$$\sum_{t=1}^{T} \ell_{t,i_t} - \ell_{t,i^\star} = \tilde{\mathcal{O}}\left(\sqrt{dL^\star \ln(1/\delta')} + d\ln(1/\delta')\right),$$

where $L^\star = \sum_{t=1}^{T} \ell_{t,i^\star}$. This finishes the proof when the adversary is oblivious. For adaptive adversaries, taking a union bound over all possible best arms $i^\star \in [d]$ and setting $\delta' = \delta/d$, we have with probability $1 - \delta$, $\text{Reg} = \widetilde{\mathcal{O}}\left(\sqrt{dL^\star \ln(d/\delta)} + d\ln(d/\delta)\right)$, finishing the proof.

**Remark 1.** Although the proof above requires tuning the initial learning rate $\eta$ in terms of the unknown quantity $L^\star$, standard doubling trick can remove this restriction (even in the bandit setting). We refer the reader to a recent work by Lee et al. [27] for detailed exposition on how to achieve so.

---

**Algorithm 3** $d$-dimensional version of Algorithm 2

---

**Input:** decision set $\Omega \subseteq \mathbb{R}^d$, a $\nu$-self-concordant barrier $\psi(w)$ for $\Omega$, initial learning rate $\eta$.

**Define:** increase factor $\kappa = e^{\frac{1}{100d\ln(\nu T)}}$, $\Psi(w, b) = 400\left(\psi\left(\frac{w}{b}\right) - 2\nu \ln b\right)$.

**Initialize:** $w_1 = \arg\min_{w \in \Omega} \psi(w)$, $\boldsymbol{H}_1 = \nabla^2 \Psi(w_1, 1)$, $\eta_1 = \eta$, $\mathcal{S} = \{1\}$.

**Define:** shrunk decision set $\Omega' = \{w \in \Omega : \pi_{w_1}(w) \leq 1 - \frac{1}{T}\}$, $J = [I_d, \mathbf{0}_d] \in \mathbb{R}^{d \times (d+1)}$.

**for** $t = 1, 2, \ldots, T$ **do**

1    Uniformly at random sample $\boldsymbol{s}_t$ from $\left(\boldsymbol{H}_t^{-\frac{1}{2}} e_{d+1}\right)^{\perp} \cap \mathbb{S}^{d+1}$.

2    Compute $\widetilde{w}_t = w_t + J\boldsymbol{H}_t^{-\frac{1}{2}}\boldsymbol{s}_t$.

3    Play $\widetilde{w}_t$, observe loss $\langle \widetilde{w}_t, \ell_t \rangle$, and construct loss estimator $\widehat{\ell}_t = d\langle \widetilde{w}_t, \ell_t \rangle J\boldsymbol{H}_t^{1/2}\boldsymbol{s}_t$.

4    Compute $w_{t+1} = \arg\min_{w \in \Omega'} \left\{ \left\langle w, \widehat{\ell}_t \right\rangle + D_{\psi_t}(w, w_t) \right\}$, where $\psi_t = \frac{1}{\eta_t}\psi$.

5    Compute $\boldsymbol{H}_{t+1} = \nabla^2 \Psi(w_{t+1}, 1)$.

6    **if** $\lambda_{\max}(\boldsymbol{H}_{t+1} - \sum_{\tau \in \mathcal{S}} \boldsymbol{H}_\tau) > 0$ **then** $\mathcal{S} \leftarrow \mathcal{S} \cup \{t+1\}$ and set $\eta_{t+1} = \eta_t \kappa$;

7    **else** set $\eta_{t+1} = \eta_t$.

---

# B  Omitted details for Section 3

## B.1  More explanation on Algorithm 2

Here, we provide a $d$-dimensional version of Algorithm 2 by removing the explicit lifting and performing OMD in $\mathbb{R}^d$; see Algorithm 3. It is clear that this version is exactly the same as Algorithm 2. Compared to the original SCRIBLE, one can see that besides the increasing learning rate schedule, the only difference is how the point $\widetilde{w}_t$ is computed. In particular, one can verify that $\widetilde{w}_t$ does not necessarily satisfy $\|\widetilde{w}_t - w_t\|_{\nabla^2\psi(w_t)} = 1$, meaning that $\widetilde{w}_t$ is not necessarily on the boundary of the Dikin ellipsoid centered at $w_t$ with respect to $\psi$. In other words, our algorithm provides a new sampling scheme for SCRIBLE.

## B.2  Preliminary for analysis

In this section, we introduce the preliminary of self-concordant barriers and normal-barriers, including the definitions and some useful properties that will be used frequently in later analysis.

**Self-concordant barriers.** Let $\psi : \text{int}(\Omega) \to \mathbb{R}$ be a $C^3$ smooth convex function. $\psi$ is called a self-concordant barrier on $\Omega$ if it satisfies:

- $\psi(x_i) \to \infty$ as $i \to \infty$ for any sequence $x_1, x_2, \cdots \in \text{int}(\Omega) \subset \mathbb{R}^d$ converging to the boundary of $\Omega$;

- for all $w \in \text{int}(\Omega)$ and $h \in \mathbb{R}^d$, the following inequality always holds:

$$\sum_{i=1}^d \sum_{j=1}^d \sum_{k=1}^d \frac{\partial^3 \psi(w)}{\partial w_i \partial w_j \partial w_k} h_i h_j h_k \leq 2\|h\|_{\nabla^2\psi(w)}^3.$$

We further call $\psi$ is a $\nu$-self-concordant barrier if it satisfies the conditions above and also

$$\langle \nabla\psi(w), h \rangle \leq \sqrt{\nu}\|h\|_{\nabla^2\psi(w)}$$

for all $w \in \text{int}(\Omega)$ and $h \in \mathbb{R}^d$.

**Lemma B.1** (Theorem 2.1.1 in [30]). *If $\psi$ is a self-concordant barrier on $\Omega$, then the Dikin ellipsoid centered at $w \in \text{int}(\Omega)$, defined as $\{v : \|v - w\|_{\nabla^2\psi(w)} \leq 1\}$, is always within $\Omega$. Moreover,*

$$\|h\|_{\nabla^2\psi(v)} \geq \|h\|_{\nabla^2\psi(w)}\left(1 - \|v - w\|_{\nabla^2\psi(w)}\right)$$

*holds for any $h \in \mathbb{R}^d$ and any $v$ with $\|v - w\|_{\nabla^2\psi(w)} \leq 1$.*

**Lemma B.2** (Theorem 2.5.1 in [30]). *For any closed convex body $\Omega \subset \mathbb{R}^d$, there exists an $\mathcal{O}(d)$-self-concordant barrier on $\Omega$.*

**Lemma B.3** (Corollary 2.3.1 in [30]). *Let $\psi$ be a self-concordant barrier for $\Omega \subset \mathbb{R}^d$. Then for any $w \in \text{int}(\Omega)$ and any $h \in \Omega$ such that $w + bh \in \Omega$ for all $b \geq 0$, we have*

$$\|h\|_{\nabla^2\psi(w)} \leq - \langle \nabla\psi(w), h \rangle.$$

**Normal Barriers.** Let $K \subseteq \mathbb{R}^d$ be a closed and proper convex cone and let $\theta \geq 1$. A function $\psi : \text{int}(K) \to \mathbb{R}$ is called a $\theta$-logarithmically homogeneous self-concordant barrier (or simply a $\theta$-normal barrier) on $K$ if it is self-concordant on $\text{int}(K)$ and is logarithmically homogeneous with parameter $\theta$, which means

$$\psi(tw) = \psi(w) - \theta \ln t, \ \forall w \in \text{int}(K), \ t > 0.$$

The following two lemmas show the relationship between $\theta$-normal barriers and $\theta$-self-concordant barriers.

**Lemma B.4** (Corollary 2.3.2 in [30]). *A $\theta$-normal barrier on $K$ is a $\theta$-self-concordant barrier on $K$.*

**Lemma B.5** (Proposition 5.1.4 in [30]). *Suppose $f$ is a $\theta$-self-concordant barrier on $K \subseteq \mathbb{R}^d$. Then the function*

$$F(w, b) = 400 \left( f\left(\frac{w}{b}\right) - 2\theta \ln b \right),$$

*is a $800\theta$-normal barrier for $con(K) \subseteq \mathbb{R}^{d+1}$, where $con(K) = \{\mathbf{0}\} \cup \{(w, b) : \frac{w}{b} \in K, w \in \mathbb{R}^d, b > 0\}$ is the conic hull of $K$ lifted to $\mathbb{R}^{d+1}$ (by appending 1 to the last coordinate).*

Note that our regularizer $\Psi$ defined in Algorithm 2 is exactly based on this formula. We point out that, however, our entire analysis works for any $\mathcal{O}(\nu)$-normal barrier $\Psi$, as we will only use the following general properties of normal barriers, instead of the concrete form of $\Psi$. As mentioned in Footnote 2, we use this concrete formula only to emphasize that, just as SCRIBLE, our algorithm requires only a self-concordant barrier of the original set $\Omega$.

**Lemma B.6** (Proposition 2.3.4 in [30]). *If $\psi$ is a $\theta$-normal barrier on $K$, then we have for all $w, u \in \text{int}(K)$,*

1. *$\|w\|^2_{\nabla^2\psi(w)} = w^\top \nabla^2\psi(w)w = \theta$,*

2. *$\nabla^2\psi(w)w = -\nabla\psi(w)$,*

3. *$\psi(u) \geq \psi(w) - \theta \ln \frac{-\langle \nabla\psi(w), u \rangle}{\theta}$.*

Next, we show the definition of Minkowsky functions, which is used to define the shrunk decision domain similar to the clipped simplex in multi-armed bandit setting.

**Minkowsky functions.** The Minkowsky function of a convex body $\Omega$ with the pole at $w \in \text{int}(\Omega)$ is a function $\pi_w : \Omega \to \mathbb{R}$ defined as

$$\pi_w(u) = \inf \left\{ t > 0 \,\middle|\, w + \frac{u - w}{t} \in \Omega \right\}.$$

The last lemma shows several useful properties using the Minkowsky function.

**Lemma B.7** (Proposition 2.3.2 in [30]). *Let $\psi$ be a $\nu$-self-concordant barrier on $\Omega \subseteq \mathbb{R}^d$ and $u, w \in \text{int}(\Omega)$. Then for any $h \in \mathbb{R}^d$, we have*

$$\|h\|_{\nabla^2\psi(u)} \leq \left( \frac{1 + 3\nu}{1 - \pi_w(u)} \right) \|h\|_{\nabla^2\psi(w)},$$

$$|\langle \nabla\psi(u), h \rangle| \leq \left( \frac{\nu}{1 - \pi_w(u)} \right) \|h\|_{\nabla^2\psi(w)},$$

$$\psi(u) - \psi(w) \leq \nu \ln \left( \frac{1}{1 - \pi_w(u)} \right).$$

## B.3 Proof of Theorem 3.1

To prove the theorem, we decompose the regret against any fixed $u^\star \in \Omega$ (with $\boldsymbol{u}^\star = (u^\star, 1) \in \boldsymbol{\Omega}$) into the following three terms:

$$\sum_{t=1}^{T} \langle \widetilde{w}_t - u^\star, \ell_t \rangle$$

$$= \sum_{t=1}^{T} \langle \widetilde{\boldsymbol{w}}_t - \boldsymbol{u}^\star, \boldsymbol{\ell}_t \rangle \qquad \qquad \qquad \text{(define } \boldsymbol{\ell}_t = (\ell_t, 0))$$

$$= \underbrace{\sum_{t=1}^{T} \left( \langle \widetilde{\boldsymbol{w}}_t, \boldsymbol{\ell}_t \rangle - \langle \boldsymbol{w}_t, \widehat{\boldsymbol{\ell}}_t \rangle + \langle \boldsymbol{u}, \widehat{\boldsymbol{\ell}}_t - \boldsymbol{\ell}_t \rangle \right)}_{\text{DEVIATION}} + \underbrace{\sum_{t=1}^{T} \langle \boldsymbol{w}_t - \boldsymbol{u}, \widehat{\boldsymbol{\ell}}_t \rangle}_{\text{REG-TERM}} + \sum_{t=1}^{T} \langle \boldsymbol{u} - \boldsymbol{u}^\star, \boldsymbol{\ell}_t \rangle, \qquad (7)$$

where $\boldsymbol{u} = \left(1 - \frac{1}{T}\right) \cdot \boldsymbol{u}^\star + \frac{1}{T} \cdot \boldsymbol{w}_1 \in \boldsymbol{\Omega}'$. Note that the last term is trivially bounded by 2 as $\sum_{t=1}^{T} \langle \boldsymbol{u} - \boldsymbol{u}^\star, \boldsymbol{\ell}_t \rangle = \sum_{t=1}^{T} \langle u - u^\star, \ell_t \rangle = \frac{1}{T} \sum_{t=1}^{T} \langle u^\star - w_1, \ell_t \rangle \leq 2$, where the last inequality is because $|\langle w, \ell_t \rangle| \leq 1$ for all $w \in \Omega$. In the following sections, we show how to bound other terms. Specifically, we bound DEVIATION in Section B.3.1 and REG-TERM in Section B.3.2. Finally we prove Theorem 3.1 in Section B.3.3.

We will use the following notations in the remaining of this section (the first two are mentioned above already):

$$\boldsymbol{\ell}_t \triangleq (\ell_t, 0), \quad \boldsymbol{u} \triangleq (u, 1) \triangleq \left(1 - \frac{1}{T}\right) \cdot \boldsymbol{u}^\star + \frac{1}{T} \cdot \boldsymbol{w}_1 \in \boldsymbol{\Omega}', \quad \rho \triangleq \max_{t \in [T]} \|\boldsymbol{u}\|_{\boldsymbol{H}_t}, \qquad (8)$$

$$L_T \triangleq \sum_{t=1}^{T} \langle \widetilde{\boldsymbol{w}}_t, \boldsymbol{\ell}_t \rangle, \quad \overline{L}_T \triangleq \sum_{t=1}^{T} |\langle \widetilde{\boldsymbol{w}}_t, \boldsymbol{\ell}_t \rangle|, \qquad (9)$$

$$\mathring{L}_T \triangleq \sum_{t=1}^{T} \mathbb{E}_t \left[ |\langle \widetilde{\boldsymbol{w}}_t, \boldsymbol{\ell}_t \rangle| \right], \quad \overline{L}_u \triangleq \sum_{t=1}^{T} |\langle u, \ell_t \rangle|. \qquad (10)$$

Before proceeding, we provide one useful lemma.

**Lemma B.8.** *We have* $\|\boldsymbol{u}\|_{\boldsymbol{H}_1} \leq 800\nu$.

*Proof.* Clearly, for any $b > 0$, we have $\boldsymbol{w}_1 + b\boldsymbol{u}$ still in the conic hull of $\boldsymbol{\Omega}$. According to Lemma B.3, we thus have $\|\boldsymbol{u}\|_{\boldsymbol{H}_1} \leq \langle -\nabla \Psi(\boldsymbol{w}_1), \boldsymbol{u} \rangle$. Note that $\Psi$ is a $800\nu$-normal barrier by Lemma B.5. By the first order optimality condition of $\boldsymbol{w}_1$ and Lemma B.6, we then have

$$0 \leq \langle \nabla \Psi(\boldsymbol{w}_1), \boldsymbol{u} - \boldsymbol{w}_1 \rangle = \langle \nabla \Psi(\boldsymbol{w}_1), \boldsymbol{u} \rangle + 800\nu.$$

Combining the above gives $\|\boldsymbol{u}\|_{\boldsymbol{H}_1} \leq \langle -\nabla \Psi(\boldsymbol{w}_1), \boldsymbol{u} \rangle \leq 800\nu$. $\qquad \square$

### B.3.1 Bounding DEVIATION

We first show that $\widehat{\boldsymbol{\ell}}_t$ is an unbiased estimator of $\boldsymbol{\ell}_t$ for the first $d$ coordinates.

**Lemma B.9.** *We have* $\mathbb{E}_t \left[ \widehat{\ell}_{t,i} \right] = \ell_{t,i}$ *for* $i \in [d]$.

*Proof.* Let $\boldsymbol{v} = \frac{\boldsymbol{H}_t^{-1/2} e_{d+1}}{\left\| \boldsymbol{H}_t^{-1/2} e_{d+1} \right\|_2}$. First note that

$$\mathbb{E}_t[\boldsymbol{s}_t \boldsymbol{s}_t^\top] = \frac{1}{d} \left( \boldsymbol{I} - \boldsymbol{v}\boldsymbol{v}^\top \right) \qquad (11)$$

by the definition of $s_t$. Then by the definition of $\widehat{\ell}_t$, we have

$$
\begin{aligned}
\mathbb{E}_t\left[\widehat{\ell}_t\right] &= \mathbb{E}_t\left[d\left\langle w_t + H_t^{-\frac{1}{2}}s_t, \ell_t\right\rangle \cdot H_t^{\frac{1}{2}}s_t\right] \\
&= \mathbb{E}_t\left[d\left\langle w_t, \ell_t\right\rangle \cdot H_t^{\frac{1}{2}}s_t + d \cdot H_t^{\frac{1}{2}}s_t\left\langle H_t^{-\frac{1}{2}}s_t, \ell_t\right\rangle\right] \\
&= d\left\langle w_t, \ell_t\right\rangle \cdot H_t^{\frac{1}{2}}\mathbb{E}_t\left[s_t\right] + \mathbb{E}_t\left[d \cdot H_t^{\frac{1}{2}}s_t s_t^\top H_t^{-\frac{1}{2}}\ell_t\right] \\
&= d \cdot H_t^{\frac{1}{2}}\mathbb{E}_t\left[s_t s_t^\top\right] H_t^{-\frac{1}{2}}\ell_t && (\mathbb{E}_t\left[s_t\right] = \mathbf{0} \text{ by symmetry}) \\
&= H_t^{\frac{1}{2}}\left(I - vv^\top\right) H_t^{-\frac{1}{2}}\ell_t && (\text{Eq. (11)}) \\
&= \ell_t - \frac{e_{d+1}e_{d+1}^\top H_t^{-1}\ell_t}{\left\|H_t^{-1/2}e_{d+1}\right\|_2^2}.
\end{aligned}
$$

Noticing that the first $d$ coordinates of $e_{d+1}e_{d+1}^\top H_t^{-1}\ell_t$ are all zeros concludes the proof. $\qquad\square$

Now we are ready to bound DEVIATION.

**Lemma B.10.** *With probability at least $1 - \delta$, we have*

$$
\text{DEVIATION} \leq 161Cd\sqrt{(\nu + \rho^2)\mathring{L}_T \ln(C/\delta)} + C\sqrt{32\overline{L}_u \ln(C/\delta)} + 64Cd\left(\sqrt{\nu} + \rho\right)\ln(C/\delta),
$$

*where $C = \Theta(\ln^2(d\nu T))$.*

*Proof.* Define $X_t \triangleq \langle \widetilde{w}_t, \ell_t\rangle - \left\langle w_t, \widehat{\ell}_t\right\rangle + \left\langle u, \widehat{\ell}_t - \ell_t\right\rangle$ and we have DEVIATION $= \sum_{t=1}^T X_t$. The goal is to apply our strengthened Freedman's inequality Theorem 2.2. To this end, first we show $\mathbb{E}_t[X_t] = 0$. Indeed, we have $\mathbb{E}_t[\widetilde{w}_t] = w_t$ and

$$
\begin{aligned}
\mathbb{E}_t[X_t] &= \langle w_t, \ell_t\rangle - \langle w_t, \ell_t\rangle + \langle u, \ell_t - \ell_t\rangle - \mathbb{E}_t\left[(w_{t,d+1} - u_{t,d+1})\widehat{\ell}_{t,d+1}\right] && (\text{Lemma B.9}) \\
&= 0. && (w_{t,d+1} = u_{t,d+1} = 1)
\end{aligned}
$$

Next, we bound $X_t$ by a $\mathcal{F}_t$-measurable random variable $B_t \triangleq 32d\sqrt{\nu} + d\|u\|_{H_t}$. This can be shown using the properties of a normal barrier:

$$
\begin{aligned}
X_t &= \langle \widetilde{w}_t, \ell_t\rangle - \left\langle w_t, \widehat{\ell}_t\right\rangle + \left\langle u, \widehat{\ell}_t - \ell_t\right\rangle \\
&= \langle \widetilde{w}_t, \ell_t\rangle - \left\langle w_t, d \cdot \langle \widetilde{w}_t, \ell_t\rangle \cdot H_t^{\frac{1}{2}}s_t\right\rangle + \left\langle u, d \cdot \langle \widetilde{w}_t, \ell_t\rangle H_t^{\frac{1}{2}}s_t - \ell_t\right\rangle \\
&= \langle \widetilde{w}_t, \ell_t\rangle\left(1 - dw_t^\top H_t^{\frac{1}{2}}s_t\right) + d\langle \widetilde{w}_t, \ell_t\rangle u^\top H_t^{\frac{1}{2}}s_t - \langle u, \ell_t\rangle \\
&\leq 2 + d\left|w_t^\top H_t^{\frac{1}{2}}s_t\right| + d\left|u^\top H_t^{\frac{1}{2}}s_t\right| && (|\langle w, \ell_t\rangle| \leq 1 \text{ for any } w \in \Omega) \\
&\leq 2 + d\|w_t\|_{H_t} + d\|u\|_{H_t} && (\text{by Cauchy-Schwarz inequality and } s_t^\top s_t = 1) \\
&\leq 2 + 20d\sqrt{2\nu} + d\|u\|_{H_t} && (\text{Lemma B.5 and Lemma B.6}) \\
&\leq 32d\sqrt{\nu} + d\|u\|_{H_t}. && (\nu \geq 1)
\end{aligned}
$$

Then, we show that $B_t$ is bounded by a constant $b \triangleq 2 \times 10^6 d\nu^2 T$ for all $t$:

$$
\begin{aligned}
B_t &\leq 32d\sqrt{\nu} + d\|u\|_{H_1} \cdot \left(\frac{1 + 2400\nu}{1 - \pi_{w_1}(w_t)}\right) && (\text{Lemma B.7}) \\
&\leq 32d\sqrt{\nu} + d\|u\|_{H_1}(1 + 2400\nu)T && (w_t \in \Omega') \\
&\leq 32d\sqrt{\nu} + 800d\nu(1 + 2400\nu)T && (\text{Lemma B.8}) \\
&\leq 2 \times 10^6 d\nu^2 T. && (\nu \geq 1)
\end{aligned}
$$

The last step before applying Theorem 2.2 is to calculate $\mathbb{E}_t[X_t^2]$. We first write

$$\mathbb{E}_t[X_t^2] = \mathbb{E}_t\left[\left(\langle\widetilde{\boldsymbol{w}}_t, \boldsymbol{\ell}_t\rangle - \left\langle\boldsymbol{w}_t, \widehat{\boldsymbol{\ell}}_t\right\rangle + \left\langle\boldsymbol{u}, \widehat{\boldsymbol{\ell}}_t - \boldsymbol{\ell}_t\right\rangle\right)^2\right]$$

$$\leq 2\mathbb{E}_t\left[\left(\langle\widetilde{\boldsymbol{w}}_t, \boldsymbol{\ell}_t\rangle - \left\langle\boldsymbol{w}_t, \widehat{\boldsymbol{\ell}}_t\right\rangle\right)^2\right] + 2\mathbb{E}_t\left[\left\langle\boldsymbol{u}, \widehat{\boldsymbol{\ell}}_t - \boldsymbol{\ell}_t\right\rangle^2\right]. \tag{12}$$

The first term is bounded by:

$$\mathbb{E}_t\left[\left(\langle\widetilde{\boldsymbol{w}}_t, \boldsymbol{\ell}_t\rangle - \left\langle\boldsymbol{w}_t, \widehat{\boldsymbol{\ell}}_t\right\rangle\right)^2\right]$$

$$= \mathbb{E}_t\left[\langle\widetilde{\boldsymbol{w}}_t, \boldsymbol{\ell}_t\rangle^2\left(1 - \left\langle\boldsymbol{w}_t, d\cdot\boldsymbol{H}_t^{\frac{1}{2}}\boldsymbol{s}_t\right\rangle\right)^2\right]$$

$$\leq \mathbb{E}_t\left[\langle\widetilde{\boldsymbol{w}}_t, \boldsymbol{\ell}_t\rangle^2\left(2d^2\left(\boldsymbol{w}_t^\top\boldsymbol{H}_t^{\frac{1}{2}}\boldsymbol{s}_t\right)^2 + 2\right)\right]$$

$$\leq \mathbb{E}_t\left[|\langle\widetilde{\boldsymbol{w}}_t, \boldsymbol{\ell}_t\rangle|\left(2d^2\left(\boldsymbol{w}_t^\top\boldsymbol{H}_t^{\frac{1}{2}}\boldsymbol{s}_t\right)^2 + 2\right)\right] \qquad (\langle\widetilde{\boldsymbol{w}}_t, \boldsymbol{\ell}_t\rangle \leq 1)$$

$$\leq \mathbb{E}_t\left[|\langle\widetilde{\boldsymbol{w}}_t, \boldsymbol{\ell}_t\rangle|\left(2d^2\|\boldsymbol{w}_t\|_{\boldsymbol{H}_t}^2\|\boldsymbol{s}_t\|_2^2 + 2\right)\right] \qquad \text{(Cauchy-Schwarz inequality)}$$

$$\leq \mathbb{E}_t\left[|\langle\widetilde{\boldsymbol{w}}_t, \boldsymbol{\ell}_t\rangle|\left(1600d^2\nu + 2\right)\right] \qquad (\|\boldsymbol{s}_t\|_2^2 = 1 \text{ and Lemma B.6})$$

$$\leq 1602d^2\nu\mathbb{E}_t\left[|\langle\widetilde{\boldsymbol{w}}_t, \boldsymbol{\ell}_t\rangle|\right].$$

Similarly, the second term is bounded by:

$$\mathbb{E}_t\left[\left\langle\boldsymbol{u}, \widehat{\boldsymbol{\ell}}_t - \boldsymbol{\ell}_t\right\rangle^2\right] \leq \mathbb{E}_t\left[\left(-\langle\boldsymbol{u}, \boldsymbol{\ell}_t\rangle + d\langle\widetilde{\boldsymbol{w}}_t, \boldsymbol{\ell}_t\rangle\boldsymbol{u}^\top\boldsymbol{H}_t^{\frac{1}{2}}\boldsymbol{s}_t\right)^2\right]$$

$$\leq \mathbb{E}_t\left[2|\langle\boldsymbol{u}, \boldsymbol{\ell}_t\rangle| + 2d^2|\langle\widetilde{\boldsymbol{w}}_t, \boldsymbol{\ell}_t\rangle|\cdot\left(\boldsymbol{u}^\top\boldsymbol{H}_t^{\frac{1}{2}}\boldsymbol{s}_t\right)^2\right] \qquad (\langle\widetilde{\boldsymbol{w}}_t, \boldsymbol{\ell}_t\rangle \leq 1)$$

$$\leq \mathbb{E}_t\left[2|\langle\boldsymbol{u}, \boldsymbol{\ell}_t\rangle| + 2d^2|\langle\widetilde{\boldsymbol{w}}_t, \boldsymbol{\ell}_t\rangle|\cdot\|\boldsymbol{u}\|_{\boldsymbol{H}_t}^2\right].$$

Plugging these bounds to Eq. (12), we have

$$\mathbb{E}_t[X_t^2] \leq 3204d^2\nu\mathbb{E}_t\left[|\langle\widetilde{\boldsymbol{w}}_t, \boldsymbol{\ell}_t\rangle|\right] + 4|\langle\boldsymbol{u}, \boldsymbol{\ell}_t\rangle| + 4d^2\mathbb{E}_t\left[|\langle\widetilde{\boldsymbol{w}}_t, \boldsymbol{\ell}_t\rangle|\right]\|\boldsymbol{u}\|_{\boldsymbol{H}_t}^2.$$

Summing over $t$ gives

$$\sum_{t=1}^T\mathbb{E}_t[X_t^2] \leq 3204d^2\sum_{t=1}^T\left(\nu + \|\boldsymbol{u}\|_{\boldsymbol{H}_t}^2\right)\mathbb{E}_t\left[|\langle\widetilde{\boldsymbol{w}}_t, \boldsymbol{\ell}_t\rangle|\right] + 4\sum_{t=1}^T|\langle\boldsymbol{u}, \boldsymbol{\ell}_t\rangle|$$

$$\leq 3204d^2\left(\nu + \max_{t\in[T]}\|\boldsymbol{u}\|_{\boldsymbol{H}_t}^2\right)\sum_{t=1}^T\mathbb{E}_t\left[|\langle\widetilde{\boldsymbol{w}}_t, \boldsymbol{\ell}_t\rangle|\right] + 4\sum_{t=1}^T|\langle\boldsymbol{u}, \boldsymbol{\ell}_t\rangle|$$

$$= 3204d^2\left(\nu + \rho^2\right)\mathring{L}_T + 4\overline{L}_u.$$

Therefore, choosing $B^\star = 32d(\sqrt{\nu}+\rho)$, $b = 2\times10^6 d\nu^2 T$, $C = \lceil\log_2 b\rceil\lceil\log_2 b^2 T\rceil = \Theta(\ln^2(d\nu T))$ and using Theorem 2.2, we obtain with probability $1 - \delta$,

$$\sum_{t=1}^T X_t = \sum_{t=1}^T\left(\langle\widetilde{\boldsymbol{w}}_t, \boldsymbol{\ell}_t\rangle - \left\langle\boldsymbol{w}_t, \widehat{\boldsymbol{\ell}}_t\right\rangle + \left\langle\boldsymbol{u}, \widehat{\boldsymbol{\ell}}_t - \boldsymbol{\ell}_t\right\rangle\right)$$

$$\leq C\sqrt{25632d^2\left(\nu + \rho^2\right)\mathring{L}_T\ln(C/\delta) + 32\overline{L}_u\ln(C/\delta)} + 64Cd\left(\sqrt{\nu} + \rho\right)\ln(C/\delta).$$

Finally, using $\sqrt{a + b} \leq \sqrt{a} + \sqrt{b}$, the first term above is bounded by

$$161C\sqrt{d^2\left(\nu + \rho^2\right)\mathring{L}_T\ln(C/\delta)} + C\sqrt{32\overline{L}_u\ln(C/\delta)},$$

which finishes the proof. $\qquad\square$

### B.3.2 Bounding REG-TERM

The goal of this section is to prove the following bound on REG-TERM.

**Lemma B.11.** *Let $\mathcal{S}$ be its final value after running [Algorithm 2](#) for $T$ rounds and $\mathcal{S}' = \mathcal{S} \backslash \{1, T+1\}$. Then as long as $\eta \leq \frac{1}{80d}$, we have*

$$\text{REG-TERM} \leq \tilde{\mathcal{O}}\left(\frac{\nu}{\eta}\right) - \frac{\sum_{s \in \mathcal{S}'} \|\boldsymbol{u}\|_{\boldsymbol{H}_s}}{5\eta a d \ln(\nu T)} + 40\eta d^2 \overline{L}_T.$$

*for $a = 100$.*

To prove this lemma, we first prove three useful lemmas. The first one shows that the number of times [Algorithm 2](#) increases the learning rate is upper bounded by $\mathcal{O}(d \log_2(d\nu T))$.

**Lemma B.12.** *Assume that $T \geq 8$. Let $n$ be the number of times [Algorithm 2](#) increases the learning rate. Then $n \leq ad \log_2(\nu T)$ for $a = 100$. Consequently, we have $\eta_t \leq 5\eta$ for all $t \in [T]$.*

*Proof.* Let $\mathcal{S} = \{t_1, \ldots, t_{n+1}\}$ be its final value after running [Algorithm 2](#) for $T$ rounds, which means $n$ is the number of times the algorithm has increased the learning rate, $t_1 = 1$, and for $i = 2, \ldots, n+1$, $\eta_{t_i} = \eta_{t_{i-1}}\kappa$ holds. Let $\boldsymbol{A}_i = \sum_{j=1}^{i} \boldsymbol{H}_{t_j}$. Then for any $i > 1$, according to the update rule, there exists a vector $p \in \mathbb{R}^{d+1}$ such that $p^\top \boldsymbol{H}_{t_i} p \geq p^\top \boldsymbol{A}_{i-1} p$ and thus $p^\top \boldsymbol{A}_i p \geq 2 p^\top \boldsymbol{A}_{i-1} p$. Since a self-concordant function is strictly convex, $\boldsymbol{A}_i$ is positive definite for all $i \in [n]$. Therefore, let $q = \boldsymbol{A}_{i-1}^{\frac{1}{2}} p$ and we have $q^\top \boldsymbol{A}_{i-1}^{-\frac{1}{2}} \boldsymbol{A}_i \boldsymbol{A}_{i-1}^{-\frac{1}{2}} q \geq 2\|q\|_2^2$. This implies that the largest eigenvalue of $\boldsymbol{A}_{i-1}^{-\frac{1}{2}} \boldsymbol{A}_i \boldsymbol{A}_{i-1}^{-\frac{1}{2}}$ is at least 2. Furthermore, the smallest eigenvalue of $\boldsymbol{A}_{i-1}^{-\frac{1}{2}} \boldsymbol{A}_i \boldsymbol{A}_{i-1}^{-\frac{1}{2}}$ is at least 1 since

$$\boldsymbol{A}_{i-1}^{-\frac{1}{2}} \boldsymbol{A}_i \boldsymbol{A}_{i-1}^{-\frac{1}{2}} = \boldsymbol{A}_{i-1}^{-\frac{1}{2}} \left(\boldsymbol{A}_{i-1} + \boldsymbol{H}_{t_i}\right) \boldsymbol{A}_{i-1}^{-\frac{1}{2}} = I + \boldsymbol{A}_{i-1}^{-\frac{1}{2}} \boldsymbol{H}_{t_i} \boldsymbol{A}_{i-1}^{-\frac{1}{2}} \succeq I.$$

Therefore, we have

$$2 \leq \det(\boldsymbol{A}_{i-1}^{-\frac{1}{2}} \boldsymbol{A}_i \boldsymbol{A}_{i-1}^{-\frac{1}{2}}) = \frac{\det(\boldsymbol{A}_i)}{\det(\boldsymbol{A}_{i-1})},$$

which implies that $\det(\boldsymbol{A}_{n+1}) \geq 2^n \det(\boldsymbol{A}_1)$.

Next we show an upper bound for $\frac{\det(\boldsymbol{A}_{n+1})}{\det(\boldsymbol{A}_1)}$. Consider any $(d+1)$-dimensional unit vector $\boldsymbol{r}$. For each $i \in [n+1]$, applying [Lemma B.7](#) with $h = \boldsymbol{H}_1^{-\frac{1}{2}} \boldsymbol{r}$, $u = \boldsymbol{w}_{t_i}$ and $w = \boldsymbol{w}_1$, we have ,

$$\|h\|_{\boldsymbol{H}_{t_i}}^2 = \boldsymbol{r}^\top \boldsymbol{H}_1^{-\frac{1}{2}} \boldsymbol{H}_{t_i} \boldsymbol{H}_1^{-\frac{1}{2}} \boldsymbol{r} \leq \left(\frac{1 + 2400\nu}{1 - \pi_{\boldsymbol{w}_1}(\boldsymbol{w}_{t_i})}\right)^2 \|h\|_{\boldsymbol{H}_1}^2 \leq (1 + 2400\nu)^2 T^2.$$

Taking a summation over all $i \in [n+1]$, we obtain

$$\boldsymbol{r}^\top \boldsymbol{A}_1^{-\frac{1}{2}} \boldsymbol{A}_{n+1} \boldsymbol{A}_1^{-\frac{1}{2}} \boldsymbol{r} \leq (n+1)(1 + 2400\nu)^2 T^2,$$

which means that

$$\lambda_{\max}\left(\boldsymbol{A}_1^{-\frac{1}{2}} \boldsymbol{A}_{n+1} \boldsymbol{A}_1^{-\frac{1}{2}}\right) \leq (n+1)(1 + 2400\nu)^2 T^2,$$

and thus

$$\frac{\det(\boldsymbol{A}_{n+1})}{\det(\boldsymbol{A}_1)} = \det\left(\boldsymbol{A}_1^{-\frac{1}{2}} \boldsymbol{A}_{n+1} \boldsymbol{A}_1^{-\frac{1}{2}}\right) \leq \left((n+1)(1 + 2400\nu)^2 T^2\right)^{d+1}.$$

Combining with $\frac{\det(\boldsymbol{A}_{n+1})}{\det(\boldsymbol{A}_1)} \geq 2^n$, we have

$$n \leq (d+1)\log_2(n+1) + 2(d+1)\log_2\left((1 + 2400\nu)T\right) \leq ad \log_2(\nu T),$$

for $a = 100$. To show that $\eta_t \leq 5\eta$ for $t$, notice that $\exp(\log_2(\nu T)/\ln(\nu T)) \leq 5$. Therefore,

$$\eta_t \leq \kappa^n \eta = \exp\left(\frac{n}{ad \ln(\nu T)}\right)\eta \leq 5\eta,$$

finishing the proof. $\qquad\square$

The second lemma gives a lower bound of the Bregman divergence between $\boldsymbol{u}$ and $\boldsymbol{w}_t$, which contains an important term to cancel DEVIATION in later analysis.

**Lemma B.13.** *For all* $t \in [T]$, $D_\Psi(\boldsymbol{u}, \boldsymbol{w}_t) \geq -800\nu \ln(800\nu T) - 800\nu + \|\boldsymbol{u}\|_{\boldsymbol{H}_t}$.

*Proof.* Note again that $\Psi$ is a $800\nu$-normal barrier of $\boldsymbol{\Omega}$ by Lemma B.5. By the definition of Bregman divergence, we have

$$D_\Psi(\boldsymbol{u}, \boldsymbol{w}_t) = \Psi(\boldsymbol{u}) - \Psi(\boldsymbol{w}_t) - \langle \nabla\Psi(\boldsymbol{w}_t), \boldsymbol{u} - \boldsymbol{w}_t \rangle$$
$$\geq -800\nu \ln \frac{-\boldsymbol{u}^\top \nabla\Psi(\boldsymbol{w}_t)}{800\nu} - \langle \nabla\Psi(\boldsymbol{w}_t), \boldsymbol{u} \rangle - 800\nu. \quad \text{(Lemma B.6 and Lemma B.8)}$$

According to Lemma B.7 and Lemma B.8, we know that

$$\left| \boldsymbol{u}^\top \nabla\Psi(\boldsymbol{w}_t) \right| \leq \left( \frac{800\nu}{1 - \pi_{\boldsymbol{w}_1}(\boldsymbol{w}_t)} \right) \|\boldsymbol{u}\|_{\boldsymbol{H}_1} \leq 800\nu T \|\boldsymbol{u}\|_{\boldsymbol{H}_1} \leq 640000\nu^2 T.$$

On the other hand, according to Lemma B.3, we have

$$-\nabla\Psi(\boldsymbol{w}_t)^\top \boldsymbol{u} \geq \|\boldsymbol{u}\|_{\boldsymbol{H}_t}.$$

Combining everything, we have

$$D_\Psi(\boldsymbol{u}, \boldsymbol{w}_t) \geq -800\nu \left( \ln(800\nu T) + 1 \right) + \|\boldsymbol{u}\|_{\boldsymbol{H}_t},$$

finishing the proof. $\qquad \square$

The third lemma gives a bound for the so-called stability term.

**Lemma B.14.** *If* $\eta \leq \frac{1}{80d}$, *then Algorithm 2 guarantees* $\|\boldsymbol{w}_t - \boldsymbol{w}_{t+1}\|_{\boldsymbol{H}_t} \leq 40\eta \|\widehat{\boldsymbol{\ell}}_t\|_{\boldsymbol{H}_t^{-1}}$ *for all* $t \in [T]$.

*Proof.* Let $F_t(\boldsymbol{w}) = \left\langle \boldsymbol{w}, \widehat{\boldsymbol{\ell}}_t \right\rangle + \frac{1}{\eta_t} D_\Psi(\boldsymbol{w}, \boldsymbol{w}_t)$. We have

$$F_t(\boldsymbol{w}_t) - F_t(\boldsymbol{w}_{t+1}) = (\boldsymbol{w}_t - \boldsymbol{w}_{t+1})^\top \widehat{\boldsymbol{\ell}}_t - \frac{1}{\eta_t} D_\Psi(\boldsymbol{w}_{t+1}, \boldsymbol{w}_t)$$
$$\leq (\boldsymbol{w}_t - \boldsymbol{w}_{t+1})^\top \widehat{\boldsymbol{\ell}}_t \leq \|\boldsymbol{w}_t - \boldsymbol{w}_{t+1}\|_{\boldsymbol{H}_t} \cdot \|\widehat{\boldsymbol{\ell}}_t\|_{\boldsymbol{H}_t^{-1}}, \qquad (13)$$

where the last line uses the nonnegativity of Bregman divergence and also Hölder's inequality. On the other hand, by Taylor's theorem, there exists a point $\boldsymbol{\xi}$ on the segment connecting $\boldsymbol{w}_t$ and $\boldsymbol{w}_{t+1}$ such that

$$F_t(\boldsymbol{w}_t) - F_t(\boldsymbol{w}_{t+1})$$
$$= \nabla F_t(\boldsymbol{w}_{t+1})^\top (\boldsymbol{w}_t - \boldsymbol{w}_{t+1}) + \frac{1}{2}(\boldsymbol{w}_t - \boldsymbol{w}_{t+1}) \nabla^2 F_t(\boldsymbol{\xi})(\boldsymbol{w}_t - \boldsymbol{w}_{t+1})$$
$$\geq \frac{1}{2}(\boldsymbol{w}_t - \boldsymbol{w}_{t+1}) \nabla^2 F_t(\boldsymbol{\xi})(\boldsymbol{w}_t - \boldsymbol{w}_{t+1})$$
$$\qquad\qquad \text{(by first order optimality of } \boldsymbol{w}_{t+1} = \operatorname{argmin}_{\boldsymbol{w} \in \boldsymbol{\Omega}'} F_t(\boldsymbol{w}))$$
$$= \frac{1}{2\eta_t} \|\boldsymbol{w}_t - \boldsymbol{w}_{t+1}\|_{\nabla^2\Psi(\boldsymbol{\xi})}^2. \qquad (14)$$

Next we will prove $\|\boldsymbol{w}_t - \boldsymbol{w}_{t+1}\|_{\nabla^2\Psi(\boldsymbol{\xi})} \geq \frac{1}{2}\|\boldsymbol{w}_t - \boldsymbol{w}_{t+1}\|_{\boldsymbol{H}_t}$. To do so, we first show $\|\boldsymbol{w}_t - \boldsymbol{w}_{t+1}\|_{\boldsymbol{H}_t} \leq \frac{1}{2}$. It is in turn sufficient to show

$$F_t(\boldsymbol{w}') \geq F_t(\boldsymbol{w}_t), \text{ for all } \boldsymbol{w}' \text{ such that } \|\boldsymbol{w}' - \boldsymbol{w}_t\|_{\boldsymbol{H}_t} = \frac{1}{2},$$

since $\boldsymbol{w}_{t+1}$ is the minimizer of the convex function $F_t$. Indeed, using Taylor's theorem again and denoting $\boldsymbol{w}' - \boldsymbol{w}_t$ by $\boldsymbol{h}$, we have a point $\boldsymbol{\xi}'$ on the segment between $\boldsymbol{w}'$ and $\boldsymbol{w}_t$ such that

$$
\begin{aligned}
F_t\left(\boldsymbol{w}'\right) &= F_t\left(\boldsymbol{w}_t\right) + \nabla F_t\left(\boldsymbol{w}_t\right)^\top \boldsymbol{h} + \frac{1}{2}\boldsymbol{h}^\top \nabla^2 F_t(\boldsymbol{\xi}')\boldsymbol{h} \\
&= F_t\left(\boldsymbol{w}_t\right) + \widehat{\boldsymbol{\ell}}_t^\top \boldsymbol{h} + \frac{1}{2\eta_t}\|\boldsymbol{h}\|^2_{\nabla^2 \Psi(\boldsymbol{\xi}')} \\
&\geq F_t\left(\boldsymbol{w}_t\right) + \widehat{\boldsymbol{\ell}}_t^\top \boldsymbol{h} + \frac{1}{2\eta_t}\|\boldsymbol{h}\|^2_{\boldsymbol{H}_t}\left(1 - \|\boldsymbol{w}_t - \boldsymbol{\xi}'\|_{\boldsymbol{H}_t}\right)^2 && \text{(Lemma B.1)} \\
&\geq F_t\left(\boldsymbol{w}_t\right) + \widehat{\boldsymbol{\ell}}_t^\top \boldsymbol{h} + \frac{1}{160\eta} && (\|\boldsymbol{h}\|_{\boldsymbol{H}_t} = \tfrac{1}{2}, \|\boldsymbol{w}_t - \boldsymbol{\xi}'\|_{\boldsymbol{H}_t} \leq \tfrac{1}{2}, \text{ and Lemma B.12}) \\
&\geq F_t\left(\boldsymbol{w}_t\right) - \|\widehat{\boldsymbol{\ell}}_t\|_{\boldsymbol{H}_t^{-1}}\|\boldsymbol{h}\|_{\boldsymbol{H}_t} + \frac{1}{160\eta} && \text{(Hölder's inequality)} \\
&\geq F_t\left(\boldsymbol{w}_t\right) - \frac{d}{2} + \frac{1}{160\eta}. && (\|\widehat{\boldsymbol{\ell}}_t\|_{\boldsymbol{H}_t^{-1}} \leq d\,|\langle \widetilde{w}_t, \ell_t\rangle| \leq d)
\end{aligned}
$$

Under the condition $\eta \leq \frac{1}{80d}$ we have thus shown $F_t(\boldsymbol{w}') \geq F_t(\boldsymbol{w}_t)$ and consequently $\|\boldsymbol{w}_t - \boldsymbol{w}_{t+1}\|_{\boldsymbol{H}_t} \leq \frac{1}{2}$ and $\|\boldsymbol{w}_t - \boldsymbol{\xi}\|_{\boldsymbol{H}_t} \leq \frac{1}{2}$. Now according to Lemma B.1 again, we have

$$
\|\boldsymbol{w}_t - \boldsymbol{w}_{t+1}\|_{\nabla^2 \Psi(\boldsymbol{\xi})} \geq \|\boldsymbol{w}_t - \boldsymbol{w}_{t+1}\|_{\boldsymbol{H}_t}(1 - \|\boldsymbol{w}_t - \boldsymbol{\xi}\|_{\boldsymbol{H}_t}) \geq \frac{1}{2}\|\boldsymbol{w}_t - \boldsymbol{w}_{t+1}\|_{\boldsymbol{H}_t}.
$$

Plugging it into Eq. (14) and combining Eq. (13) give

$$
\|\widehat{\boldsymbol{\ell}}_t\|_{\boldsymbol{H}_t^{-1}} \geq \frac{1}{8\eta_t}\|\boldsymbol{w}_t - \boldsymbol{w}_{t+1}\|_{\boldsymbol{H}_t} \geq \frac{1}{40\eta}\|\boldsymbol{w}_t - \boldsymbol{w}_{t+1}\|_{\boldsymbol{H}_t},
$$

where the last inequality uses Lemma B.12. Rearranging finishes the proof. $\square$

Now we are ready to prove the bound for REG-TERM stated in Lemma B.11.

**Proof of Lemma B.11.** We first verify that $\boldsymbol{u}$ is in $\Omega'$. Indeed, according to the definition of $\boldsymbol{u}$, we have

$$
\boldsymbol{w}_1 + \frac{1}{1 - \frac{1}{T}} \cdot (\boldsymbol{u} - \boldsymbol{w}_1) = \boldsymbol{u}^\star \in \Omega,
$$

which by the definition of Minkowsky function shows that $\pi_{\boldsymbol{w}_1}(\boldsymbol{u}) \leq 1 - 1/T$ and thus $\boldsymbol{u} \in \Omega'$. According to the standard analysis of Online Mirror Descent, for example, Lemma 6 of [37], we then have

$$
\left\langle \boldsymbol{w}_t, \widehat{\boldsymbol{\ell}}_t \right\rangle - \left\langle \boldsymbol{u}, \widehat{\boldsymbol{\ell}}_t \right\rangle \leq D_{\Psi_t}(\boldsymbol{u}, \boldsymbol{w}_t) - D_{\Psi_t}(\boldsymbol{u}, \boldsymbol{w}_{t+1}) + \left\langle \boldsymbol{w}_t - \boldsymbol{w}_{t+1}, \widehat{\boldsymbol{\ell}}_t \right\rangle. \tag{15}
$$

We first focus on the term $D_{\Psi_t}(\boldsymbol{u}, \boldsymbol{w}_t) - D_{\Psi_t}(\boldsymbol{u}, \boldsymbol{w}_{t+1})$. Taking a summation over $t = 1, 2, \ldots, T$, we have

$$
\begin{aligned}
\sum_{t=1}^{T} D_{\Psi_t}(\boldsymbol{u}, \boldsymbol{w}_t) - D_{\Psi_t}(\boldsymbol{u}, \boldsymbol{w}_{t+1}) &\leq D_{\Psi_1}(\boldsymbol{u}, \boldsymbol{w}_1) + \sum_{t=1}^{T-1}\left(D_{\Psi_{t+1}}(\boldsymbol{u}, \boldsymbol{w}_{t+1}) - D_{\Psi_t}(\boldsymbol{u}, \boldsymbol{w}_{t+1})\right) \\
&\leq D_{\Psi_1}(\boldsymbol{u}, \boldsymbol{w}_1) + \sum_{i=2}^{n}\left(\frac{1}{\eta_{t_i}} - \frac{1}{\eta_{t_i - 1}}\right) D_{\Psi}(\boldsymbol{u}, \boldsymbol{w}_{t_i}),
\end{aligned}
$$

where we recall the definition of $t_1, \ldots, t_n$ defined in the beginning of the proof of Lemma B.12. The first term can be bounded by

$$
\begin{aligned}
D_{\Psi_1}(\boldsymbol{u}, \boldsymbol{w}_1) = \frac{1}{\eta} D_{\Psi}(\boldsymbol{u}, \boldsymbol{w}_1) &= \frac{\Psi(\boldsymbol{u}) - \Psi(\boldsymbol{w}_1)}{\eta} - \frac{1}{\eta} \cdot \langle \nabla\Psi(\boldsymbol{w}_1), \boldsymbol{u} - \boldsymbol{w}_1\rangle \\
&\leq \frac{\Psi(\boldsymbol{u}) - \Psi(\boldsymbol{w}_1)}{\eta} && \text{(by first order optimality of } \boldsymbol{w}_1) \\
&\leq \frac{800\nu \ln T}{\eta}. && \text{(Lemma B.7)}
\end{aligned}
$$

For the second term, using $1 - \kappa \leq -\frac{1}{ad\ln(\nu T)}$ for $a = 100$ and Lemma B.12, we have

$$\frac{1}{\eta_{t_i}} - \frac{1}{\eta_{t_i-1}} \leq \frac{1-\kappa}{\eta_{t_i}} \leq -\frac{1}{5\eta ad\ln(\nu T)}.$$

Therefore,

$$\sum_{t=1}^{T} D_{\Psi_t}(\boldsymbol{u}, \boldsymbol{w}_t) - D_{\Psi_t}(\boldsymbol{u}, \boldsymbol{w}_{t+1})$$

$$\leq \frac{800\nu\ln T}{\eta} - \sum_{i=2}^{n} \frac{1}{5\eta ad\ln(\nu T)} \cdot D_{\Psi}(\boldsymbol{u}, \boldsymbol{w}_{t_i})$$

$$\leq \tilde{\mathcal{O}}\left(\frac{\nu}{\eta}\right) - \frac{1}{5\eta ad\ln(\nu T)} \cdot \sum_{i=2}^{n} \left(\|\boldsymbol{u}\|_{\boldsymbol{H}_{t_i}} - 800\nu - 800\nu\ln(800\nu T)\right) \qquad \text{(Lemma B.13)}$$

$$= \tilde{\mathcal{O}}\left(\frac{\nu}{\eta}\right) - \frac{1}{5\eta ad\ln(\nu T)} \sum_{i=2}^{n} \|\boldsymbol{u}\|_{\boldsymbol{H}_{t_i}}.$$

For the second term in Eq. (15), that is, $\left\langle \boldsymbol{w}_t - \boldsymbol{w}_{t+1}, \widehat{\boldsymbol{\ell}}_t \right\rangle$, taking summation over $t \in [T]$ we have

$$\sum_{t=1}^{T} \left\langle \boldsymbol{w}_t - \boldsymbol{w}_{t+1}, \widehat{\boldsymbol{\ell}}_t \right\rangle \leq \sum_{t=1}^{T} \|\boldsymbol{w}_t - \boldsymbol{w}_{t+1}\|_{\boldsymbol{H}_t} \|\widehat{\boldsymbol{\ell}}_t\|_{\boldsymbol{H}_t^{-1}} \qquad \text{(Hölder's inequality)}$$

$$\leq 40\eta \sum_{t=1}^{T} \|\widehat{\boldsymbol{\ell}}_t\|_{\boldsymbol{H}_t^{-1}}^2 \qquad \text{(Lemma B.14)}$$

$$= 40\eta \sum_{t=1}^{T} d^2 \langle \widetilde{w}_t, \ell_t \rangle^2 \boldsymbol{s}_t^\top \boldsymbol{H}_t^{1/2} \boldsymbol{H}_t^{-1} \boldsymbol{H}_t^{1/2} \boldsymbol{s}_t$$

$$\leq 40\eta \sum_{t=1}^{T} d^2 |\langle \widetilde{w}_t, \ell_t \rangle| = 40\eta d^2 \overline{L}_T.$$

Combining everything finishes the proof. $\qquad\qquad\qquad\qquad\qquad\qquad\qquad\qquad\square$

### B.3.3   Proof of Theorem 3.1

To prove Theorem 3.1, we first prove the following main lemma.

**Lemma B.15.** *Algorithm 2 with $\eta \leq \frac{1}{640aCd^2\ln(\nu T)\ln(C/\delta)}$ guarantees that with probability at least $1 - \delta$,*

$$\sum_{t=1}^{T} \langle \widetilde{w}_t - u^\star, \ell_t \rangle$$

$$\leq \tilde{\mathcal{O}}\left(\frac{\nu}{\eta} + \eta d^2\overline{L}_T + \sqrt{\overline{L}_u\ln(1/\delta)}\right) + (\sqrt{\nu} + \rho)\left(161Cd\sqrt{\ln(C/\delta)\mathring{L}_T} - \frac{1}{10\eta ad\ln(\nu T)}\right),$$

*where $a = 100$, $C = \Theta(\ln^2(d\nu T))$ is defined in Lemma B.9, and we recall all other notations defined in Equations (8)-(10).*

*Proof.* Recall the decomposition of regret shown in Eq. (7). Combining the result of Lemma B.10 and Lemma B.11, we have when $\eta \leq \frac{1}{80d}$,

$$
\begin{aligned}
\sum_{t=1}^{T} \langle \widetilde{w}_t - u^\star, \ell_t \rangle &\leq \tilde{\mathcal{O}}\left(\frac{\nu}{\eta}\right) - \frac{\sum_{s \in \mathcal{S}'} \|\boldsymbol{u}\|_{\boldsymbol{H}_s}}{5\eta a d \ln(\nu T)} + 40\eta d^2 \overline{L}_T + 64Cd\left(\sqrt{\nu} + \rho\right)\ln(C/\delta) \\
&\quad + 161Cd\sqrt{(\nu + \rho^2)\,\mathring{L}_T \ln(C/\delta)} + C\sqrt{32\overline{L}_u \ln(C/\delta)}. \\
&= \tilde{\mathcal{O}}\left(\frac{\nu}{\eta} + \eta d^2 \overline{L}_T + \sqrt{\overline{L}_u \ln(C/\delta)}\right) - \frac{\sum_{s \in \mathcal{S}'} \|\boldsymbol{u}\|_{\boldsymbol{H}_s}}{5\eta a d \ln(\nu T)} \\
&\quad + 64Cd\left(\sqrt{\nu} + \rho\right)\ln(C/\delta) + 161C\sqrt{d^2\,(\nu + \rho^2)\,\mathring{L}_T \ln(C/\delta)}. \quad (16)
\end{aligned}
$$

Now consider the value of $\rho = \|\boldsymbol{u}\|_{\boldsymbol{H}_{t^\star}}$ where $t^\star \in \arg\max_{t \in [T]} \|\boldsymbol{u}\|_{\boldsymbol{H}_t}$, compared to the negative term above. Suppose $t^\star \in \mathcal{S}$, then we have

$$
\rho \leq \max\left\{\|\boldsymbol{u}\|_{\boldsymbol{H}_1}, \sum_{s \in \mathcal{S}'} \|\boldsymbol{u}\|_{\boldsymbol{H}_s}\right\} \leq 800\nu + \sum_{s \in \mathcal{S}'} \|\boldsymbol{u}\|_{\boldsymbol{H}_s},
$$

where we use Lemma B.8 again to bound $\|\boldsymbol{u}\|_{\boldsymbol{H}_1}$. On the other hand, if $t^\star \notin \mathcal{S}$, then according to the update rule of $\mathcal{S}$ in Algorithm 2, we have $\boldsymbol{H}_{t^\star} \preceq \boldsymbol{H}_1 + \sum_{s \in \mathcal{S}'} \boldsymbol{H}_s$, which means

$$
\rho = \sqrt{\|\boldsymbol{u}\|_{\boldsymbol{H}_{t^\star}}^2} \leq \sqrt{\|\boldsymbol{u}\|_{\boldsymbol{H}_1}^2 + \sum_{s \in \mathcal{S}'} \|\boldsymbol{u}\|_{\boldsymbol{H}_s}^2} \leq 800\nu + \sum_{s \in \mathcal{S}'} \|\boldsymbol{u}\|_{\boldsymbol{H}_s}.
$$

Therefore, we continue to bound the last three terms in Eq. (16) as

$$
\begin{aligned}
&\frac{800\nu - \rho}{5\eta a d \ln(\nu T)} + 64Cd\left(\sqrt{\nu} + \rho\right)\ln(C/\delta) + 161Cd\sqrt{(\nu + \rho^2)\,\mathring{L}_T \ln(C/\delta)} \\
&\leq \mathcal{O}\left(\frac{\nu}{\eta}\right) - \frac{\sqrt{\nu} + \rho}{5\eta a d \ln(\nu T)} + 64Cd\left(\sqrt{\nu} + \rho\right)\ln(C/\delta) + 161Cd\sqrt{(\nu + \rho^2)\,\mathring{L}_T \ln(C/\delta)} \\
&\leq \mathcal{O}\left(\frac{\nu}{\eta}\right) - \frac{\sqrt{\nu} + \rho}{10\eta a d \ln(\nu T)} + 161Cd\sqrt{(\nu + \rho^2)\,\mathring{L}_T \ln(C/\delta)} \quad (\eta \leq \frac{1}{640aCd^2 \ln(\nu T) \ln(C/\delta)}) \\
&\leq \mathcal{O}\left(\frac{\nu}{\eta}\right) + \left(\sqrt{\nu} + \rho\right)\left(161Cd\sqrt{\ln(C/\delta)\mathring{L}_T} - \frac{1}{10\eta a d \ln(\nu T)}\right).
\end{aligned}
$$

Plugging this back into Eq. (16) finishes the proof. $\qquad\square$

Now we are ready to prove the main theorem. For convenience, we restate the theorem below.

**Theorem B.16.** *Algorithm 2 with an appropriate choice of $\eta$ ensures that with probability at least $1 - \delta$:*

$$
\text{Reg} = \begin{cases} \tilde{\mathcal{O}}(d^2\nu\sqrt{T \ln\frac{1}{\delta}} + d^2\nu \ln\frac{1}{\delta}), & \text{against an oblivious adversary;} \\ \tilde{\mathcal{O}}(d^2\nu\sqrt{dT \ln\frac{1}{\delta}} + d^3\nu \ln\frac{1}{\delta}), & \text{against an adaptive adversary.} \end{cases}
$$

*Moreover, if $\langle w, \ell_t \rangle \geq 0$ for all $w \in \Omega$ and all $t$, then $T$ in the bounds above can be replaced by $L^\star = \min_{u \in \Omega} \sum_{t=1}^{T} \langle u, \ell_t \rangle$, that is, the total loss of the best action.*

*Proof.* Using Lemma B.15 and the fact that $|\langle \widetilde{w}_t, \ell_t \rangle| \leq 1$ and $|\langle u, \ell_t \rangle| \leq 1$ for all $t \in [T]$, we have

$$
\sum_{t=1}^{T} \langle \widetilde{w}_t - u^\star, \ell_t \rangle \leq \tilde{\mathcal{O}}\left(\frac{\nu}{\eta} + \eta d^2 T + \sqrt{T \ln\frac{1}{\delta}}\right) + \left(\sqrt{\nu} + \rho\right)\left(161Cd\sqrt{T \ln(C/\delta)} - \frac{1}{10\eta a d \ln(\nu T)}\right).
$$

With

$$
\eta = \min\left\{\frac{1}{640aCd^2 \ln(\nu T) \ln(C/\delta)}, \frac{1}{1610aCd^2 \ln(\nu T)\sqrt{T \ln(C/\delta)}}\right\},
$$

the last term becomes nonpositive, and we arrive at

$$\sum_{t=1}^{T} \langle \widetilde{w}_t - u^\star, \ell_t \rangle \leq \tilde{\mathcal{O}} \left( d^2 \nu \sqrt{T \ln \frac{1}{\delta}} + d^2 \nu \ln \frac{1}{\delta} \right), \tag{17}$$

for any fixed $u^\star \in \Omega$, which completes the proof for the oblivious case. To obtain a bound for an adaptive adversary, we discrete the feasible set $\Omega$ and then take a union bound. Specifically, define $B_\Omega$ as follows:

$$B_\Omega \triangleq \lceil \alpha \rceil \lceil \beta \rceil, \quad \alpha \triangleq \max_{w,w' \in \Omega} \|w - w'\|_\infty, \quad \beta \triangleq \max_{\ell \in \Omega^\circ} \|\ell\|_\infty,$$

where $\Omega^\circ \triangleq \{\ell : |\langle w, \ell \rangle| \leq 1, \ \forall w \in \Omega\}$ is the set of feasible loss vectors. Then we discretize $\Omega$ into a finite set $\overline{\Omega}$ of $(B_\Omega T)^d$ points, such that for any $u^\star \in \Omega$, there exists $\overline{u} \in \overline{\Omega}$, such that $\|\overline{u} - u^\star\|_\infty \leq \frac{1}{\lceil \beta \rceil T}$. This means that

$$\left| \sum_{t=1}^{T} \langle \overline{u} - u^\star, \ell_t \rangle \right| \leq \sum_{t=1}^{T} \frac{d}{\lceil \beta \rceil T} \cdot \max_i \ell_{t,i} \leq d.$$

Therefore, it suffices to only consider regret against the points in $\overline{\Omega}$. Taking a union bound and replacing $\delta$ with $\frac{\delta}{(B_\Omega T)^d}$ in Eq. (17) finish the proof for the worst-case bound for adaptive adversaries.

In the remaining of the proof, we show that if $\langle w, \ell_t \rangle \in [0, 1]$ for all $w \in \Omega$ and $t \in [T]$, $T$ can be replaced by $L^\star$ in both bounds. As $\langle w, \ell_t \rangle$ is always positive, we have $\mathbb{E}_t [|\langle \widetilde{w}_t, \ell_t \rangle|] = \mathbb{E}_t [\langle \widetilde{w}_t, \ell_t \rangle] = \langle w_t, \ell_t \rangle$, $\overline{L}_u = \sum_{t=1}^{T} \langle u, \ell_t \rangle \leq L^\star + 2$, and $\overline{L}_T = L_T = \sum_{t=1}^{T} \langle \widetilde{w}_t, \ell_t \rangle$. Using standard Freedman's inequality, we have with probability at least $1 - \delta$,

$$\mathring{L}_T - \overline{L}_T \leq \frac{\mathring{L}_T}{2} + 3 \ln(1/\delta).$$

Rearranging gives

$$\mathring{L}_T \leq 2 L_T + 6 \ln(1/\delta).$$

Using Lemma B.15 again, we have

$$\sum_{t=1}^{T} \langle \widetilde{w}_t - u^\star, \ell_t \rangle \leq \tilde{\mathcal{O}} \left( \frac{\nu}{\eta} + \eta d^2 L_T + \sqrt{L^\star \ln \frac{1}{\delta}} \right)$$

$$+ (\sqrt{\nu} + \rho) \left( 161 C d \sqrt{\ln(C/\delta) \left( 2 L_T + 6 \ln \frac{1}{\delta} \right)} - \frac{1}{10 \eta a d \ln(\nu T)} \right).$$

With $\eta = \min \left\{ \frac{1}{640 a C d^2 \ln(\nu T) \ln(C/\delta)}, \frac{1}{1610 a C d^2 \ln(\nu T) \sqrt{(2 L_T + 6 \ln(1/\delta)) \ln(C/\delta)}} \right\}$, the last term becomes nonpositive, and we arrive at

$$\sum_{t=1}^{T} \langle \widetilde{w}_t - u^\star, \ell_t \rangle \leq \tilde{\mathcal{O}} \left( d^2 \nu \sqrt{L_T \ln \frac{1}{\delta}} + \sqrt{L^\star \ln \frac{1}{\delta}} + d^2 \nu \ln \frac{1}{\delta} \right)$$

Solving the quadratic inequality in terms of $\sqrt{L_T}$ gives the following high probability regret bound

$$\sum_{t=1}^{T} \langle \widetilde{w}_t - u^\star, \ell_t \rangle \leq \tilde{\mathcal{O}} \left( d^2 \nu \sqrt{L^\star \ln \frac{1}{\delta}} + d^2 \nu \ln \frac{1}{\delta} \right).$$

This finishes the proof for the case with oblivious adversaries, and the case with adaptive adversaries is again by taking a union bound as done earlier. $\square$

**Remark 2.** The tuning of $\eta$ in the proof above depends on the unknown quantity $L_T$. In fact, the issue seems even more severe than that pointed out in Remark 1 because $L_T$ depends on the algorithm's behavior, which in turns depends on $\eta$ itself. We point out that, however, this can again be addressed using a doubling trick, making the algorithm completely parameter-free. We omit the details but refer the reader to Lee et al. [27, Algorithm 4] for very similar ideas.

# C  Omitted details for Section 4

## C.1  Preliminary

In this section, we introduce the concept of *occupancy measure* (used in previous works already; see [25]), which helps reformulate adversarial MDP problems in a way very similar to adversarial MAB problems. For a state $x$, let $k(x)$ denote the index of the layer to which state $x$ belongs. Given a policy $\pi$ and a transition function $P$, we define occupancy measure $w^{P,\pi} \in \mathbb{R}^{X \times A \times X}$ as follows:

$$w^{P,\pi}(x, a, x') = \mathbb{P}\left[x_k = x, a_k = a, x_{k+1} = x' | P, \pi\right],$$

where $k = k(x)$. In other words, $w^{P,\pi}(x, a, x')$ is the probability of visiting the triple $(x, a, x')$ if we execute policy $\pi$ in an MDP with transition function $P$.

According to this definition, we have the following two properties for any occupancy measure $w$. First, based on the layered structure, we know that each layer is visited exactly once in each episode, which means for each $k = 0, 1, \ldots, J$, we have

$$\sum_{x \in X_k, a \in A, x' \in X_{k+1}} w(x, a, x') = 1. \tag{18}$$

Second, the probability of entering one state when coming from the previous layer equals to the probability of leaving the state to the next layer. Therefore, for each $k = 1, 2, \ldots, J - 1$, we have

$$\sum_{x' \in X_{k-1}, a \in A} w(x', a, x) = \sum_{x' \in X_{k+1}, a \in A} w(x, a, x'), \tag{19}$$

for all $x \in X_k$.

Moreover, the following lemma shows that if $w$ satisfies the above two properties, then $w$ is an occupancy measure with respect to some transition function $P^w$ and policy $\pi^w$.

**Lemma C.1** (Lemma 3.1 in [34]).  *For any $w \in [0, 1]^{|X| \times |A| \times |X|}$, it satisfies Eq. (18) and Eq. (19) if and only if it is a valid occupancy measure associated with the following induced transition function $P^w$ and policy $\pi^w$:*

$$P^w(x'|x, a) = \frac{w(x, a, x')}{\sum_{y \in X_{k(x)+1}} w(x, a, y)}, \quad \pi^w(a|x) = \frac{\sum_{x' \in X_{k(x)+1}} w(x, a, x')}{\sum_{a' \in A} \sum_{x' \in X_{k(x)+1}} w(x, a', x')}.$$

Following [25], we denote by $\Delta$ the set of all valid occupancy measures. For a fixed transition function, we denote by $\Delta(P) \subseteq \Delta$ the set of occupancy measures whose induced transition function $P^w$ is exactly $P$. In addition, we denote by $\Delta(\mathcal{P}) \subseteq \Delta$ the set of occupancy measures whose induced transition function $P^w$ belongs to a set of transition functions $\mathcal{P}$. With a slightly abuse of notation, we define $w(x, a) = \sum_{x' \in X_{k(x)+1}} w(x, a, x')$ for all $x \neq x_J$ and $a \in A$. Using the notations introduced above, we know that the expected loss of using policy $\pi$ at round $t$ is exactly $\langle w^{P,\pi}, \ell_t \rangle \triangleq \sum_{x,a} w^{P,\pi}(x, a)\ell_t(x, a)$. Let $\pi_t$ be the policy chosen at round $t$. Then the total expected loss (with respect to randomness of the transition function) is $\sum_{t=1}^{T} \langle w^{P,\pi_t}, \ell_t \rangle$ and the total regret can be written as:

$$\text{Reg} = \sum_{t=1}^{T} \ell_t(\pi_t) - \min_{\pi} \sum_{t=1}^{T} \ell_t(\pi) = \sum_{t=1}^{T} \langle w_t - u^\star, \ell_t \rangle = L_T - L^\star, \tag{20}$$

where $u^\star = w^{P,\pi^\star}$ is the occupancy measure induced by the optimal policy $\pi^\star = \arg\min_{\pi} \sum_{t=1}^{T} \ell_t(\pi)$, $w_t = w^{P,\pi_t}$, $L_T \triangleq \sum_{t=1}^{T} \langle w_t, \ell_t \rangle$, and $L^\star \triangleq \sum_{t=1}^{T} \langle u^\star, \ell_t \rangle$. When the regret is written in this way, it is clear that the problem is very similar to MAB or linear bandits with $\Delta(P)$ being the decision set and $\ell_t$ parametrizing the linear loss function at time $t$.

## C.2  Algorithm for MDPs

In this section, we introduce our algorithm that achieves high-probability small-loss regret bound for the MDP setting. The full pseudocode of the algorithm is shown in Algorithm 4. The algorithm is very similar to UOB-REPS introduced in [25], except for the following two modifications.

---

**Algorithm 4** Upper Occupancy Bound Log Barrier Policy Search

---

**Input:** state space $X$, action space $A$, learning rate $\eta$, and confidence parameter $\delta$.

**Define:** $\kappa = e^{\frac{1}{T \ln T}}$, Comp-UOB is Algorithm 3 of [25], and

$$\Omega = \left\{ \hat{w} : \hat{w}(x, a, x') \geq \frac{1}{T^3 |X|^2 |A|}, \forall k \in \{0, 1, \dots, J-1\}, x \in X_k, a \in A, x' \in X_{k+1} \right\}.$$

**Initialization:** Set epoch index $i = 1$ and confidence set $\mathcal{P}_1$ as the set of all transition functions. For all $k = 0, \dots, J-1$, $(x, a, x') \in X_k \times A \times X_{k+1}$, set

$$\widehat{w}_1(x, a, x') = \frac{1}{|X_k||A||X_{k+1}|}, \quad \pi_1 = \pi^{\widehat{w}_1}, \quad \eta_1(x, a) = \eta, \quad \rho_1(x, a) = 2|X_k||A|,$$

$$\phi_1(x, a) = \text{Comp-UOB}(\pi_1, x, a, \mathcal{P}_1), \quad N_0(x, a) = N_1(x, a) = G_0(x'|x, a) = G_1(x'|x, a) = 0.$$

1 **for** $t = 1, 2, \dots, T$ **do**
2      Execute policy $\pi_t$ for $J$ steps and obtain trajectory $x_k, a_k, \ell_t(x_k, a_k)$ for $k = 0, \dots, J-1$.
3      Construct loss estimators for all $(x, a) \in X \times A$:

$$\widehat{\ell}_t(x, a) = \frac{\ell_t(x, a)}{\phi_t(x, a)} \mathbb{1}_t(x, a), \quad \text{where} \quad \mathbb{1}_t(x, a) = \mathbb{1}\{x_{k(x)} = x, a_{k(x)} = a\}. \quad (21)$$

4      Update counters: for each $k = 0, 1, \dots, J-1$,

$$N_i(x_k, a_k) \leftarrow N_i(x_k, a_k) + 1, \quad G_i(x_{k+1}|x_k, a_k) \leftarrow G_i(x_{k+1}|x_k, a_k) + 1.$$

5      **if** $\exists k, N_i(x_k, a_k) \geq \max\{1, 2N_{i-1}(x_k, a_k)\}$ **then**
6          Increase epoch index $i \leftarrow i + 1$.
7          Initialize new counters: $N_i = N_{i-1}, G_i = G_{i-1}$ (copy all entries).
8          Compute confidence set

$$\mathcal{P}_i = \{\hat{P} : \left| \hat{P}(x'|x, a) - \bar{P}_i(x'|x, a) \right| \leq \epsilon_i(x'|x, a),$$

$$\forall (x, a, x') \in X_k \times A \times X_{k+1}, k = 0, 1, \dots, J-1\},$$

         where $\bar{P}_i(x'|x, a) = \frac{G_i(x'|x, a)}{\max\{1, N_i(x, a)\}}$ and

$$\epsilon_i(x'|x, a) \triangleq 4 \sqrt{\frac{\bar{P}_i(x'|x, a) \ln\left(\frac{T|X||A|}{\delta}\right)}{\max\{1, N_i(x, a) - 1\}}} + \frac{28 \ln\left(\frac{T|X||A|}{\delta}\right)}{3 \max\{1, N_i(x, a) - 1\}}.$$

9      Compute $\widehat{w}_{t+1} = \operatorname{argmin}_{w \in \Delta(\mathcal{P}_i) \cap \Omega} \left\{ \langle w, \widehat{\ell}_t \rangle + D_{\psi_t}(w, \widehat{w}_t) \right\}$, where

$$\psi_t(w) = \sum_{k=0}^{J-1} \sum_{(x, a, x') \in X_k \times A \times X_{k+1}} \frac{1}{\eta_t(x, a)} \ln\left(\frac{1}{w(x, a, x')}\right). \quad (22)$$

10     Update policy $\pi_{t+1} = \pi^{\widehat{w}_{t+1}}$.
11     **for** *each* $(x, a) \in X \times A$ **do**
12         Update upper occupancy bound:

$$\phi_{t+1}(x, a) = \max_{\hat{P} \in \mathcal{P}_i} w^{\hat{P}, \pi_{t+1}}(x, a) = \text{Comp-UOB}(\pi_{t+1}, x, a, \mathcal{P}_i). \quad (23)$$

13         **if** $\frac{1}{\phi_{t+1}(x,a)} \geq \rho_t(x, a)$ **then** $\rho_{t+1}(x, a) = \frac{2}{\phi_{t+1}(x,a)}, \eta_{t+1}(x, a) = \eta_t(x, a) \cdot \kappa$.
14         **else** $\rho_{t+1}(x, a) = \rho_t(x, a), \eta_{t+1}(x, a) = \eta_t(x, a)$.

---

First, in [25], they propose a loss estimator akin to the importance-weighted estimator using the so-called *upper occupancy bound*, denoted by $\phi_t(x, a)$ in our notation. Indeed, the actual probability

$w_t(x, a)$ of visiting state-action pair $(x, a)$ is unknown (due to the unknown transition), and thus standard unbiased importance-weighted estimators do not apply directly. Instead, since the algorithm maintains a confidence set $\mathcal{P}_i$ (for epoch $i$) of all the plausible transition functions based on observations, one can calculate the largest probability of visiting state-action pair $(x, a)$ under policy $\pi_t$, among all the plausible transition functions, which is exactly the definition of $\phi_t(x, a)$ and can be computed efficiently via the sub-routine COMP-UOB as shown in [25]. In addition, Jin et al. [25] also apply the idea of implicit exploration from [31] and introduce an extra bias with a parameter $\gamma > 0$, leading to the following loss estimator:

$$\widehat{\ell}_t(x, a) = \frac{\ell_t(x, a)}{\phi_t(x, a) + \gamma} \mathbb{1}_t(x, a),$$

which is crucial for them to derive a high-probability bound. As one can see in Eq. (21), the first difference of our algorithm is that we remove this implicit exploration (that is, $\gamma = 0$), similarly to our MAB algorithm in Section 2. As we later explain in Appendix C.4, removing this implicit exploration is important for obtaining a small-loss bound.

Second, while UOB-REPS uses the entropy regularizer with a fixed learning rate, we use the log-barrier regularizer with time-varying and individual learning rates for each state-action pair, defined in Eq. (22), which is a direct generalization of Algorithm 1 for MAB. The way we increase the learning rate is also essentially identical to the MAB case; see the last part of Algorithm 4. We also point out that the analogue of the clipped simplex used in Algorithm 1 is now $\Delta(\mathcal{P}_i) \cap \Omega$ where $\Delta(\mathcal{P}_i)$ is the set of occupancy measures with induced transition functions in the confidence set $\mathcal{P}_i$, and $\Omega$ (defined at the beginning of Algorithm 4) contains all $\widehat{w}$ with each entry not smaller than $1/(T^3|X|^2|A|)$, which ensures that the learning rates cannot be increased by too many times.

## C.3 Proof of Theorem 4.1

In this section, we analyze Algorithm 4 and prove Theorem 4.1. We start with decomposing the regret into five terms (recall the definitions of $w_t$ and $u^\star$ in Eq. (20) and $\widehat{w}_t$ and $\widehat{\ell}_t$ in Algorithm 4):

$$\sum_{t=1}^{T} \langle w_t - u^\star, \ell_t \rangle = \underbrace{\sum_{t=1}^{T} \langle w_t - \widehat{w}_t, \ell_t \rangle}_{\text{ERROR}} + \underbrace{\sum_{t=1}^{T} \left\langle \widehat{w}_t, \ell_t - \widehat{\ell}_t \right\rangle}_{\text{BIAS-1}} + \underbrace{\sum_{t=1}^{T} \left\langle \widehat{w}_t - u, \widehat{\ell}_t \right\rangle}_{\text{REG-TERM}}$$

$$+ \underbrace{\sum_{t=1}^{T} \left\langle u, \widehat{\ell}_t - \ell_t \right\rangle}_{\text{BIAS-2}} + \underbrace{\sum_{t=1}^{T} \langle u - u^\star, \ell_t \rangle}_{\text{BIAS-3}}.$$

Here $u$ is defined as

$$u = \left(1 - \frac{1}{T}\right) u^\star + \frac{1}{T|A|} \sum_{a \in A} w^{P_0, \pi_a}, \tag{24}$$

where $\pi_a$ is the policy that chooses action $a$ at every state, and the definition of the transition function $P_0$ is deferred to Lemma C.4. Note that $u$ is random in the case with adaptive adversaries.

In the remaining of this subsection, we first provide a few useful lemmas in Appendix C.3.1, and then bound ERROR in Appendix C.3.2, BIAS-1 in Appendix C.3.3, BIAS-2 in Appendix C.3.4, and REG-TERM in Appendix C.3.5. Note that BIAS-3 can be trivially bounded by $J$ as

$$\text{BIAS-3} = \sum_{t=1}^{T} \langle u - u^\star, \ell_t \rangle \leq \frac{1}{T|A|} \sum_{a \in A} \sum_{t=1}^{T} \langle w^{\pi_a, P_0}, \ell_t \rangle \leq J. \tag{25}$$

We finally put everything together and prove Theorem 4.1 in Appendix C.3.6.

### C.3.1 Useful lemmas

The first two lemmas are from [25].

**Lemma C.2** (Lemma 2 in [25]). *With probability at least $1 - 4\delta$, we have for all $k = 0, 1, \ldots, J - 1$ and $(x, a, x') \in X_k \times A \times X_{k+1}$,*

$$\left| P(x'|x, a) - \bar{P}_i(x'|x, a) \right| \leq \frac{\epsilon_i(x'|x, a)}{2}. \tag{26}$$

*Consequently, we have $P \in \mathcal{P}_i$ for all $i$.*

**Lemma C.3** (Lemma 10 in [25]). *With probability at least $1 - \delta$, we have for all $k = 0, \ldots, J - 1$,*

$$\sum_{t=1}^{T} \sum_{x \in X_k, a \in A} \frac{w_t(x, a)}{\max\{1, N_{i_t}(x, a)\}} = \tilde{\mathcal{O}}\left( |X_k| \cdot |A| + \ln \frac{1}{\delta} \right)$$

$$\sum_{t=1}^{T} \sum_{x \in X_k, a \in A} \frac{w_t(x, a) - \mathbb{1}_t(x, a)}{\sqrt{\max\{1, N_{i_t}(x, a)\}}} \leq \sum_{t=1}^{T} \sum_{x \in X_k, a \in A} \frac{w_t(x, a)}{\max\{1, N_{i_t}(x, a)\}} + \tilde{\mathcal{O}}\left( \ln \frac{1}{\delta} \right)$$

$$\leq \tilde{\mathcal{O}}\left( |X_k| \cdot |A| + \ln \frac{1}{\delta} \right),$$

*where $i_t$ is the index of the epoch to which episode $t$ belongs.*

Next, we prove a lemma showing that there exists a transition function $P_0$ that always lies in the confidence set $\mathcal{P}_i$ of the algorithm, such that for any action $a \in A$ and any two states $x, x'$ in consecutive layers, the probability of reaching $x'$ by taking action $a$ at state $x$ is at least $\frac{1}{T|X|}$.

**Lemma C.4.** *With probability at least $1 - 4\delta$, there exists $P_0 \in \cap_i \mathcal{P}_i$ such that for all $k < J, x \in X_k, a \in A$, and $x' \in X_{k+1}$, we have $P_0(x'|x, a) \geq \frac{1}{T|X|}$.*

*Proof.* The construction of $P_0$ is as follows. First we start with $P_0 = P$. Then for each fixed $(x, a)$, we focus on the distribution $P_0(\cdot|x, a)$. In particular, for all $x' \in X_{k(x)+1}$ such that $P_0(x'|x, a) < \frac{1}{T|X|}$, we move the weight from the largest entry of $P_0(\cdot|x, a)$ to this entry so that $P_0(x'|x, a) = \frac{1}{T|X|}$ and $P_0(\cdot|x, a)$ remains a valid distribution. Repeat the same for all $(x, a)$ pairs finishes the construction of $P_0$.

Clearly, $P_0$ satisfies $P_0(x'|x, a) \geq \frac{1}{T|X|}$, and it remains to show $P_0 \in \mathcal{P}_i$ for all $i$. To this end, we first note that

$$|P_0(x'|x, a) - P(x'|x, a)| \leq \frac{|X_{k(x')}|}{T|X|} \leq \frac{1}{T} \leq \frac{\epsilon_i(x'|x, a)}{2}$$

holds for all $k = 0, 1, \ldots, J - 1$ and $(x, a, x') \in X_k \times A \times X_{k+1}$. Combining this with Eq. (26) then shows that $\left| P_0(x'|x, a) - \bar{P}_i(x'|x, a) \right| \leq \epsilon_i(x'|x, a)$, indicating that $P_0$ is indeed in $\mathcal{P}_i$ by the definition of $\mathcal{P}_i$. $\qquad\square$

The next lemma shows that the upper occupancy bound for each state-action pair is lower bounded.

**Lemma C.5.** *We have $\phi_t(x, a) \geq \frac{1}{T^3|X|^2|A|}$ for all $x \in X$ and $a \in A$.*

*Proof.* This is simply by the definition of $\phi_t$ in Eq. (23) and the definition of $\Omega$: $\phi_t(x, a) \geq \widehat{w}_t(x, a) \geq \frac{1}{T^3|X|^2|A|}$. $\qquad\square$

The last lemma is an improvement of [25, Lemma 4] and is important for bounding ERROR and BIAS-1 in terms of $\sqrt{L_T}$, as opposed to $\sqrt{T}$ (which is the case in [25]).

**Lemma C.6.** *With probability at least $1 - 6\delta$, for any $t$ and any collection of transition functions $\{P_t^x\}_{x \in X}$ such that $P_t^x \in \mathcal{P}_{i_t}$ for all $x$ (where $i_t$ is the index of the epoch to which episode $t$ belongs), we have*

$$\sum_{t=1}^{T} \sum_{x \in X, a \in A} \left| w^{P_t^x, \pi_t}(x, a) - w_t(x, a) \right| \ell_t(x, a)$$

$$= \tilde{\mathcal{O}}\left( |X|\sqrt{J|A|L_T \ln \frac{1}{\delta}} + |X|^5|A|^2 \ln \frac{1}{\delta} + |X|^4|A| \ln^2 \frac{1}{\delta} \right).$$

*Proof.* The proof is technical but mostly follows the same ideas of that for [25, Lemma 4]. We first assume that the events of Lemma C.2 and Lemma C.3 hold, which happens with probability at least $1 - 5\delta$. According to the proof of [25, Lemma 4] (specifically their Eq. (15)), we have for any pair $(x, a)$,

$$\left| w^{P_t^x, \pi_t}(x, a) - w_t(x, a) \right| \cdot \ell_t(x, a)$$

$$\leq \sum_{m=0}^{k(x)-1} \sum_{x_m, a_m, x_{m+1}} \epsilon_{i_t}^\star(x_{m+1}|x_m, a_m) w_t(x_m, a_m) w^{P_t^x, \pi_t}(x, a|x_{m+1}) \cdot \ell_t(x, a), \qquad (27)$$

where $\epsilon_{i_t}^\star(x'|x, a) = \mathcal{O}\left( \sqrt{\frac{P(x'|x, a) \ln\left(\frac{T|X||A|}{\delta}\right)}{\max\{1, N_{i_t}(x, a)\}}} + \frac{\ln\left(\frac{T|X||A|}{\delta}\right)}{\max\{1, N_{i_t}(x, a)\}} \right)$, and for an occupancy measure $w$, $w(x, a|x')$ denotes the probability of encountering the pair $(x, a)$ given that $x'$ was visited earlier, under policy $\pi^w$ and $P^w$. By their Eq. (16), we also have

$$|w^{P_t^x, \pi_t}(x, a|x_{m+1}) - w_t(x, a|x_{m+1})|$$

$$\leq \pi_t(a|x) \sum_{h=m+1}^{k(x)-1} \sum_{x_h', a_h', x_{h+1}'} \epsilon_{i_t}^\star(x_{h+1}'|x_h', a_h') w_t(x_h', a_h'|x_{m+1}), \qquad (28)$$

Combining Eq. (27) and Eq. (28), summing over all $t$ and $(x, a)$ and using shorthands $z_m \triangleq (x_m, a_m, x_{m+1})$ and $z_h' \triangleq (x_h', a_h', x_{h+1}')$, we have

$$\sum_{t=1}^{T} \sum_{x \in X, a \in A} |w^{P_t^x, \pi_t}(x, a) - w_t(x, a)| \cdot \ell_t(x, a)$$

$$\leq \sum_{t, x, a} \sum_{m=0}^{k(x)-1} \sum_{z_m} \epsilon_{i_t}^\star(x_{m+1}|x_m, a_m) w_t(x_m, a_m) w_t(x, a|x_{m+1}) \ell_t(x, a)$$

$$+ \sum_{t, x, a} \sum_{m=0}^{k(x)-1} \sum_{z_m} \epsilon_{i_t}^\star(x_{m+1}|x) w_t(x_m, a_m) \cdot \left( \pi_t(a|x) \sum_{h=m+1}^{k(x)-1} \sum_{z_h'} \epsilon_{i_t}^\star(x_{h+1}'|x_h', a_h') w_t(x_h', a_h'|x_{m+1}) \right)$$

$$(\ell_t(x, a) \leq 1)$$

$$= \sum_{t} \sum_{k < J} \sum_{m=0}^{k-1} \sum_{z_m} \epsilon_{i_t}^\star(x_{m+1}|x_m, a_m) w_t(x_m, a_m) \sum_{x \in X_k, a \in A} w_t(x, a|x_{m+1}) \ell_t(x, a)$$

$$+ \sum_{t} \sum_{0 \leq m < h < k < J} \sum_{z_m, z_h'} \epsilon_{i_t}^\star(x_{m+1}|x) w_t(x_m, a_m) \epsilon_{i_t}^\star(x_{h+1}'|x_h', a_h') w_t(x_h', a_h'|x_{m+1}) \cdot \left( \sum_{x \in X_k, a \in A} \pi_t(a|x) \right)$$

$$= \sum_{0 \leq m < k < J} \sum_{t, z_m} \epsilon_{i_t}^\star(x_{m+1}|x_m, a_m) w_t(x_m, a_m) \sum_{x \in X_k, a \in A} w_t(x, a|x_{m+1}) \ell_t(x, a)$$

$$+ \sum_{0 \leq m < h < k < J} |X_k| \sum_{t, z_m, z_h'} \epsilon_{i_t}^\star(x_{m+1}|x) w_t(x_m, a_m) \epsilon_{i_t}^\star(x_{h+1}'|x_h', a_h') w_t(x_h', a_h'|x_{m+1})$$

$$\leq \sum_{0 \leq m < k < J} \sum_{t, z_m} \epsilon_{i_t}^\star(x_{m+1}|x_m, a_m) w_t(x_m, a_m) \sum_{x \in X_k, a \in A} w_t(x, a|x_{m+1}) \ell_t(x, a)$$

$$+ |X| \sum_{0 \leq m < h < J} \sum_{t, z_m, z_h'} \epsilon_{i_t}^\star(x_{m+1}|x) w_t(x_m, a_m) \epsilon_{i_t}^\star(x_{h+1}'|x_h', a_h') w_t(x_h', a_h'|x_{m+1})$$

$$\triangleq B_1 + |X| B_2.$$

It remains to bound $B_1$ and $B_2$. First, $B_2$ is exactly the same as in the proof of [25, Lemma 4]. Below, we outline the proof with the dependence on all parameters explicit (indeed, this is hidden in their

proof). First, according to their analysis, $B_2$ is bounded by

$$\tilde{\mathcal{O}}\left(\sum_{0\leq m<h<J}\sum_{t,z_m,z'_h}\sqrt{\frac{P(x_{m+1}|x_m,a_m)\ln\frac{1}{\delta}}{\max\{1,N_{i_t}(x_m,a_m)\}}}w_t(x_m,a_m)\sqrt{\frac{P(x'_{h+1}|x'_h,a'_h)\ln\frac{1}{\delta}}{\max\{1,N_{i_t}(x'_h,a'_h)\}}}w_t(x'_h,a'_h|x_{m+1})\right)$$

$$+\tilde{\mathcal{O}}\left(\sum_{0\leq m<h<J}\sum_{t,z_m,z'_h}\frac{w_t(x_m,a_m)\ln\frac{1}{\delta}}{\max\{1,N_{i_t}(x_m,a_m)\}}\right)+\tilde{\mathcal{O}}\left(\sum_{0\leq m<h<J}\sum_{t,z_m,z'_h}\frac{w_t(x'_h,a'_h)\ln\frac{1}{\delta}}{\max\{1,N_{i_t}(x'_h,a'_h)\}}\right).$$

They show that the first term is bounded by $\tilde{\mathcal{O}}(|X|^2|A|\ln^2(1/\delta))$. For the second term, we have

$$\sum_{0\leq m<h<J}\sum_{t,z_m,z'_h}\frac{w_t(x_m,a_m)\ln\frac{1}{\delta}}{\max\{1,N_{i_t}(x_m,a_m)\}}$$

$$\leq\left(\sum_{h=0}^{J-1}|X_h|\cdot|A|\cdot|X_{h+1}|\ln\frac{1}{\delta}\right)\sum_{m=0}^{J-1}|X_{m+1}|\cdot\sum_{t,x\in X_m,a\in A}\frac{w_t(x_m,a_m)}{\max\{1,N_{i_t}(x_m,a_m)\}}$$

$$\leq\mathcal{O}\left(|X|^2|A|\ln\frac{1}{\delta}\right)\cdot\tilde{\mathcal{O}}\left(|X|^2|A|+|X|\ln\frac{1}{\delta}\right)\qquad\text{(Lemma C.3)}$$

$$\leq\tilde{\mathcal{O}}\left(|X|^4|A|^2\ln\frac{1}{\delta}+|X|^3|A|\ln\frac{1}{\delta}\right).$$

The third term can be bounded in the exact same way. Therefore, we arrive at

$$|X|B_2\leq\tilde{\mathcal{O}}\left(|X|^5|A|^2\ln(1/\delta)+|X|^4|A|\ln^2(1/\delta)\right).\qquad(29)$$

Next we show that $B_1$ is bounded by $\tilde{\mathcal{O}}(|X|\sqrt{J|A|L_T\ln(1/\delta)}+|X|^3|A|\ln(1/\delta))$. According to the definition of $\epsilon^\star_{i_t}$, we have

$$B_1=\mathcal{O}\left(\sum_{0\leq m<k<J}\sum_{t,z_m}w_t(x_m,a_m)\left(\sum_{x\in X_k,a\in A}w_t(x,a|x_{m+1})\ell_t(x,a)\right)\right.$$

$$\left.\cdot\sqrt{\frac{P(x_{m+1}|x_m,a_m)\ln\left(\frac{T|X||A|}{\delta}\right)}{\max\{1,N_{i_t}(x_m,a_m)\}}}\right)+\mathcal{O}\left(\sum_{0\leq m<k<J}\sum_{t,z_m}\frac{w_t(x_m,a_m)\ln\left(\frac{T|X||A|}{\delta}\right)}{\max\{1,N_{i_t}(x_m,a_m)\}}\right).$$

$$(30)$$

According to Lemma C.3, the second term is bounded as

$$\mathcal{O}\left(\sum_{0\leq m<k<J}\sum_{t,z_m}\frac{w_t(x_m,a_m)\ln\left(\frac{T|X||A|}{\delta}\right)}{\max\{1,N_{i_t}(x_m,a_m)\}}\right)\leq\tilde{\mathcal{O}}\left(J|X|^2|A|+J|X|\ln\frac{1}{\delta}\right).\qquad(31)$$

In the following, we define $\ell_t(k|x,a)\triangleq\sum_{x_k\in X_k,a_k\in A}\ell_t(x_k,a_k)w_t(x_k,a_k|x,a)$ where $w_t(x',a'|x,a)$ is the probability of encountering pair $(x',a')$ given that pair $(x,a)$ was encoun-

tered earlier, under policy $\pi_t$ and transition $P$. For the first term of Eq. (30), we then have

$$
\sum_{0 \leq m < k < J} \sum_{t, z_m} w_t(x_m, a_m) \left( \sum_{x \in X_k, a \in A} w_t(x, a | x_{m+1}) \ell_t(x, a) \right) \cdot \sqrt{\frac{P(x_{m+1} | x_m, a_m) \ln\left(\frac{T|X||A|}{\delta}\right)}{\max\{1, N_{i_t}(x_m, a_m)\}}}
$$

$$
\leq \sum_{0 \leq m < k < J} \sum_{t, z_m} w_t(x_m, a_m) \sqrt{\left( \sum_{x \in X_k, a \in A} w_t(x, a | x_{m+1}) \ell_t(x, a) \right) \cdot \frac{P(x_{m+1} | x_m, a_m) \ln\left(\frac{T|X||A|}{\delta}\right)}{\max\{1, N_{i_t}(x_m, a_m)\}}}
$$

$$
\leq \sum_{0 \leq m < k < J} \sum_{t, x_m, a_m} w_t(x_m, a_m) \sqrt{|X_{m+1}| \cdot \frac{\ell_t(k | x_m, a_m) \ln\left(\frac{T|X||A|}{\delta}\right)}{\max\{1, N_{i_t}(x_m, a_m)\}}}
$$

(Cauchy-Schwarz inequality)

$$
\leq \sum_{0 \leq m < k < J} \sqrt{|X_{m+1}| \ln\left(\frac{T|X||A|}{\delta}\right)}
$$

$$
\cdot \sum_{t, x_m, a_m} \left( \mathbb{1}_t(x_m, a_m) \sqrt{\frac{\ell_t(k | x_m, a_m)}{\max\{1, N_{i_t}(x_m, a_m)\}}} + \frac{w_t(x_m, a_m) - \mathbb{1}_t(x_m, a_m)}{\sqrt{\max\{1, N_{i_t}(x_m, a_m)\}}} \right). \tag{32}
$$

According to Lemma C.3 again, we have for all $m = 0, 1, \ldots, J - 1$,

$$
\sum_{t=1}^{T} \sum_{x_m, a_m} \frac{w_t(x_m, a_m) - \mathbb{1}_t(x_m, a_m)}{\sqrt{\max\{1, N_{i_t}(x_m, a_m)\}}} \leq \tilde{\mathcal{O}} \left( |X_m||A| + \ln(1/\delta) \right).
$$

For the term $\sum_{t, x_m, a_m} \mathbb{1}_t(x_m, a_m) \sqrt{\frac{\ell_t(k | x_m, a_m)}{\max\{1, N_{i_t}(x_m, a_m)\}}}$, using Cauchy-Schwarz inequality, we have

$$
\sum_{t, x_m, a_m} \mathbb{1}_t(x_m, a_m) \sqrt{\frac{\ell_t(k | x_m, a_m)}{\max\{1, N_{i_t}(x_m, a_m)\}}}
$$

$$
\leq \sum_{x_m, a_m} \sqrt{\sum_{t=1}^{T} \frac{\mathbb{1}_t(x_m, a_m)}{\max\{1, N_{i_t}(x_m, a_m)\}}} \cdot \sqrt{\sum_{t=1}^{T} \mathbb{1}_t(x_m, a_m) \ell_t(k | x_m, a_m)}
$$

(Cauchy-Schwarz inequality)

$$
\leq \mathcal{O} \left( \sqrt{|X_m||A| \left( \sum_{t=1}^{T} \sum_{x \in X_m, a \in A} \mathbb{1}_t(x_m, a_m) \ell_t(k | x, a) \right) \cdot \ln T} \right), \tag{33}
$$

where the last step uses Cauchy-Schwarz inequality again and the fact $\sum_{t=1}^{T} \frac{\mathbb{1}_t(x_m, a_m)}{\max\{1, N_{i_t}(x_m, a_m)\}} \leq \mathcal{O}(\ln T)$. Combining Eq. (32) and Eq. (33), we have

$$
\sum_{0 \leq m < k < J} \sum_{t, z_m} w_t(x_m, a_m) \left( \sum_{x \in X_k, a \in A} w_t(x, a | x_{m+1}) \ell_t(x, a) \right) \cdot \sqrt{\frac{P(x_{m+1} | x_m, a_m) \ln\left(\frac{T|X||A|}{\delta}\right)}{\max\{1, N_{i_t}(x_m, a_m)\}}}
$$

$$
\leq \tilde{\mathcal{O}} \left( \sum_{0 \leq m < k < J} \sqrt{|X_m||A||X_{m+1}| \ln\frac{1}{\delta}} \sqrt{\sum_{t=1}^{T} \sum_{x \in X_m, a \in A} \mathbb{1}_t(x_m, a_m) \ell_t(k | x, a)} \right)
$$

$$
\leq \tilde{\mathcal{O}} \left( \sum_{m=0}^{J-1} \sqrt{J|X_m||A||X_{m+1}| \sum_{t=1}^{T} \sum_{k > m} \sum_{x \in X_m, a \in A} \mathbb{1}_t(x_m, a_m) \ell_t(k | x, a) \ln\frac{1}{\delta}} \right).
$$

(Cauchy-Schwarz inequality)

Further note that

$$\mathbb{E}_t \left[ \sum_{k>m} \sum_{x \in X_m, a \in A} \mathbb{1}_t(x_m, a_m) \ell_t(k|x, a) \right]$$

$$= \sum_{k>m} \sum_{x \in X_m, a \in A} w_t(x, a) \ell_t(k|x, a)$$

$$= \sum_{x \in X_m, a \in A} w_t(x, a) \sum_{k>m} \sum_{x' \in X_k, a' \in A} w_t(x', a'|x, a) \ell_t(x', a')$$

$$= \sum_{k>m} \sum_{x' \in X_k, a' \in A} w_t(x', a') \ell_t(x', a')$$

$$\leq \langle w_t, \ell_t \rangle$$

and

$$\mathbb{E}_t \left[ \left( \sum_{k>m} \sum_{x \in X_m, a \in A} \mathbb{1}_t(x_m, a_m) \ell_t(k|x, a) \right)^2 \right] \leq J \langle w_t, \ell_t \rangle .$$

Using Freedman inequality $J$ times with parameter $\delta/J$ for $m = 0, 1, \ldots, J-1$ and taking a union bound, we have with probability $1 - \delta$, for all $m = 0, 1, \ldots, J-1$,

$$\sum_{t=1}^{T} \sum_{k>m} \sum_{x \in X_m, a \in A} \mathbb{1}_t(x_m, a_m) \ell_t(k|x, a) - \sum_{t=1}^{T} \langle w_t, \ell_t \rangle$$

$$\leq \tilde{\mathcal{O}} \left( \sqrt{J \sum_{t=1}^{T} \langle w_t, \ell_t \rangle \ln \frac{1}{\delta}} + J \ln \frac{1}{\delta} \right) = \tilde{\mathcal{O}} \left( \sqrt{J L_T \ln \frac{1}{\delta}} + J \ln \frac{1}{\delta} \right).$$

Therefore, using AM-GM inequality, we have

$$\sum_{t=1}^{T} \sum_{k>m} \sum_{x \in X_m, a \in A} \mathbb{1}_t(x_m, a_m) \ell_t(k|x, a) \leq \tilde{\mathcal{O}} \left( L_T + J \ln \frac{1}{\delta} \right).$$

Combining the results above and Eq. (31), we know that with probability at least $1 - \delta$,

$$B_1 \leq \tilde{\mathcal{O}} \left( |X| \sqrt{J|A| L_T \ln \frac{1}{\delta}} + J|X| \sqrt{|A|} \ln \frac{1}{\delta} \right) + \tilde{\mathcal{O}} \left( J|X|^2 |A| + J|X| \ln \frac{1}{\delta} \right)$$

$$\leq \tilde{\mathcal{O}} \left( |X| \sqrt{J|A| L_T \ln \frac{1}{\delta}} + |X|^3 |A| \ln \frac{1}{\delta} \right). \tag{34}$$

Finally, combining Eq. (29) and Eq. (34) and considering the probability of the events of Lemma C.2 and Lemma C.3, we have with probability $1 - 6\delta$,

$$B_1 + |X| B_2 \leq \tilde{\mathcal{O}} \left( |X| \sqrt{J|A| L_T \ln \frac{1}{\delta}} + |X|^5 |A|^2 \ln \frac{1}{\delta} + |X|^4 |A| \ln^2 \frac{1}{\delta} \right),$$

finishing the proof. $\qquad \square$

### C.3.2 Bounding ERROR

**Lemma C.7.** *With probability at least $1 - 6\delta$, we have*

$$\text{ERROR} = \sum_{t=1}^{T} \langle w_t - \widehat{w}_t, \ell_t \rangle = \tilde{\mathcal{O}} \left( |X| \sqrt{J|A| L_T \ln \frac{1}{\delta}} + |X|^5 |A|^2 \ln \frac{1}{\delta} + |X|^4 |A| \ln^2 \frac{1}{\delta} \right),$$

*Proof.* Note that according to the definition of $\widehat{w}_t$, the transition function $P^{\widehat{w}_t}$ induced by $\widehat{w}_t$ is in $\mathcal{P}_{i_t}$. Therefore, applying Lemma C.6, we know that with probability at least $1 - 6\delta$,

$$
\begin{aligned}
\text{ERROR} &= \sum_{t=1}^{T} \langle \widehat{w}_t - w_t, \ell_t \rangle \\
&\leq \sum_{t=1}^{T} \sum_{x \in X, a \in A} |\widehat{w}_t(x,a) - w_t(x,a)| \, \ell_t(x,a) \\
&\leq \tilde{\mathcal{O}} \left( |X| \sqrt{J|A|L_T \ln \frac{1}{\delta}} + |X|^5 |A|^2 \ln \frac{1}{\delta} + |X|^4 |A| \ln^2 \frac{1}{\delta} \right),
\end{aligned}
$$

completing the proof. $\qquad\square$

### C.3.3 Bounding BIAS-1

**Lemma C.8.** *With probability at least $1 - 7\delta$, we have*

$$
\text{BIAS-1} = \sum_{t=1}^{T} \left\langle \widehat{w}_t, \ell_t - \widehat{\ell}_t \right\rangle \leq \tilde{\mathcal{O}} \left( |X| \sqrt{J|A|L_T \ln \frac{1}{\delta}} + |X|^5 |A|^2 \ln \frac{1}{\delta} + |X|^4 |A| \ln^2 \frac{1}{\delta} \right).
$$

*Proof.* First we write

$$
\sum_{t=1}^{T} \left\langle \widehat{w}_t, \ell_t - \widehat{\ell}_t \right\rangle = \sum_{t=1}^{T} \left\langle \widehat{w}_t, \mathbb{E}_t \left[ \widehat{\ell}_t \right] - \widehat{\ell}_t \right\rangle + \sum_{t=1}^{T} \left\langle \widehat{w}_t, \ell_t - \mathbb{E}_t \left[ \widehat{\ell}_t \right] \right\rangle.
$$

Since $\widehat{w}_t(x,a) \leq \phi_t(x,a)$ by the definition of $\phi_t$, we have

$$
\left\langle \widehat{w}_t, \widehat{\ell}_t \right\rangle \leq \sum_{k=1}^{J} \sum_{x \in X_k, a \in A} \frac{\widehat{w}_t(x,a)}{\phi_t(x,a)} \cdot \mathbb{1}_t(x,a) \leq J,
$$

$$
\mathbb{E}_t \left[ \left\langle \widehat{w}_t, \widehat{\ell}_t \right\rangle^2 \right] \leq \mathbb{E}_t \left[ J \cdot \left\langle \widehat{w}_t, \widehat{\ell}_t \right\rangle \right] = J \sum_{x,a} \widehat{w}_t(x,a) \cdot \frac{\ell_t(x,a)}{\phi_t(x,a)} \cdot w_t(x,a) \leq J \cdot \langle w_t, \ell_t \rangle,
$$

and thus according to Freedman inequality, we have with probability at least $1 - \delta$,

$$
\sum_{t=1}^{T} \left\langle \widehat{w}_t, \mathbb{E}_t \left[ \widehat{\ell}_t \right] - \widehat{\ell}_t \right\rangle \leq \mathcal{O} \left( \sqrt{J \sum_{t=1}^{T} \langle w_t, \ell_t \rangle \ln \frac{1}{\delta}} + J \cdot \ln \frac{1}{\delta} \right) = \mathcal{O} \left( \sqrt{J L_T \ln \frac{1}{\delta}} + |X| \ln \frac{1}{\delta} \right).
\tag{35}
$$

For the second term, we have

$$
\sum_{t=1}^{T} \left\langle \widehat{w}_t, \ell_t - \mathbb{E}_t \left[ \widehat{\ell}_t \right] \right\rangle = \sum_{t,x,a} \widehat{w}_t(x,a) \ell_t(x,a) \cdot \left( 1 - \frac{w_t(x,a)}{\phi_t(x,a)} \right) \leq \sum_{t,x,a} |\phi_t(x,a) - w_t(x,a)| \cdot \ell_t(x,a).
$$

By the definition of $\phi_t$, one has $\phi_t = w^{P_t^x, \pi_t}$ for $P_t^x = \text{argmax}_{\hat{P} \in \mathcal{P}_{i_t}} \sum_a w^{\hat{P}, \pi_t}(x,a)$. Therefore, according to Lemma C.7, we have with probability at least $1 - 6\delta$,

$$
\sum_{t=1}^{T} \left\langle \widehat{w}_t, \ell_t - \mathbb{E}_t \left[ \widehat{\ell}_t \right] \right\rangle \leq \tilde{\mathcal{O}} \left( |X| \sqrt{J|A|L_T \ln \frac{1}{\delta}} + |X|^5 |A|^2 \ln \frac{1}{\delta} + |X|^4 |A| \ln^2 \frac{1}{\delta} \right).
\tag{36}
$$

Combining Eq. (35) and Eq. (36) proves the lemma.

$\qquad\square$

### C.3.4 Bounding BIAS-2

**Lemma C.9.** *With probability at least* $1 - 5\delta$, *we have*

$$\text{BIAS-2} = \sum_{t=1}^{T} \left\langle u, \widehat{\ell}_t - \ell_t \right\rangle \leq C \sum_{x \in X, a \in A} u(x,a) \sqrt{8\rho_T(x,a) \sum_{t=1}^{T} \ell_t(x,a) \ln \frac{C|X||A|}{\delta}}$$
$$+ 2C \left\langle u, \rho_T \right\rangle \ln \frac{C|X||A|}{\delta},$$

*for some constant* $C = \tilde{\mathcal{O}}(1)$.

*Proof.* First we write

$$\sum_{t=1}^{T} \left\langle u, \widehat{\ell}_t - \ell_t \right\rangle = \sum_{t=1}^{T} \left\langle u, \mathbb{E}_t \left[ \widehat{\ell}_t \right] - \ell_t \right\rangle + \sum_{t=1}^{T} \left\langle u, \widehat{\ell}_t - \mathbb{E}_t \left[ \widehat{\ell}_t \right] \right\rangle.$$

The first term is nonpositive under the event of Lemma C.2 as for any $(x,a) \in X \times A$, $w_t(x,a) \leq \phi_t(x,a)$ by the definition of $\phi_t$ and thus

$$\mathbb{E}_t \left[ \widehat{\ell}_t(x,a) \right] - \ell_t(x,a) = w_t(x,a) \cdot \frac{\ell_t(x,a)}{\phi_t(x,a)} - \ell_t(x,a) \leq 0. \tag{37}$$

For the second term, note that for each $(x,a) \in X \times A$, we have

$$\widehat{\ell}_t(x,a) = \frac{\ell_t(x,a)}{\phi_t(x,a)} \cdot \mathbb{1}_t(x,a) \leq T^3 |X|^2 |A|, \qquad \text{(Lemma C.5)}$$

$$\widehat{\ell}_t(x,a) = \frac{\ell_t(x,a)}{\phi_t(x,a)} \cdot \mathbb{1}_t(x,a) \leq \rho_t(x,a),$$

and

$$\sum_{t=1}^{T} \mathbb{E}_t \left[ \widehat{\ell}_t(x,a)^2 \right] \leq \sum_{t=1}^{T} \mathbb{E}_t \left[ \frac{\ell_t(x,a)}{\phi_t(x,a)^2} \cdot \mathbb{1}_t(x,a) \right] \leq \rho_T(x,a) \sum_{t=1}^{T} \ell_t(x,a).$$

Therefore, using Theorem 2.2 with $X_t = \widehat{\ell}_t(x,a) - \mathbb{E}_t \left[ \widehat{\ell}_t(x,a) \right]$, $B_t = \rho_t(x,a)$, $B^\star = \rho_T(x,a)$, $b = T^3 |X|^2 |A|$, $C = \lceil \log_2 b \rceil \lceil \log_2 b^2 T \rceil = \tilde{\mathcal{O}}(1)$, we have with probability at least $1 - \frac{\delta}{|X||A|}$,

$$\sum_{t=1}^{T} \widehat{\ell}_t(x,a) - \mathbb{E}_t \left[ \widehat{\ell}_t(x,a) \right] \leq C \left( \sqrt{8\rho_T(x,a) \sum_{t=1}^{T} \ell_t(x,a) \ln \frac{C|X||A|}{\delta}} + 2\rho_T(x,a) \ln \frac{C|X||A|}{\delta} \right).$$

Taking a union bound over all $(x,a) \in X \times A$, multiplying both sides by $u(x,a)$, and summing up all these inequalities, we have with probability at least $1 - \delta$,

$$\sum_{t=1}^{T} \left\langle u, \widehat{\ell}_t - \mathbb{E}_t \left[ \widehat{\ell}_t \right] \right\rangle$$

$$\leq C \sum_{x \in X, a \in A} u(x,a) \left( \sqrt{8\rho_T(x,a) \sum_{t=1}^{T} \ell_t(x,a) \ln \frac{C|X||A|}{\delta}} + 2\rho_T(x,a) \ln \frac{C|X||A|}{\delta} \right). \tag{38}$$

Combining Eq. (37) and Eq. (38) finishes the proof.

$\square$

### C.3.5 Bounding REG-TERM

**Lemma C.10.** *With probability at least $1 - 4\delta$, we have*

$$\text{REG-TERM} = \sum_{t=1}^{T} \left\langle \widehat{w}_t - u, \widehat{\ell}_t \right\rangle \leq \tilde{\mathcal{O}}\left( \frac{|X|^2 |A|}{\eta} \right) + 5\eta \overline{L}_T - \frac{\langle u, \rho_T \rangle}{70\eta \ln T},$$

*where $\overline{L}_T = \sum_{t=1}^{T} \sum_{x \in X, a \in A} \mathbb{1}_t(x, a) \ell_t(x, a)$.*

*Proof.* We condition on the event of Lemma C.2. First, we prove that $u \in \Delta(\mathcal{P}_i) \cap \Omega$ for all $i$ (recall its definition in Eq. (24)). Indeed, for any fixed $(x, a, x') \in X_k \times A \times X_{k+1}$, $k = 0, 1, \ldots, J-1$, we have (with $w^{P_0, \pi_a}(x)$ being the probability of visiting $x$ under $P_0$ and $\pi_a$)

$$u(x, a, x') \geq \frac{1}{T|A|} w^{P_0, \pi_a}(x, a, x')$$

$$= \frac{1}{T|A|} w^{P_0, \pi_a}(x) P_0(x'|x, a)$$

$$\geq \frac{1}{T|A|} \left( \sum_{x'' \in X_{k(x)-1}} w^{P_0, \pi_a}(x'') \cdot P_0(x|x'', a) \right) \cdot \frac{1}{T|X|} \qquad \text{(Lemma C.4)}$$

$$\geq \frac{1}{T^3 |X|^2 |A|} \left( \sum_{x'' \in X_{k(x)-1}} w^{P_0, \pi_a}(x'') \right), \qquad \text{(Lemma C.4 again)}$$

$$= \frac{1}{T^3 |X|^2 |A|},$$

which shows $u \in \Omega$. On the other hand, since $P \in \mathcal{P}_i$ under Lemma C.2 and $P_0 \in \mathcal{P}_i$ as well by Lemma C.4, we have $u^\star \in \Delta(\mathcal{P}_i)$ and $w^{P_0, \pi_a} \in \Delta(\mathcal{P}_i)$, which indicates that, as a convex combination of $u^\star$ and $w^{P_0, \pi_a}$ for all $a$, $u$ has to be in $\Delta(\mathcal{P}_i)$ as well.

Therefore, by standard OMD analysis (e.g., [5, Lemma 12]), we have

$$\left\langle \widehat{w}_t - u, \widehat{\ell}_t \right\rangle$$

$$\leq D_{\psi_t}(u, \widehat{w}_t) - D_{\psi_t}(u, \widehat{w}_{t+1}) + \sum_{k=0}^{J-1} \sum_{(x,a,x') \in X_k \times A \times X_{k+1}} \eta_t(x, a) \widehat{w}_t^2(x, a, x') \widehat{\ell}_t^2(x, a)$$

$$\leq D_{\psi_t}(u, \widehat{w}_t) - D_{\psi_t}(u, \widehat{w}_{t+1}) + \sum_{k=0}^{J-1} \sum_{(x,a) \in X_k \times A} \eta_t(x, a) \widehat{w}_t^2(x, a) \widehat{\ell}_t^2(x, a)$$

$$\qquad\qquad\qquad\qquad\qquad\qquad (\textstyle\sum_{x' \in X_{k+1}} \widehat{w}_t(x, a, x')^2 \leq \widehat{w}_t(x, a)^2)$$

$$\leq D_{\psi_t}(u, \widehat{w}_t) - D_{\psi_t}(u, \widehat{w}_{t+1}) + \sum_{x \in X, a \in A} \eta_t(x, a) \mathbb{1}_t(x, a) \ell_t(x, a). \qquad (\widehat{w}_t(x, a) \leq \phi_t(x, a))$$

Summing over $t$ gives

$$\sum_{t=1}^{T} \left\langle \widehat{w}_t - u, \widehat{\ell}_t \right\rangle \leq D_{\psi_1}(u, \widehat{w}_1) + \sum_{t=1}^{T-1} D_{\psi_{t+1}}(u, \widehat{w}_{t+1}) - D_{\psi_t}(u, \widehat{w}_{t+1})$$

$$+ \sum_{t=1}^{T} \sum_{x \in X, a \in A} \eta_t(x, a) \mathbb{1}_t(x, a) \ell_t(x, a). \qquad (39)$$

Next, for a fixed $(x, a)$ pair, let $n(x, a)$ be the total number of times the learning rate for $(x, a)$ has increased, such that $\eta_T(x, a) = \eta \kappa^{n(x,a)}$, and let $t_1, \ldots, t_{n(x,a)}$ be the rounds where $\eta_t(x, a)$ is increased, such that $\eta_{t_i+1}(x, a) = \eta_{t_i}(x, a) \kappa$. Then since $\frac{1}{\phi_{t_{n(x,a)}+1}(x, a)} > \rho_{t_{n(x,a)}}(x, a) >$

$2\rho_{t_{n(x,a)-1}}(x,a) > \cdots > 2^{n(x,a)-1}\rho_1(x,a) > 2^{n(x,a)}|A|$ and $\frac{1}{\phi_{t_{n(x,a)+1}}(x,a)} \leq T^3|X|^2|A|$ (Lemma C.5), we have $n \leq \log_2\left(T^3|X|^2\right) \leq 7\log_2 T$.

Therefore, we have $\eta_t(x,a) \leq \eta e^{\frac{7\log_2 T}{7\ln T}} \leq 5\eta$ for any $t$, $x \in X$, and $a \in A$, and the last term in Eq. (39) is thus bounded by $5\eta \overline{L}_T$. For the second term, with $h(y) = y - 1 - \ln y$, we have

$$\sum_{t=1}^{T-1} D_{\psi_{t+1}}(u,\widehat{w}_{t+1}) - D_{\psi_t}(u,\widehat{w}_{t+1})$$

$$\leq \sum_{t=1}^{T-1}\sum_{k=0}^{J-1}\sum_{x\in X_k,a\in A,x'\in X_{k+1}} \left(\frac{1}{\eta_{t+1}(x,a)} - \frac{1}{\eta_t(x,a)}\right) h\left(\frac{u(x,a,x')}{\widehat{w}_{t+1}(x,a,x')}\right)$$

$$\leq \sum_{k=0}^{J-1}\sum_{x\in X_k,a\in A,x'\in X_{k+1}} \frac{1-\kappa}{\eta \cdot \kappa^{n(x,a)}} \cdot h\left(\frac{u(x,a,x')}{\widehat{w}_{t_{n(x,a)+1}}(x,a,x')}\right)$$

$$\leq -\frac{1}{35\eta\ln T}\sum_{k=0}^{J-1}\sum_{x\in X_k,a\in A,x'\in X_{k+1}} h\left(\frac{u(x,a,x')}{\widehat{w}_{t_{n(x,a)+1}}(x,a,x')}\right)$$

$$(1-\kappa \leq -\frac{1}{7\ln T} \text{ and } \kappa^{n(x,a)} \leq e^{\frac{7\log_2 T}{7\ln T}} \leq 5)$$

$$= -\frac{1}{35\eta\ln T}\sum_{k=0}^{J-1}\sum_{x\in X_k,a\in A,x'\in X_{k+1}} \left(\frac{u(x,a,x')}{\widehat{w}_{t_{n(x,a)+1}}(x,a,x')} - 1 - \ln\frac{u(x,a,x')}{\widehat{w}_{t_{n(x,a)+1}}(x,a,x')}\right)$$

$$\leq \frac{|X|^2|A|(1+6\ln T)}{35\eta\ln T} - \frac{1}{35\eta\ln T}\sum_{k=0}^{J-1}\sum_{x\in X_k,a\in A,x'\in X_{k+1}} \frac{u(x,a,x')}{\widehat{w}_{t_{n(x,a)+1}}(x,a,x')}$$

$$(\ln\frac{u(x,a,x')}{\widehat{w}_{t_{n(x,a)+1}}(x,a,x')} \leq 6\ln T)$$

$$\leq \frac{|X|^2|A|}{5\eta} - \frac{1}{35\eta\ln T}\sum_{k=0}^{J-1}\sum_{x\in X_k,a\in A,x'\in X_{k+1}} \frac{u(x,a,x')}{\phi_{t_{n(x,a)+1}}(x,a)}$$

$$= \frac{|X|^2|A|}{5\eta} - \frac{1}{35\eta\ln T}\sum_{k=0}^{J-1}\sum_{x\in X_k,a\in A} \frac{u(x,a)}{\phi_{t_{n(x,a)+1}}(x,a)}$$

$$= \frac{|X|^2|A|}{5\eta} - \frac{\langle u,\rho_T\rangle}{70\eta\ln T}. \qquad (\rho_T(x,a) = \frac{2}{\phi_{t_{n(x,a)+1}}(x,a)})$$

Finally, we bound the first term in Eq. (39):

$$D_{\psi_1}(u,\widehat{w}_1) = \frac{1}{\eta}\left(\sum_{k=0}^{J-1}\sum_{(x,a,x')\in X_k\times A\times X_{k+1}} h\left(\frac{u(x,a,x')}{\widehat{w}_1(x,a,x')}\right)\right)$$

$$= \frac{1}{\eta}\left(\sum_{k=0}^{J-1}\sum_{(x,a,x')\in X_k\times A\times X_{k+1}} h\left(|X_k|\cdot|A|\cdot|X_{k+1}|\cdot u(x,a,x')\right)\right)$$

$$= \frac{1}{\eta}\left(\sum_{k=0}^{J-1}\sum_{(x,a,x')\in X_k\times A\times X_{k+1}} \ln\left(\frac{1}{|X_k|\cdot|A|\cdot|X_{k+1}|\cdot u(x,a,x')}\right)\right)$$

$$\leq \tilde{\mathcal{O}}\left(\frac{|X|^2|A|}{\eta}\right).$$

Combining all the bounds finishes the proof. $\qquad\square$

### C.3.6 Putting everything together

Now we are ready to prove Theorem 4.1. For completeness, we restate the theorem below.

**Theorem C.11.** *Algorithm 4 with a suitable choice of $\eta$ ensures that with probability at least $1 - \delta$,* $\mathrm{Reg} = \tilde{\mathcal{O}}\left(|X|\sqrt{J|A|L^\star \ln \frac{1}{\delta}} + |X|^5|A|^2 \ln^2 \frac{1}{\delta}\right).$

*Proof.* First, note that

$$\mathbb{E}_t\left[\sum_{k=0}^{J-1}\sum_{x \in X_k, a \in A} \mathbb{1}_t(x,a) \cdot \ell_t(x,a)\right] = \langle w_t, \ell_t\rangle \le J,$$

$$\mathbb{E}_t\left[\left(\sum_{k=0}^{J-1}\sum_{x \in X_k, a \in A} \mathbb{1}_t(x,a) \cdot \ell_t(x,a)\right)^2\right] \le J \cdot \langle w_t, \ell_t\rangle.$$

Therefore, using Freedman's inequality, we have with probability at least $1 - \delta$

$$\overline{L}_T - L_T \le 2\sqrt{J L_T \ln \frac{1}{\delta}} + J \ln \frac{1}{\delta},$$

where $\overline{L}_T$ is defined in Lemma C.10. Furthermore, using AM-GM inequality, we have with probability at least $1 - \delta$,

$$\overline{L}_T \le 2L_T + 2J \ln \frac{1}{\delta}. \tag{40}$$

Choosing $\eta \le \frac{1}{280C \ln(C|X||A|/\delta) \ln T}$, combining Lemma C.7, Lemma C.8, Lemma C.9 and Lemma C.10 and letting $L_u \triangleq \sum_{t=1}^T \langle u, \ell_t\rangle$, we have with probability at least $1 - 22\delta$:

$L_T - L^\star$

$$\le \tilde{\mathcal{O}}\left(|X|\sqrt{J|A|L_T \ln \frac{1}{\delta}} + \frac{|X|^2|A|}{\eta}\right) + 5\eta\overline{L}_T + \underbrace{\left(2C\langle u, \rho_T\rangle \ln \frac{C|X||A|}{\delta} - \frac{\langle u, \rho_T\rangle}{140\eta \ln T}\right)}_{\text{TERM1}}$$

$$+ \sum_{x \in X, a \in A} u(x,a)\underbrace{\left(C\sqrt{8\rho_T(x,a)\sum_{t=1}^T \ell_t(x,a)\ln\frac{C|X||A|}{\delta}} - \frac{\rho_T(x,a)}{140\eta \ln T}\right)}_{\text{TERM2}}$$

$$+ \tilde{\mathcal{O}}\left(|X|^5|A|^2 \ln \frac{1}{\delta} + |X|^4|A|\ln^2 \frac{1}{\delta}\right)$$

$$\le \tilde{\mathcal{O}}\left(|X|\sqrt{J|A|L_T \ln \frac{1}{\delta}} + \frac{|X|^2|A|}{\eta} + \eta L_u \ln \frac{1}{\delta} + |X|^5|A|^2 \ln \frac{1}{\delta} + |X|^4|A|\ln^2 \frac{1}{\delta}\right) + 10\eta L_T$$

<div align="center">(TERM1 is nonpositive, AM-GM inequality for TERM2, and Eq. (40))</div>

$$\le \tilde{\mathcal{O}}\left(|X|\sqrt{J|A|L_T \ln \frac{1}{\delta}} + \frac{|X|^2|A|}{\eta} + \eta L^\star \ln \frac{1}{\delta} + |X|^5|A|^2 \ln \frac{1}{\delta} + |X|^4|A|\ln^2 \frac{1}{\delta}\right) + 10\eta L_T.$$

<div align="right">(Eq. (25))</div>

As $\eta \le \frac{1}{280C \ln(C|X||A|/\delta) \ln T} \le \frac{1}{20}$, rearranging the terms gives

$$L_T - L^\star \le \tilde{\mathcal{O}}\left(|X|\sqrt{J|A|L_T \ln \frac{1}{\delta}} + \frac{|X|^2|A|}{\eta} + \eta L^\star \ln \frac{1}{\delta} + |X|^5|A|^2 \ln \frac{1}{\delta} + |X|^4|A|\ln^2 \frac{1}{\delta}\right).$$

Finally, choosing $\eta = \min\left\{\sqrt{\frac{|X|^2|A|}{L^\star \ln \frac{1}{\delta}}}, \frac{1}{280C \ln(C|X||A|/\delta) \ln T}\right\}$, $\delta = \delta'/22$, and solving the quadratic inequality, we have with probability at least $1 - \delta'$,

$$L_T - L^\star \le \tilde{\mathcal{O}}\left(|X|\sqrt{J|A|L^\star \ln \frac{1}{\delta'}} + |X|^5|A|^2 \ln \frac{1}{\delta'} + |X|^4|A|\ln^2 \frac{1}{\delta'}\right),$$

finishing the proof. $\qquad\square$

**Remark 3.** Similarly to the MAB case, the proof above requires tuning the initial learning rate $\eta$ in terms of the unknown quantity $L^\star$, and again, using standard doubling trick can remove this restriction, as pointed out in Remark 1.

## C.4 Issues of other potential approaches

In this section, we discuss why the idea of clipping [7] or implicit exploration [31] may not be directly applicable to achieve near-optimal high-probability small-loss bounds.

**Implicit exploration.** First, we consider the idea of implicit exploration. As mentioned in Appendix C.2, this means using the following loss estimator: $\widehat{\ell}_t = \frac{\ell_t(x,a)}{\phi_t(x,a)+\gamma} \cdot \mathbb{1}_t\{x,a\}$ for all $x \in X$ and $a \in A$ and some parameter $\gamma > 0$, and without using our increasing learning schedule. The concentration results of [25, Lemma 12] show that the deviation contains a term of order $1/\gamma$, meaning that $\gamma$ cannot be too small.

Repeating the same analysis, one can see that the main difficulty of obtaining high-probability small-loss bounds in this case is to bound BIAS-2 by the loss of the algorithm $L_T = \sum_{t=1}^T \langle w_t, \ell_t \rangle$ or $L^\star$, instead of the number of episodes $T$. Indeed, consider the term $\sum_{t=1}^T \left\langle \widehat{w}_t, \ell_t - \mathbb{E}_t\left[\widehat{\ell}_t\right] \right\rangle$:

$$
\sum_{t=1}^T \left\langle \widehat{w}_t, \ell_t - \mathbb{E}_t\left[\widehat{\ell}_t\right] \right\rangle = \sum_{t=1}^T \sum_{x \in X, a \in A} \widehat{w}_t(x,a)\ell_t(x,a) \cdot \left(1 - \frac{w_t(x,a)}{\phi_t(x,a)+\gamma}\right)
$$

$$
\leq \sum_{t=1}^T \sum_{x \in X, a \in A} |\phi_t(x,a) - w_t(x,a)|\ell_t(x,a) + \frac{\gamma}{\gamma + \phi_t(x,a)} \cdot \widehat{w}_t(x,a)\ell_t(x,a).
$$

The first term can still be bounded by $\mathcal{O}\left(|X|\sqrt{J|A|L_T} + |X|^5|A|^2 \ln\frac{1}{\delta} + |X|^4|A|\ln^2\frac{1}{\delta}\right)$ according to Lemma C.8. For the second term, while it is at most $\gamma \sum_{t=1}^T \sum_{x,a} \ell_t(x,a) \leq \gamma|X||A|T$, it is not clear at all how to bound it in terms of $L_T$ or $L^\star$. For MAB (where there is only one state $x_0$), it is possible to show that $\sum_{t=1}^T \ell_t(x_0,a) \leq \sum_{t=1}^T \ell_t(x_0,a^\star) + \tilde{\mathcal{O}}(\frac{1}{\eta} + \frac{1}{\gamma})$ for all $a \neq a^\star$ where $a^\star$ is the best action, making it possible to connect the second term with $L^\star$. However, we do not see a way of doing similar analysis for general MDPs.

**Clipping.** On the other hand, the idea of clipping for MAB is to clip all small probabilities so that actions with probability smaller than $\gamma$ are never selected. Even from an algorithmic perspective, it is not clear how to generalize this idea to MDPs, because it is possible that for a state $x$, $\widehat{w}_t(x,a)$ is smaller than $\gamma$ for *all* $a$. In this case, the clipping idea suggests not to "pick" $(x,a)$ at all for any $a$, but there is no way to ensure that if the transition function is such that $x$ can always be visited with some positive probability regardless of the policy we execute.

Moreover, even if there is a way to fix this, the analysis of clipping for MAB is also similar to the idea of implicit exploration in terms of obtaining small-loss bounds of order $\tilde{\mathcal{O}}(\sqrt{L^\star})$, and as we argued already, even for implicit exploration there are difficulties in generalizing the analysis to MDPs.