[Reviews · NeurIPS 2020]

Review 1

Summary and Contributions: This paper studies regret minimization in adversarial linear bandits against an adaptive adversary, and presents a new approach to deriving data-dependent (a.k.a. small-loss) regret bounds that hold with high probability (as opposed to the other well-studied type of expected regret). In contrast to existing approaches to deriving high-probability regret bounds in the considered setup where a biased estimator is used, the paper presents a OMD-type method involving an unbiased estimator of losses. The OMD method here uses a logarithmic self-concordant barrier as its regularizer. Furthermore the presented method uses a carefully chosen sequence of increasing learning rates to derive the desired regret bounds. More specifically, the main algorithmic contribution is a variant of SCRIBLE [3], equipped with a new sampling scheme inspired by the aforementioned idea. The modified algorithm is claimed to be the first general and efficient linear bandit algorithm enjoying a high-probability regret guarantee against an adaptive adversary – The original SCRIBLE [3] is developed for the expected regret against an oblivious adversary. The modified algorithm achieves high-probability and data-dependent regret bounds. Finally, the authors present an application of their technique to learning in adversarial MDPs establishing data-dependent high-probability regret bounds against adaptive adversaries.

Strengths: The paper studies a well-established but challenging problem in adversarial bandits. While similar approaches have been around for many years (e.g., Abernethy and Rakhlin [2009] and Neu [2015]), all seem to fail at providing a unified technique yielding data-dependent high-probability regret bounds against adaptive adversaries. Also some led to inefficient implementations (e.g., in linear bandits). Building on some ideas in recent papers, the paper provides a generic method of the aforementioned desideratum, which happen to be efficiently implementable. Overall, while some of the used ideas have been already presented in recent papers (such as using a increasing sequence of learning rates and using logarithmic self-concordant barrier in OMD), I believe that orchestrating these and deriving the desired bounds pose additional challenge and require in turn some necessary technical tools, as used in the present regret analyses. Overall, this is a well-executed paper, presenting solid results, and I recommend acceptance.

Weaknesses: I have a few minor comments: - While the main algorithmic contribution (Algorithm 2) enjoys strong regret guarantees, the fairly large constants involved in the choice of \kappa and \Psi there introduces some concerns about the empirical performance. As a consequence, the multiplicative numerical constants in the final regret bounds turn out to be fairly large. Could you explain whether this can be improved or not, and whether the presented algorithm would be an empirically viable choice in applications (I understand that this is theoretical paper, however I would like to hear the authors’ opinions on this). - I may disagree with the statement in Line 331: “[25] discovered the deep connection …”. To the best of my knowledge, such a connection has been pointed out in previous relevant studies prior to [25]. A powerful proxy to establish such a connection is the notion of “occupancy measure” presented in a series of prior work (though in slightly different setups), e.g., (Zimin and Neu, NIPS 2013) and later on in (Rosenberg and Mansour, ICML 2019). Please clarify, and revise accordingly -- At the very least, provide appropriate references for the first part of your statement “Based on several prior work, ...”

Correctness: The presented generic method to achieve high-probability regret bounds and the deign of presented algorithms make perfect sense to me. Although I was unable to check all the proofs, the results appear correct to me.

Clarity: The paper is overall very well-written and well-organized. In particular, the results are presented clearly and precisely.

Relation to Prior Work: The authors provide a satisfactory literature review (see also the comment below). In particular, they clearly and precisely explain the potential gains obtained by their method when applied to various online learning problems. While the authors have cited most papers relevant to linear bandits, some pointers to recent related papers on adversarial MDPs seem to be missing. In particular, one relevant recent work (and perhaps the first studying “a variant” of this problem under bandit feedback) is: Rosenberg and Mansour, Online Stochastic Shortest Path with Bandit Feedback and Unknown Transition Function, NeurIPS 2019.

Reproducibility: Yes

Additional Feedback: The paper is very well-polished. However, I found a few typos reported below: l. 67: a … self-concordant barriers ==> … barrier l. 89: has been proven ==> has proven l. 91: any … barriers ==> any … barrier l. 122: as a mean ==> as a means l. 145: to realize that arms that were ... ==> I think you meant “to realize those arms that were ...” l. 168: the Freedman’s inequality ==> Freedman’s inequality l. 224: from the boundary ==> to the boundary l. 237: notation ==> the notation l. 250: is orthogonal H_t…. ==> is orthogonal to … l. 31: Please provide some relevant references. ==== AFTER REBUTTAL ==== Thanks for your responses. I have read the other reviews and the rebuttal, and will keep my current score.


Review 2

Summary and Contributions: The paper derives high-probability bounds for various bandits setting. The main contribution is a new analysis technique which is based on increasing the learning rates using Online Mirror Descent. Using the new technique, the paper derives new high probability bounds in a variety of settings: MAB: showing first order high probability regret bound, which depends on L*, the loss of the best action Linear bandits: efficient high probability regret algorithm. adversarial MDP: High probability bounds that depend on the loss of the best policy (on the sequence of losses)

Strengths: The main strength of the paper is the new analysis to derive high probability regret abounds, and the applications to multiple settings (MAB, Linear bandits and adversarial MDP)

Weaknesses: none really.

Correctness: correct

Clarity: readble

Relation to Prior Work: done adequely.

Reproducibility: Yes

Additional Feedback: Following the rebuttal I did not change my score


Review 3

Summary and Contributions: The paper studies three bandit learning problems in the adversarial setting: the multi-armed bandit problem, the linear bandit problem, and learning finite horizon layered MDPs with stochastic but unknown transitions and adversarial rewards under bandit feedback. The main results are in the form of high probability regret bounds and high probability small-loss regret bounds. Main results can be found in Theorem 2.3, Theorem 3.1 and Theorem 4.1 respectively. A side result of independent interest is a slightly more general version of Freedman's inequality for martingale differences which is used throughout the paper and can be found in Theorem 2.2. --------------------------------- I have read the response of the authors and the other reviews. My main questions have been addressed and I keep my current score.

Strengths: To the best of my knowledge the results in Theorem 3.1 and Theorem 4.1 are novel. While similar results to Theorem 2.3 are available in prior work, the algorithm and approach for analysis is novel. In particular there are two main components contributing to the analysis of each of the algorithms -- one use the log-barrier OMD update with a non-decreasing step-size schedule, similar to Agarwal et. al to accumulate a negative regret term, and two, use the negative regret term to control the variance of the martingale difference sequence together with the provided version of Freedman's inequality. While, the negative regret term has appeared in prior work and has been useful mainly for combining multiple bandit algorithms or achieving quicker convergence to equilibria in games, the idea to use it as a term to control the variance of the regret is novel and could probably inspire new high-probability bounds for different bandit problems, apart from the ones studied in this current work.

Weaknesses: Overall I do not find any major weaknesses with the current work. However, I am not too familiar with the work of Jin et al. 2020 on which the algorithm for episodic MDPs and analysis builds. It seems in the prior work, Jin et al. have used OMD with an entropy regularizer and from what I understood the OMD step can be carried out efficiently. Can the same thing be claimed for the update on line 9 of Algorithm 4? Other minor questions and remarks: In Algorithm 2, how does one sample efficiently in line 2? Is the sampling equivalent to sampling from S^{d+1} and then projecting onto the subspace perpendicular to H_{t}^{-1/2}e_{d+1}? On page 13 of the appendix should A.3 be titled Proof of Theorem 2.3? On page 16 of the appendix, line 547 -- is there a missing inner product in the logarithm? On page 18, line 587, I couldn't understand how exactly 1-\pi_{w_1}(w_t) > 1/T.

Correctness: As far as I could verify the proofs in Appendix A and Appendix B are correct. I can not comment on Appendix C as I am not entirely familiar with the prior work on which the analysis builds.

Clarity: The paper is well written and the ideas are clearly presented with good intuition about how to derive the results.

Relation to Prior Work: Prior work is adequately discussed.

Reproducibility: Yes

Additional Feedback: Appendix C might benefit from including the procedure COMP-UCB from Jin et al. 2020.


Review 4

Summary and Contributions: The paper proposes a new approach to obtain high probability regret bounds for online learning with adversarial bandit feedback. The approach differs from prior work in their use of standard unbiased estimators and relying on a combination of (i) increasing learning rate schedule, (ii) logarithmically homogeneous self-concordant barriers, and (iii) a strengthened Freedman’s inequality.

Strengths: The regret bounds obtained are data-dependent which have greater practical usage than worst-case bounds General and efficient for linear adversarial bandits while previous approaches were either inefficient or applicable to action sets Application to learn adversarial MDPs with an algorithm for high probability small loss bound It helps solve the open problem for obtaining a high probability regret bound for adversarial linear bandits.

Weaknesses: The generalization to the MDP section feels hurried and it could be pretty dense for readers.

Correctness: The method seems largely correct though I didn't go through all the mathematical details.

Clarity: The paper is generally well motivated and well written, though it might be difficult to follow all the details if you are not very familiar with the area of bandit optimization in general.

Relation to Prior Work: Yes.

Reproducibility: Yes

Additional Feedback: May be better to provide some simulations or experiments on synthetic data (particularly to illustrate the regret bounds for MDP) to illustrate the performance of the proposed algorithms against prior art. I have read the authors' feedback and stick to my rating.

[Author Response · NeurIPS 2020]

**Reviewer #1**   We thank the reviewer for the valuable comments.

1. Large constants in $\Psi$ and $\kappa$:

   The constant 400 in $\Psi$ comes from Lemma B.5, which shows how to construct a normal barrier in the lifted domain from an arbitrary self-concordant barrier in the original decision set. However, as we mention in Footnote 2, our algorithm works with *any* normal barrier, not just this particular one (since our proofs only use those general properties of normal barriers from Section B.2). Therefore, the constant could be much smaller as long as a normal barrier with a smaller constant exists, which is indeed true for several canonical examples (see Nesterov and Nemirovskii, 1994). Similarly, the constant in $\kappa$ is determined by the constant in $\Psi$ (see the proof of Lemma B.12), and thus could be smaller as well.

   As we discuss in the paper, our algorithm is essentially SCRiBLe with a new sampling scheme and a new adaptive learning rate schedule. We thus believe that our algorithm is at least as empirically viable as SCRiBLe.

2. Other minor comments and missing related works:

   Thanks for pointing these out! We will revise the paper accordingly.

**Reviewer #2**   We thank the reviewer for the valuable comments.

**Reviewer #3**   We thank the reviewer for the valuable comments.

1. Efficiency of OMD with log-barrier regularizer for MDPs:

   In general, since the OMD step is a convex optimization problem with $\mathrm{poly}(|S|,|A|)$ linear constraints, one could apply any standard convex solver to implement the algorithm efficiently. Jin et al. 2020 were able to reduce the problem to another optimization problem with only positivity constraints due to the special structure of the entropy regularizer (which does not hold for log-barrier), but in the end they still require applying a convex solver to implement the OMD step.

2. How to sample $s_t$ efficiently:

   Yes, the way you mention is correct and efficient (except that in the end one also needs to normalize the norm to 1). Another efficient way to sample $s_t$ is to first uniformly randomly sample a point $s$ from a $d$-dimensional unit sphere, then let $s_t = H_t^{\frac{1}{2}} \begin{bmatrix} s \\ 0 \end{bmatrix}$, and finally normalize $s_t$. We will add these discussions to the final version.

3. Typos and clarification:

   Thanks for pointing out the typos on Page 13 and Page 16. We will revise the paper accordingly. As for the question regarding line 587, note that in the description of Algorithm 2, we restrict the choice of $w_t$ in $\Omega' = \left\{ \boldsymbol{w} = (w, 1) : w \in \Omega, \pi_{w_1}(w) \leq 1 - \frac{1}{T} \right\}$, therefore, we have $1 - \pi_{\boldsymbol{w}_1}(\boldsymbol{w}_t) = 1 - \pi_{w_1}(w_t) \geq \frac{1}{T}$.

**Reviewer #4**   We thank the reviewer for the valuable comments and suggestions, and will take them into account when revising the paper.

[Meta-Review · NeurIPS 2020]

There is a strong agreement among reviewers that this is a very good work.